# The fast-evolving FIKK kinase family of *Plasmodium falciparum* can be inhibited by a single compound

Hugo Belda[1,2,11], David Bradley[3,4,5,11], Evangelos Christodoulou [6], Stephanie D. Nofal [1,2], Malgorzata Broncel[1], David Jones [1], Heledd Davies[1], M. Teresa Bertran [7], Andrew G. Purkiss[6], Roksana W. Ogrodowicz[6], Dhira Joshi [8], Nicola O'Reilly [8], Louise Walport[7], Andrew Powell [9], David House [9], Svend Kjaer [6], Antoine Claessens [10], Christian R. Landry [3,4,5] & Moritz Treeck [1,2] ✉

Of 250 *Plasmodium* species, 6 infect humans, with *P. falciparum* causing over 95% of 600,000 annual malaria-related deaths. Its pathology arises from host cell remodelling driven by over 400 exported parasite proteins, including the FIKK kinase family. About one million years ago, a bird-infecting *Plasmodium* species crossed into great apes and a single non-exported FIKK kinase gained an export element. This led to a rapid expansion into 15–21 atypical, exported Ser/Thr effector kinases. Here, using genomic and proteomic analyses, we demonstrate FIKK differentiation via changes in subcellular localization, expression timing and substrate motifs, which supports an individual important role in host–pathogen interactions. Structural data and AlphaFold2 predictions reveal fast-evolving loops in the kinase domain that probably enabled rapid functional diversification for substrate preferences. One FIKK evolved exclusive tyrosine phosphorylation, previously thought absent in *Plasmodium*. Despite divergence of substrate preferences, the atypical ATP binding pocket is conserved and we identified a single compound that inhibits all FIKKs. A pan-specific inhibitor could reduce resistance development and improve malaria control strategies.

Malaria is caused by the infection of red blood cells (RBCs) with *Plasmodium* parasites. In 2022, 249 million infections and 608,000 deaths were observed, with severe cases occurring primarily in children under 5 years[1]. Among the six known *Plasmodium* species infecting humans, *P. falciparum* causes over 95% of all fatalities. This species remodels RBCs to cytoadhere to the host endothelium causing sequestration of infected RBCs (iRBCs), preventing passage through the spleen where iRBCs are destroyed. While benefiting the parasite, cytoadhesion can lead to blood clot formation in capillaries, reducing oxygen supply to highly vascularized organs such as the brain, lungs, kidneys or placenta in pregnant women.

*P. falciparum* exports ~10% of its proteome into the host cell[2]. Exported proteins fulfil a variety of functions in the iRBC[3]. They facilitate transport and anchoring of the major cytoadhesion ligand *P. falciparum* erythrocyte membrane protein 1 (*Pf*EMP1)[4] into parasite-derived structures underneath the erythrocyte membrane (knobs)[5], the creation of new nutrient permeability pathways in the plasma membrane[6,7] and the formation of intracytoplasmic membranous structures

(Maurer's clefts) involved in trafficking parasite proteins to the host cell surface[8].

Among the parasite exported proteins is a family of serine/threonine kinases called the FIKK (from a conserved Phe-Ile-Lys-Lys motif) kinases, which are exclusive to apicomplexan parasites[9]. While most *Apicomplexa* possess one non-exported FIKK (FIKK8 in *P. falciparum*), gene expansion in *P. falciparum* resulted in a family of 21 paralogues, including 2 predicted pseudogenes in the 3D7 reference genome[10]. All FIKKs, except for FIKK8, are predicted to be exported into the RBC. Only *Plasmodium* species from the *Laverania* subgenus (that includes *P. falciparum* and *Plasmodium* species infecting great apes[11,12]) possess the expanded FIKK family. No *Plasmodium* species outside the *Laverania* contains any predicted exported kinase. Ten of the 19 active *P. falciparum fikk* genes are conserved in syntenic loci in all *Laverania* species and all *fikk* genes are conserved in syntenic loci in at least 4 of the *Laverania* species (Supplementary Table 1, data from PlasmoDB[13–15]). The minimum number of FIKKs present in any *Laverania* species is 16. This number may be higher because of low-quality genome regions of some *Laverania* species. This indicates that *fikk* genes rapidly multiplied and diversified early during *Laverania* evolution. This expansion was followed by a long stasis in terms of the *fikk* copy number, suggesting that FIKKs play individually important roles in host–pathogen interactions in *Laverania* hosts.

At least one FIKK (FIKK4.1) is important for *Pf*EMP1 surface translocation and cytoadhesion[16], while FIKK4.2 is important for iRBC rigidification[17]. We previously observed no growth defect upon individual conditional deletion of any exported FIKK[16], suggesting that either exported FIKKs play no role in growth under standard cell culture conditions or that a level of redundancy exists between them. Determining the degree of redundancy between FIKKs is paramount to design experiments understanding their functions during *P. falciparum* infections.

The N-terminus of each FIKK is unique between paralogues, but conserved between orthologues. All FIKKs share a conserved C-terminal kinase domain containing the eponymous Phe-Ile-Lys-Lys (F-I-K-K) motif. FIKKs lack the glycine triad involved in binding ATP[9,18] but at least 14 FIKKs have demonstrated activity[16,17,19–26], indicating that ATP coordination is non-canonical. The unique ATP-binding pocket along with a small gate-keeper residue[27] found in most FIKKs may provide opportunities for developing highly specific pan-FIKK inhibitors targeting several FIKKs simultaneously.

Here, we provide evidence that a core set of FIKKs is under strong purifying selection and required for human infection. Our data suggest that FIKK specificity underwent a rapid diversification during their expansion, partly due to a fast-evolving loop in the kinase's substrate-binding region. This diversification appears conserved among distantly related *Plasmodium* species, suggesting evolutionary constraint linked to important functions in host–pathogen interaction with great apes and humans. Finally, we demonstrate that chemical inhibition of the FIKKs is achievable and that their highly conserved kinase domain allows for the development of pan-FIKK inhibitors.

## Results

### Potential overlapping and non-overlapping FIKK functions

To identify potentially overlapping functions between FIKKs, we searched for FIKKs that are co-expressed and co-localize. Nineteen active FIKKs appear to be transcribed in asexual parasite stages (Fig. 1a and Supplementary Table 2), although some only at very low levels. This is in line with our previous observation that some FIKKs are barely detectable as HA-tagged variants[16]. Their main subcellular localizations are the RBC periphery/knobs and Maurer's clefts or related structures (Fig. 1b). In addition to FIKK8 (refs. 16,25), four FIKKs (FIKK9.2 (ref. 19), FIKK3, FIKK9.5 (ref. 28) and FIKK5[16]) have reported localizations within the parasite, although antibodies raised against FIKKs have not been verified using available FIKK knockout lines.

Two *Pf*FIKKs (FIKK7.2 and FIKK14) are annotated as pseudokinases in the 3D7 reference genome but not in other genetic backgrounds.

Therefore, some FIKKs may still be evolving and dispensable for human infections. A search in 2,085 available field-isolate genomes revealed FIKKs with internal stop codons or deletions (Supplementary Tables 3 and 4), with 55.16% (1,150/2,085) and 2.73% (57/2,085) of all field isolates containing a stop codon in *fikk7.2* or *fikk14* genes, respectively. A total of 93.7% (1,078/1,150) of *fikk7.2* mutations are identical (W413*) and equally distributed between Southeast Asia (SEA) and Africa, suggesting an ancient origin. In contrast, *fikk14* shows premature stop codons throughout the gene, predominantly in African isolates (94.7% (54/57)), indicating that inactivating mutations in *fikk14* are not systematically eliminated by natural selection. Interestingly, 11.44% (137/1,197) and 12.05% (107/888) of African and SEA samples, respectively, have *fikk14* deletions, so the preponderance for stop codons in *fikk14* in African isolates is not observed for gene deletions (Supplementary Table 4). *fikk9.2* encodes an active kinase in the 3D7 reference strain, but 4.65% (97/2,085) of the field-isolate genomes contain an internal stop codon, mainly in SEA isolates (86.6% (84/97)). However, prevalences from SEA should be interpreted with caution as antimalarial drugs have recently reduced the *P. falciparum* population genetic diversity.

A dN/dS ratio analysis of all FIKK sequences in the MalariaGen database[29,30] shows that FIKKs not annotated as pseudokinases have a value <1 (range of 0.31–0.94, mean of 0.54) (Supplementary Table 5). This suggests that a core set of 18 FIKKs is evolving under purifying (that is, important) selection in humans, while three FIKKs are under relaxed selection, since inactivating mutations can arise in the field. Relaxed selection could come from redundancy among the FIKKs. Alternatively, since these kinases probably evolved in the great ape-infecting ancestors of *P. falciparum*, they may have fulfilled important functions which are now expendable during human infection. An argument for the latter hypothesis is the observation that both *fikk7.2* and *fikk14* are predicted to be functional in *P. praefalciparum*, *P. gaboni* and *P. adleri*[12].

At least 12 FIKKs appear expressed in gametocytes[31], the sexual stages of the parasite that are ingested by a mosquito for onwards transmission (Supplementary Table 6). We confirmed the expression of FIKK1, FIKK4.1, FIKK8, FIKK9.3, FIKK12 and FIKK13 in stage III gametocytes using HA-tagged conditional knockout lines[16] (Fig. 1c). Transcriptomic data also suggest that some FIKKs may be expressed during liver infection and/or in parasite stages present in the mosquito, although this has not been experimentally tested (Supplementary Table 6).

Collectively, these data suggest that some FIKKs are separated in time and space within the iRBC and therefore probably evolved unique functions. Other FIKKs however have similar localizations within the cell and partially overlapping expression timings and could therefore have functional overlaps.

### FIKK4.1 and FIKK4.2 partially overlap subcellularly

To test whether co-localizing FIKKs partially overlap in their function, we focused on FIKK4.1 and FIKK4.2. These two kinase genes are phylogenetically closely related (Extended Data Fig. 1) and probably originated from a gene duplication event. Both proteins co-localize at the iRBC periphery (Fig. 2a) where they phosphorylate host cytoskeleton and exported parasite proteins (Fig. 1b, top right)[16]. FIKK4.1 deletion, but not FIKK4.2, reduces *Pf*EMP1 surface translocation by ~50% (refs. 16,32). While this demonstrates that FIKK4.2 cannot fully compensate for FIKK4.1 deletion, a partial substrate overlap and partial rescue cannot be excluded.

To gain high-resolution information on their subcellular localization, we determined their local protein environment by proximity labelling using TurboID fusion proteins[32,33] (Supplementary Fig. 1) and mass spectrometry (Fig. 2b) and overlaid this proximity network with previously obtained phosphoproteome data for the FIKKs[16]. Ninety-one proteins were biotinylated by FIKK4.1::TurboID and FIKK4.2::TurboID and are therefore in close spatial proximity to both kinases (Fig. 2c). However, each of the two fusion proteins also labels a unique subset of proteins indicating that they are not in identical locations. We found no

**a**

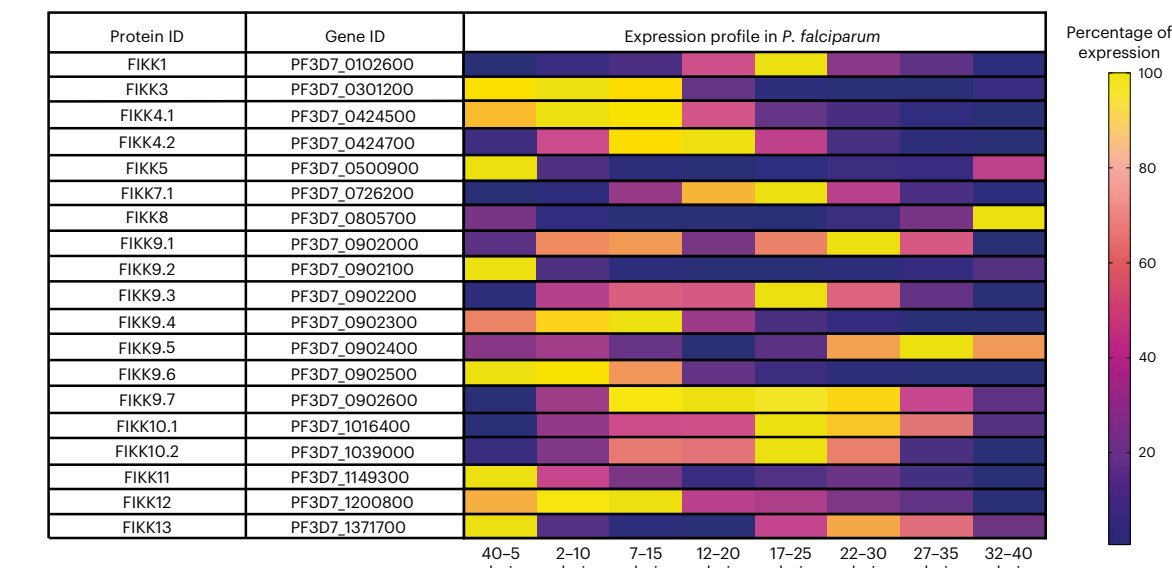

**b**

**c**

**Fig. 1 | Expression timings and localizations of *P. falciparum* FIKK kinases.**
**a**, A heat map built using data from Hoeijmakers et al.[74] RNA-sequencing dataset available on PlasmoDB (www.PlasmoDB.org), showing the percentage expression for each FIKK during the *P. falciparum* asexual replication cycle. Yellow, maximum expression; dark blue, minimum expression. The TPM values used to calculate the percentage expression relative to maximum expression across the 48 h lifecycle are available in Supplementary Table 2. **b**, A diagram illustrating *P. falciparum* FIKKs expression and localizations in iRBCs. Top left: an illustration of the *P. falciparum* lifecycle. FIKK1, FIKK4.1, FIKK8, FIKK9.3, FIKK12 and FIKK13 expressed in gametocytes are shown in orange. Bottom

right: *P. falciparum* iRBC showing the localization of the FIKKs. Blue, FIKK3, FIKK5, FIKK8, FIKK9.2 and FIKK9.5 in the parasite; green, FIKK1, FIKK7.1, FIKK9.1, FIKK9.3, FIKK10.1, FIKK10.2 and FIKK12 in Maurer's clefts; red, FIKK1, FIKK4.1, FIKK4.2 and FIKK12 at the RBC periphery. *Localization data from publications from other laboratories. Top right: knob structure at the RBC periphery. Yellow stars show FIKK4.1 substrates and orange stars show FIKK4.2 substrates (data from ref. 16). EDVs, electron dense vesicles. **c**, Western blots confirming expression of HA-tagged *P. falciparum* FIKKs in gametocytes stage III. GAP50 antibody (bottom) demonstrates equal loading. The arrows show FIKK bands at expected sizes (shown in the labels at the top). †Rapamycin treatment.

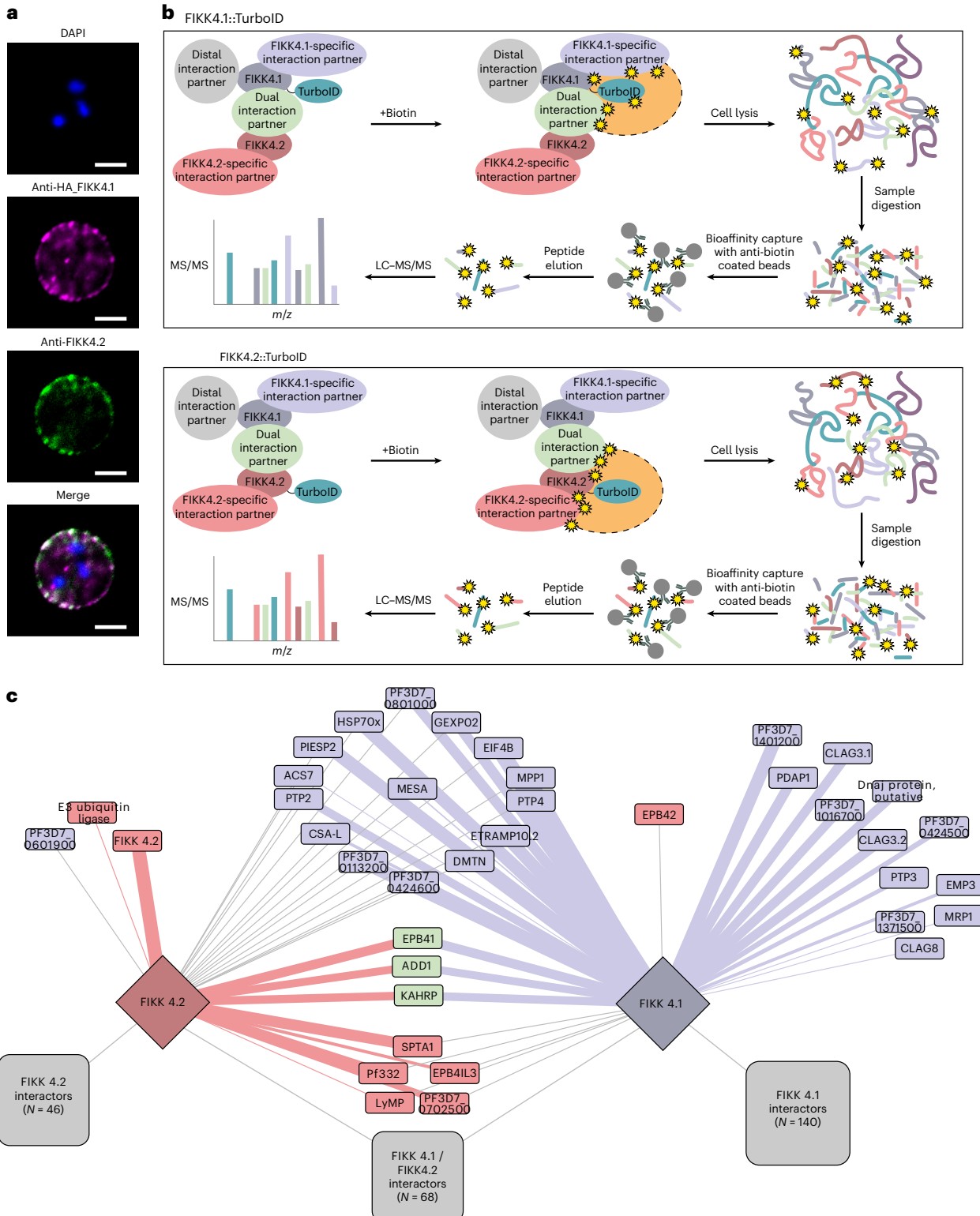

**Fig. 2 | Investigation of FIKK4.1 and FIKK4.2 local protein environment.**
**a**, The subcellular localization of FIKK4.1 and FIKK4.2 investigated by immunofluorescence assay using anti-HA antibodies (magenta) targeting the C-terminally HA-tagged FIKK4.1 and anti-FIKK4.2 antibodies (green). DAPI (blue) is used for nuclear staining. Scale bars, 5 µm. The assay was performed three times with similar results. **b**, A diagram representing the proximity labelling workflow. FIKK4.1 (top) and FIKK4.2 (bottom) were tagged with a TurboID biotin ligase. Upon addition of biotin, proteins in the vicinity (represented by an orange area with a dashed outline) of the bait are biotinylated on lysine residues (represented by yellow stars). iRBCs were lysed in 8 M urea in 50 mM

HEPES and proteins were trypsin digested into peptides. Biotinylated peptides were enriched using beads coated with two different anti-biotin antibodies and analysed by LC–MS/MS. **c**, A network analysis of FIKK4.1 and FIKK4.2::TurboID data. The connecting lines indicate a protein that is probably in the vicinity of the TurboID-tagged protein. Blue depicts proteins identified as potential FIKK4.1 direct targets in ref. 16. Red depicts proteins that have been identified as potential FIKK4.2 direct targets, and green depicts proteins identified as potential targets of both FIKK4.1 and FIKK4.2. The thickness of the connection represents how well the phosphorylation site matches the corresponding in vitro preferred phosphorylation motifs (of FIKK4.1 or FIKK4.2) from Supplementary Fig. 3.

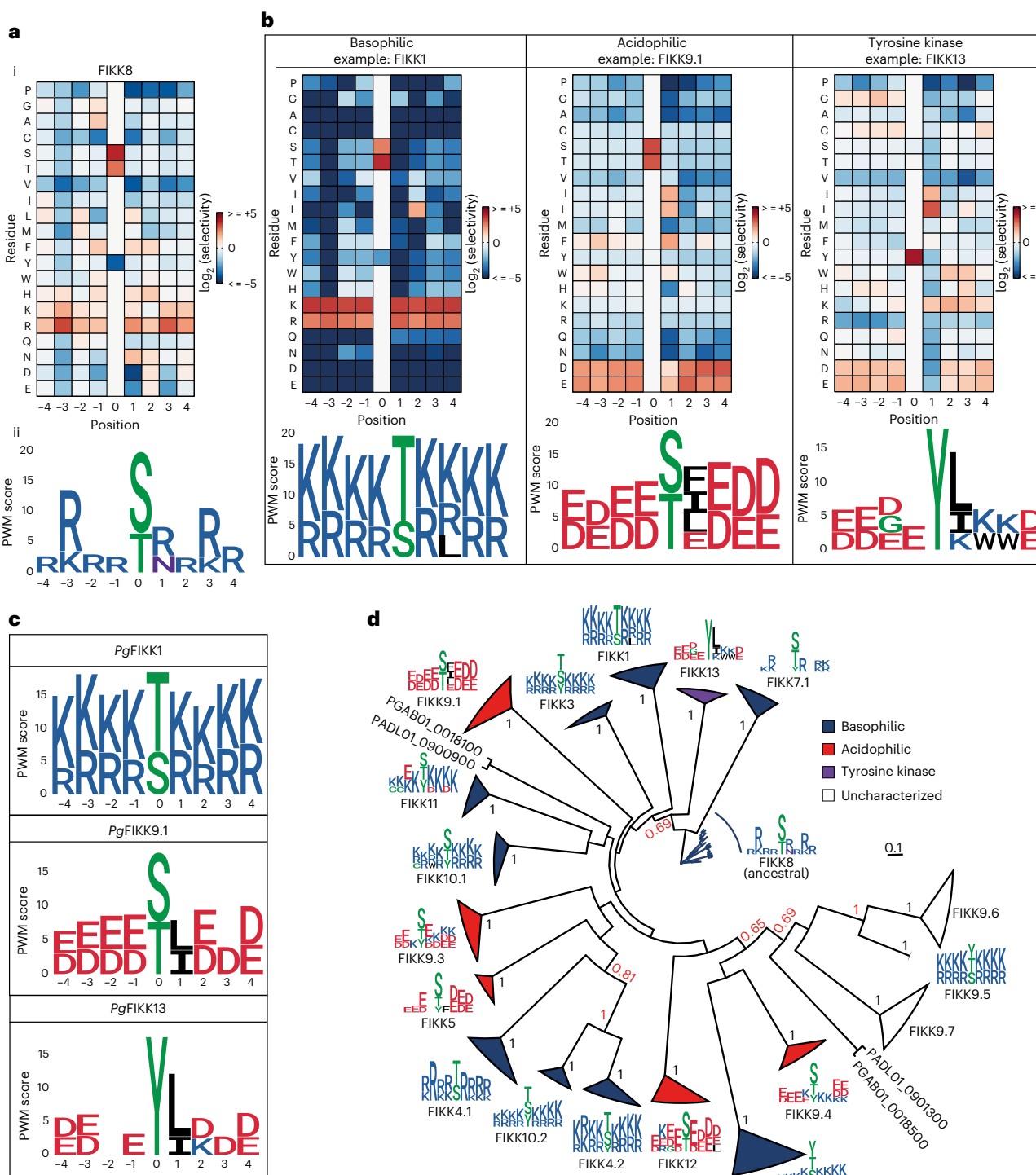

**Fig. 3 | FIKK kinases evolved divergent substrate specificities conserved among *Laverania* species. a**, Extended Data Fig. 3 data represented as a heat map (i). The $^{32}$P incorporation values were normalized to 20 (the number of possible natural amino acids) and are shown as $\log_2(x)$ where negative values (blue cells) indicate disfavoured amino acids and positive values (red cells) indicate favoured amino acids. A PWM logo generated with FIKK8 raw OPAL data (ii). PWMs depict the preference of the kinase for all 20 amino acids at every substrate position. For ease of visualization, the PWM logo displays amino acids with scores above an arbitrary threshold of 2.5 (Methods). Amino acid colours are set as follows: acidic negatively charged (D, E), red; basic positively charged (R, K, H), blue; polar uncharged (N, Q), purple; non-polar (A, I, L, M, F, V, P), black; phosphorylatable or special (S, T, Y, C, G), green. **b**, A heat map representation of OPAL data for basophilic FIKK1 (left), acidophilic FIKK9.1 (middle) and tyrosine kinase FIKK13 (right). PWM logos generated from raw OPAL data are

displayed below the corresponding heat maps. OPAL membrane images are available in Supplementary Fig. 5. **c**, PWM logos generated with *Pg*FIKKs raw OPAL data. See **a**(i) caption and Supplementary Fig. 6. **d**, A maximum-likelihood phylogenetic tree of *Laverania* FIKK sequences built using FIKK8 kinases and two avian malaria FIKKs (*P. relictum* FIKK PRELSG_0112400 and *P. gallinaceum* FIKK PGAL8A_00108200) as an outgroup. One hundred bootstrap replicates were generated to assess branch support[89]. All orthologue clades have maximum branch support (one out of one). Branches between paralogues are highlighted in red if they are >0.5. The triangle length represents the divergence between FIKK sequences within a specific clade. The colour code identifies the kinases substrate specificities as follows: blue, basophilic; red, acidophilic; purple, tyrosine kinase; white, uncharacterized. Sequence logos for each clade are given for the *P. falciparum* kinase copy. The tree contains $N = 131$ sequences in total.

evidence of reciprocal biotinylation of FIKK4.1- and FIKK4.2::TurboID fusions (Supplementary Table 7). Only three proteins (α-adducin, protein 4.1 and KAHRP) were found phosphorylated by both FIKK4.1 and FIKK4.2 but on non-overlapping residues. Phosphorylation of all other proteins in proximity of both kinases is exclusively dependent on only one of the two kinases. These data suggest that FIKK4.1 and FIKK4.2 are in very close, but not direct, proximity and evolved to phosphorylate different targets.

## FIKKs evolved unique phosphorylation motifs

To understand how FIKKs may have evolved to phosphorylate specific targets, we determined their preferred phosphorylation motifs. Most eukaryotic protein kinases (ePKs) preferentially phosphorylate S, T or Y residues within a specific amino acid sequence context (motif). These motifs are broadly classified into acidic, basic or proline directed[34]. *P. falciparum* kinases phosphorylate S and T residues within acidic and basic motifs. Phosphorylated Y residues and proline-directed motifs are rarely found[35] and the *Plasmodium* kinome does not include enzymes from the TyrK group[36].

We recombinantly expressed the kinase domains of all predicted active *Pf*FIKKs (Extended Data Fig. 2, Supplementary Fig. 2 and Supplementary Table 8) and assessed substrate specificity on S, T and Y residues using Oriented Peptide Array Library (OPAL) libraries[37]. Of the 19 FIKKs, only FIKK9.6 and FIKK9.7 were refractory to expression.

As previously reported[22], FIKK8, probably the closest relative to the ancestral kinase from which all FIKKs in *Laverania* evolved, shows a preference for basic residues (Fig. 3a(i) and Extended Data Fig. 3). Position weight matrices (PWMs) indicate strong preference for arginine and/or lysine residues in position P−3 and P+3 (Fig. 3a(ii)). Eleven FIKKs prefer basic and positively charged amino acids surrounding the phosphorylated residue, while five FIKKs favour acidic motifs (Fig. 3b and Supplementary Figs. 3 and 4). Within both groups, nuanced preferences emerge. FIKK1 strongly prefers a hydrophobic residue in the P+2 position, distinguishing it from other FIKKs. FIKK9.3 and FIKK9.4 show both basic and acidic residues in the motif; here, assignment to a group was based on the dominant charge.

Interestingly, we observed several FIKKs that phosphorylate Y-based peptides. For several (FIKK5, FIKK8, FIKK9.1 or FIKK12), S/T residues in the flanking regions of the central Y may explain the signal, while others (FIKK3, FIKK4.2, FIKK9.2, FIKK9.3, FIKK9.4, FIKK9.5 and FIKK11) exhibited dual specificity. FIKK13 showed exclusively Y phosphorylation activity (Fig. 3b). This is surprising considering the absence of known bona fide tyrosine kinases in *Plasmodium* or any *Apicomplexa*[38]. This suggests that FIKK13 has evolved from a S/T kinase into a tyrosine kinase, potentially to interact with specific host cell proteins. To validate this result, we screened a DNA-encoded cyclic peptide library (RaPID selection; Extended Data Fig. 4a)[39–42] and enriched four cyclic peptides that bind to FIKK13 (FIKK13_2 $K_D$ = 310 ± 290 nM, FIKK13_3 $K_D$ = 7 ± 5 nM, FIKK13_4 $K_D$ = 17 ± 0.4 nM and FIKK13_5 $K_D$ = 120 ± 156 nM) (Extended Data Fig. 4b and Supplementary Table 9). One peptide (FIKK13_4: cyclic-d(Y)PLRFLSKYHC(S-)G-CONH₂) was identified as an in vitro substrate for FIKK13 and phosphorylation depended on the tyrosine residue (Extended Data Fig. 4c,d) lending further support for FIKK13 tyrosine kinase function.

In summary, FIKKs evolved divergent substrate specificities from a basophilic ancestor, thereby expanding the repertoire of proteins the parasite can regulate. The motif diversity of the FIKKs highlights the rapid evolution of this relatively young protein family[14], probably driven by selection to subvert various host cell functions in great apes. This contrasts with ancient kinase families such as CK1, MAPK or PKA kinases that possess more conserved phosphorylation motifs[43–46].

## Orthologous conservation of FIKK phosphorylation motifs

To confirm the tyrosine specificity of FIKK13, we expressed its orthologue from *P. gaboni*, the most distantly related *Laverania* species to *P. falciparum*, estimated to have diverged ~1 million years ago. If *Pg*FIKK13 is evolving under purifying selection against changes to specificity, it should possess a similar tyrosine-based preferred phosphorylation motif. We also expressed *Pg*FIKK1 and *Pg*FIKK9.1 to test whether basophilic and acidophilic charge preferences are conserved. Strikingly, the motifs of the *P. falciparum* FIKKs are nearly identical to their *P. gaboni* orthologues (Fig. 3c and Supplementary Fig. 6). FIKK orthologues are more similar between species than paralogues within the same species (Fig. 3d). The strong motif preference observed for *Pf* and *Pg*FIKK1, 9.1 and 13 suggests that FIKK substrate specificity is probably conserved across all *Laverania* species. We also observe almost equal divergence between FIKK paralogues in terms of their sequence identity (Extended Data Fig. 5). Therefore, in most cases, the precise evolutionary relationship between paralogues cannot be resolved with confidence, in agreement with the rapid and early diversification of the FIKK family. Exceptions are the pair of recent paralogues FIKK9.5–FIKK9.6, and the FIKK4.1–FIKK4.2–FIKK10.2 clade that are both predicted with high confidence (Fig. 3d and Extended Data Fig. 5).

The broad diversity and deep conservation between orthologues of phosphorylation motifs suggest that FIKK8's ancestor rapidly diversified into 16+ exported copies in the *Laverania* subgenus.

A global dN/dS ratio for FIKK sequences across all *Plasmodium* species (*Laverania* and non-*Laverania*) was calculated using the phylogenetic tree in Fig. 3d and revealed purifying selection on all FIKKs (dN/dS of 0.25). However, the dN/dS ratio for the non-exported FIKK is significantly lower (0.09) compared with exported FIKKs (0.34),

---

**Fig. 4 | Investigation of FIKK1, FIKK4.1 and FIKK4.2 substrate specificities.**
**a**, Left: a representation of FIKK phosphoproteome peptides membrane, each dot represents one target-peptide species. Yellow, FIKK1 targets; orange, FIKK4.1 targets; green, FIKK4.2 targets; blue, FIKK10.2 targets; red, host cell peptides phosphorylated in iRBCs. The list of peptides with sequences is provided in Supplementary Table 10. Right: FIKK1 activity on the phosphoproteome peptides membrane. **b**, Left: correlation of FIKK1 activity on the phosphoproteome peptides membrane (log₁₀ transformed) against FIKK1 motif score (matrix similarity score) for each peptide (*n* = 163). Pearson's correlation for the *y* = log(*x*) curve. Right: FIKK1 phosphorylation signal (Phosphosignal) (log₁₀ transformed) for peptides with or without matching FIKK1 motif, for peptides with an S (*n* = 269, Cohen's *D* = 1.2, *P* = 1.5 × 10⁻⁸, Wilcoxon test, one sided) or T phosphoacceptor (*n* = 95, Cohen's *D* = 0.57, *P* = 0.13, Wilcoxon test, one sided). The centre line shows the median, the box limits the upper and lower quartiles, the whiskers show 1.5× the interquartile range and each point indicates an outlier. **c**, Left: FIKK1 specificity logo derived from the OPAL membrane. Right: FIKK1 specificity logo derived from natural peptides phosphorylated by FIKK1 on the peptide membrane. **d**, Left: FIKK4.1 activity (log₁₀ transformed) against predicted target peptides of FIKK4.1 (orange) and of FIKK4.2 (green) (*n* = 174 peptides, Cohen's *D* = 0.57, *P* = 4.0 × 10⁻³, Wilcoxon test, one sided). Right: FIKK4.2 activity (log₁₀ transformed) against predicted target peptides of FIKK4.1 (orange) and of FIKK4.2 (green) (*n* = 174 peptides, Cohen's *D* = 0.10, *P* = 0.77, Wilcoxon test, one sided). The black dot shows the median and the black line shows the upper and lower quartiles. **e**, FIKK4.1 and FIKK4.2 activity on KAHRP_345, PTP4_1091 and PIESP2_267. The results are represented as mean ± s.e.m. fold change compared with the no substrate luminescent signal. Statistical significance was determined using a one-way analysis of variance followed by Dunnett's multiple comparison post-test (for FIKK4.1, KAHRP_345 versus no substrate, *P* < 0.0001; PTP4_1091 versus no substrate, *P* < 0.0001; PIESP2_267 versus no substrate, *P* = 0.0001; FIKK4.2, KAHRP_345 versus no substrate, *P* = 0.9658; PTP4_1091 versus no substrate, *P* = 0.9069; PIESP2_267 versus no substrate, *P* = 0.4315) (*n* = 3 biological replicates). **f**, FIKK4.1 and FIKK4.2 PWM logos made using data from Supplementary Fig. 3. Values are log₂ transformed. A positive value depicts favoured amino acids and a negative value depicts disfavoured amino acids. See Fig. 3 caption for the colour code and Supplementary Fig. 8 for log₂-transformed PWM logos for all recombinant FIKK tested.

indicating relaxed purifying selection during the evolution of exported kinases ($P = 3.05 \times 10^{-135}$, d.f. of 1, likelihood ratio test).

## Phosphorylation motifs divergence between co-localizing FIKKs

FIKK1, FIKK4.1 and FIKK4.2 localize to the RBC periphery (Fig. 1) and are basophilic, although their specificity maps differ slightly. FIKK1 prefers a hydrophobic leucine residue in the P+2 position, while FIKK4.1 has a strong preference for an arginine residue in the P−3 position (Fig. 3 and Supplementary Fig. 3).

To test whether the in vitro phosphorylation motifs identified here match the targets we previously identified by conditional FIKK deletion and phosphoproteomics in cell culture[16], we performed activity assays on membranes containing 215 peptides predicted to be targets of FIKK1, FIKK4.1 and FIKK4.2. We also included 89 peptides targeted by FIKK10.2, a basophilic FIKK that localizes to the Maurer's clefts, and 93 peptides from host cell proteins found to be more phosphorylated upon *P. falciparum* infection (Fig. 4a and see Supplementary Table 10 for peptides sequences). The membranes were incubated with either recombinant FIKK1, FIKK4.1, FIKK4.2

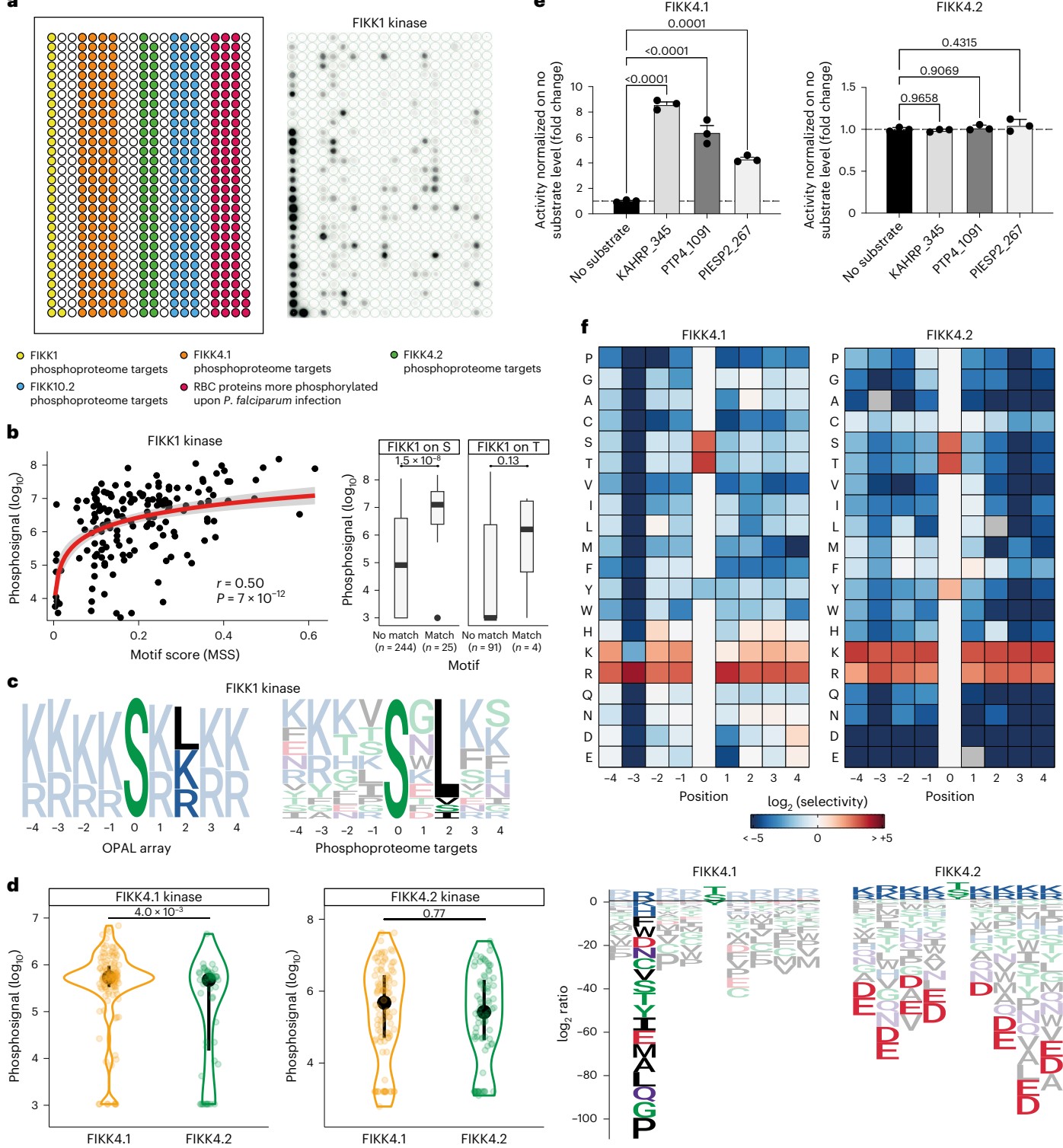

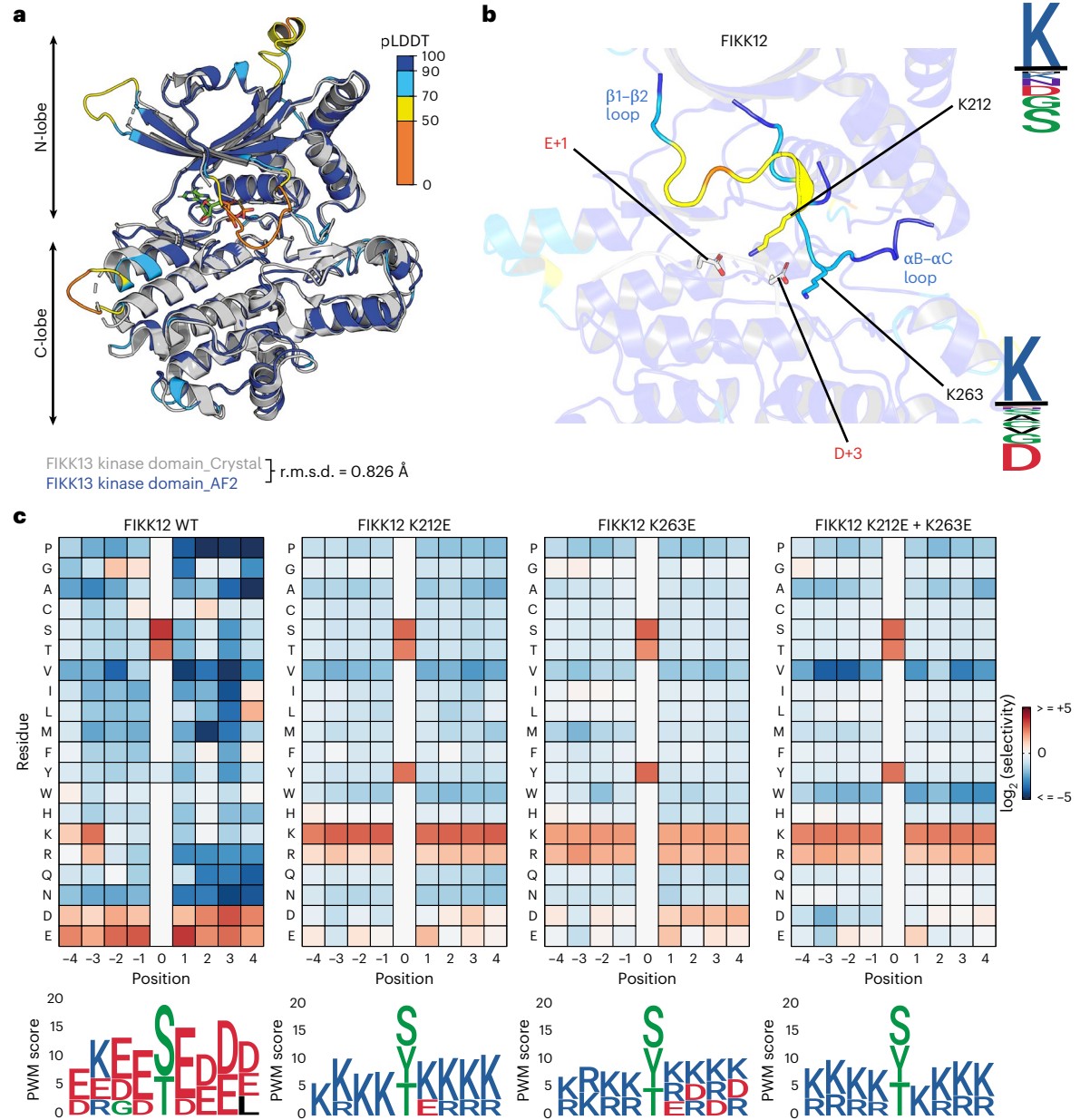

**Fig. 5 | Mutating specificity determinant residues identified using FIKK13 D379N structure allows for changes in FIKK substrate specificity. a**, Overlay of FIKK13 D379N kinase domain crystal structure with ATPγS (grey) and the FIKK13 kinase domain AlphaFold structure prediction coloured according to the residues pLDDT score. The r.m.s.d. was calculated using PyMol[115]. **b**, A target peptide (EKKASEGDN) of FIKK12 was modelled into the substrate-binding groove of the FIKK12 AlphaFold structure (Methods). The K212 and K263 kinase residues are predicted to bind to the peptide at the +1 and +3 positions. K212 is found on the large β1–β2 loop that is N-terminal to the first β-strand on the kinase N-lobe. Boundaries of the β1–β2 loop lies between residues 202 and 217

(inclusive). K263 is found on the loop between the αB and αC helices (boundaries between residues 257 and 267 (inclusive)). Secondary structure regions are referred to (for reference) in Extended Data Figs. 6 and 10. The sequence logos show the residue conservation between FIKK12 *Plasmodium* sequences (top), and basophilic *Plasmodium* sequences (bottom). The FIKK12 kinase domain is coloured according to the pLDDT score, the same as for **a. c**, FIKK12 WT and FIKK12 mutants phosphorylation activity on OPAL membranes represented as heat maps (see Fig. 3a(i) caption). Below is represented the PWM logos (see Fig. 3a(ii) caption).

or FIKK10.2 kinase domains and [γ-32P]-ATP (Fig. 4a and Supplementary Fig. 7a).

The natural peptides strongly phosphorylated in this assay largely concur with the OPAL phosphorylation motifs (Fig. 4b and Supplementary Fig. 7b). This is shown in Fig. 4c for FIKK1 confirming its basophilicity and the importance of a leucine at the P+2 position—a FIKK1 signature—making it the most highly specific kinase from this dataset.

FIKK4.1 strongly phosphorylates FIKK4.1 target peptides, while FIKK4.2 cannot discriminate between FIKK4.1 and FIKK4.2 substrates

(Fig. 4d). However, three peptides previously identified as FIKK4.1 substrates[16] (KAHRP_345: GSRYS**S**FSSVN; PTP4_1091: HTRSM**S**VANTK; PIESP2_267: EIRQE**S**RTLIL) are exclusively phosphorylated by FIKK4.1 but not FIKK4.2 (Fig. 4e). Analysing disfavoured amino acids in the OPAL libraries data (Supplementary Fig. 8) reveals differences between FIKK4.1 and FIKK4.2, with FIKK4.2 disfavouring negatively charged amino acids, whereas FIKK4.1 can accommodate more variety, except in position P−3 (Fig. 4f). This aligns with FIKK4.1's strong arginine preference at P−3, a specificity determinant present in all three peptides

tested in Fig. 4e. Collectively, these data show that the FIKKs evolved distinct phosphorylation motifs allowing the specific regulation of targets in certain subcellular contexts.

## FIKK structure informs on specificity determinant residues

We determined the crystal structure of the FIKK13 kinase domain harbouring a catalytic Asp-379 mutation (D379N) to prevent autophosphorylation (Extended Data Fig. 6a), which could introduce microheterogeneity adversely affecting protein crystal formation. Crystallization was facilitated by two anti-FIKK13 nanobodies generated through llama immunization (Extended Data Fig. 6b). FIKK13 kinase domain was co-crystallized with the non-hydrolysable ATP-analogue ATPγS and adopts, despite the low sequence identity, the classical ePKs bi-lobal fold[47] with a few notable additions, shown and further described in Extended Data Fig. 6c. Alignment of the Alpha-Fold model and the experimental structure of FIKK13 kinase domain revealed substantial overlap (root mean squared deviation (r.m.s.d. of 0.826 Å)) (Fig. 5a) affirming the accuracy of AlphaFold models, not only for FIKK13 but probably for other FIKK kinase domains, as the average predicted local distance difference test (pLDDT) score of the FIKK kinase domains is high (88.8) (see Supplementary Table 11 for pLDDT and r.m.s.d. values for FIKK AlphaFold models superimposed to the non-exported FIKK8).

Molecular docking could not reveal the basis for the tyrosine specificity of FIKK13 compared with other FIKKs, but combined with AlphaFold models, successfully predicted specificity determinants for other FIKKs. Modelling interactions of potential FIKK target peptides[16] or preferred phosphorylation motifs (Fig. 3 and Supplementary Figs. 3 and 4) predicted specific residues in several FIKKs as specificity determinants: FIKK1 (E517 and E522), FIKK1 (V321), FIKK9.1 (K240) and FIKK12 (K212 and K263) (Fig. 5b and Extended Data Fig. 7). To test the predictions, we reversed the charges of the amino acids (positively charged K mutated to negatively charged E and vice versa) or replaced the hydrophobic V321 in FIKK1 with a charged D. A single mutation in FIKK12 (K212E or K263E) shifted the substrate specificity from acidophilic to basophilic for all positions in the preferred phosphorylation motif (Fig. 5c). The double mutation (K212E + K263E) achieved total conversion to basophilicity. For FIKK1 and FIKK9.1, the changes were more subtle, with an increased overall preference for oppositely charged residues but no complete inversion (Extended Data Fig. 8a). Mutation of the V321 residue in FIKK1, homologous to K263 in FIKK12, was sufficient for ablating the leucine specificity at P+2 for this kinase (Extended Data Fig. 8b). A similar effect could be observed for FIKK1 E517K + E522K, but not for the single mutants. Therefore, E517 and E522 combined could be required for optimal positioning of the peptide leading to the loss of the P+2 specificity when mutated.

Thus, a single mutation in the kinase domain can dramatically change the preferred phosphorylation motif of FIKK12, and, to a lesser extent, that of FIKK1 and FIKK9.1. In contrast to canonical kinases, where peptide specificity is largely determined by cognate subpockets on the kinase domain[48–51], the validated determinants FIKK12 K212 and K263 map to kinase loop regions. These loop regions are rapidly evolving (Extended Data Fig. 9) and probably flexible given their low pLDDT scores in the AlphaFold models[52,53].

## Identification of pan-FIKK specific inhibitors in vitro

The structural analysis of the kinase domain ATP-binding site revealed some conserved features among the FIKKs that distinguish them from most ePKs. (1) The glycine loop found in ePKs, known to position ATP for catalysis, is not present in the FIKKs. An equivalent loop exists but has a low degree of conservation among family members and is unstructured in the experimental FIKK13 structure. However, a basic residue (K/R) is conserved throughout the FIKKs at the position of Lys-205 (Extended Data Fig. 10) and could help position ATP for its catalysis. (2) The F-I-K-K motif plays a role in ATP binding, or in this case, ATPγS. The invariant Phe-228 (Extended Data Fig. 10) is stacked upon the adenine in the back of the nucleotide-binding pocket (Extended Data Fig. 6d). The bulky and hydrophobic nature of the Phe side-chain reduces the size of the back-pocket with the equivalent residues in ePKs often having small side chains such as Ala, or in rare instances Val[54]. ATP coordination is probably supported by Lys-230, equivalent to Lys-72 in PKA[47], which coordinates the phosphates of the nucleotide, thereby sensing nucleotide pocket occupation and forms a salt bridge with the conserved Glu-261 on the C-helix (Glu-91 in PKA), a hallmark of active ePKs[47]. (3) Most FIKKs possess a small gatekeeper residue not found in most human kinases[55]. These fundamental, and conserved, differences in the nucleotide-binding pocket could enable drug development specifically targeting the FIKK family.

We first tested six different staurosporine analogues, which inhibit the majority of human kinases (>85%) by competing with ATP[56–58] on recombinant FIKK8. None inhibited FIKK8 activity (Fig. 6a), highlighting the distinctive features of the FIKK kinase domain. A screen of the published kinase inhibitor set (PKIS) library (containing 868 ATP analogues), developed for human kinases[59,60], identified 12 compounds inhibiting FIKK8 activity by >75% at 10 μM concentration (Fig. 6b). Their median inhibitory concentration ($IC_{50}$) values ranged between 11 nM and 332 nM (Supplementary Table 12). Further biochemical screening revealed structure–activity relationships (SAR) for several analogues, including some close structural analogues with weak FIKK potency that could serve as negative compounds. From this set, three compounds were prioritized as inhibitor tools for FIKKs. GW779439X and GSK2181306A are potent FIKK inhibitors from different chemical series. GSK3184025A was selected as a weakly active (>10 μM) compound from the same chemical series as GW779439X (Fig. 6c). Most recombinant FIKK kinase domains were inhibited in vitro by either GW779439X or GSK2181306A (Fig. 6d), while no inhibition was observed with GSK3184025A.

**Fig. 6 | PKIS library screen allows for the identification of several pan-FIKK kinases inhibitors that target at least one FIKK kinase in ATP-depleted iRBCs. a**, FIKK8 activity in the presence of increasing concentrations of staurosporine analogues (GW272220X, SB-219551, GW442389X, GW471214X, GW470969X and SB-505576). $n = 6$ technical replicates for each inhibitor. Shown is the mean ± s.e.m. **b**, A ranked plot showing the results of the PKIS library screen on recombinant FIKK8. A threshold of >75% inhibition was arbitrarily set and identified the 12 most potent PKIS compounds on recombinant FIKK8 kinase domain ($n = 2$). Each data point represents the mean percentage inhibition in both replicates. **c**, A SAR assay identifies closely related compounds with different behaviours towards recombinant FIKK8 kinase domain. A total of 333 compounds were identified from the three original PKIS chemical templates. The $IC_{50}$ on recombinant FIKK8 kinase domain was measured in biological triplicate for each one of the compounds and are indicated here for the selected ones ±s.d. **d**, A heat map representing inhibition (%) of selected compounds on recombinant FIKK kinase domains ($n = 3$ biological replicates). **e**, Western blot showing adducin S726 phosphorylation in RBCs pretreated with 1,228 μM iodoacetamide and 2,046 μM inosine, infected with WT NF54 *P. falciparum* and treated with different concentrations of either GSK2236790B or GW779439X. The GAP50 antibody demonstrates equal loading. **f**, Western blot showing adducin S726 phosphorylation in RBCs pretreated with 1,228 μM iodoacetamide and 2,046 μM inosine, infected with FIKK1 conditional knockout (cKO) DMSO-treated *P. falciparum* and treated with different concentrations of either GSK2236790B or GW779439X. GAPDH antibody demonstrates equal loading. **g**, Immunofluorescence assays showing adducin S726 phosphorylation and protein export in ATP-depleted iRBC treated with different concentrations of either GSK2236790B or GW779439X. Protein export is investigated with anti-HA antibodies targeting the C-terminal HA-tag fused to FIKK1 kinase domain and with anti-SBP1 antibodies. DAPI (blue) is used as a nuclear staining. Scale bars, 5 μm. The immunofluorescence assays in **e**, **f** and **g** were performed at least three times with similar results.

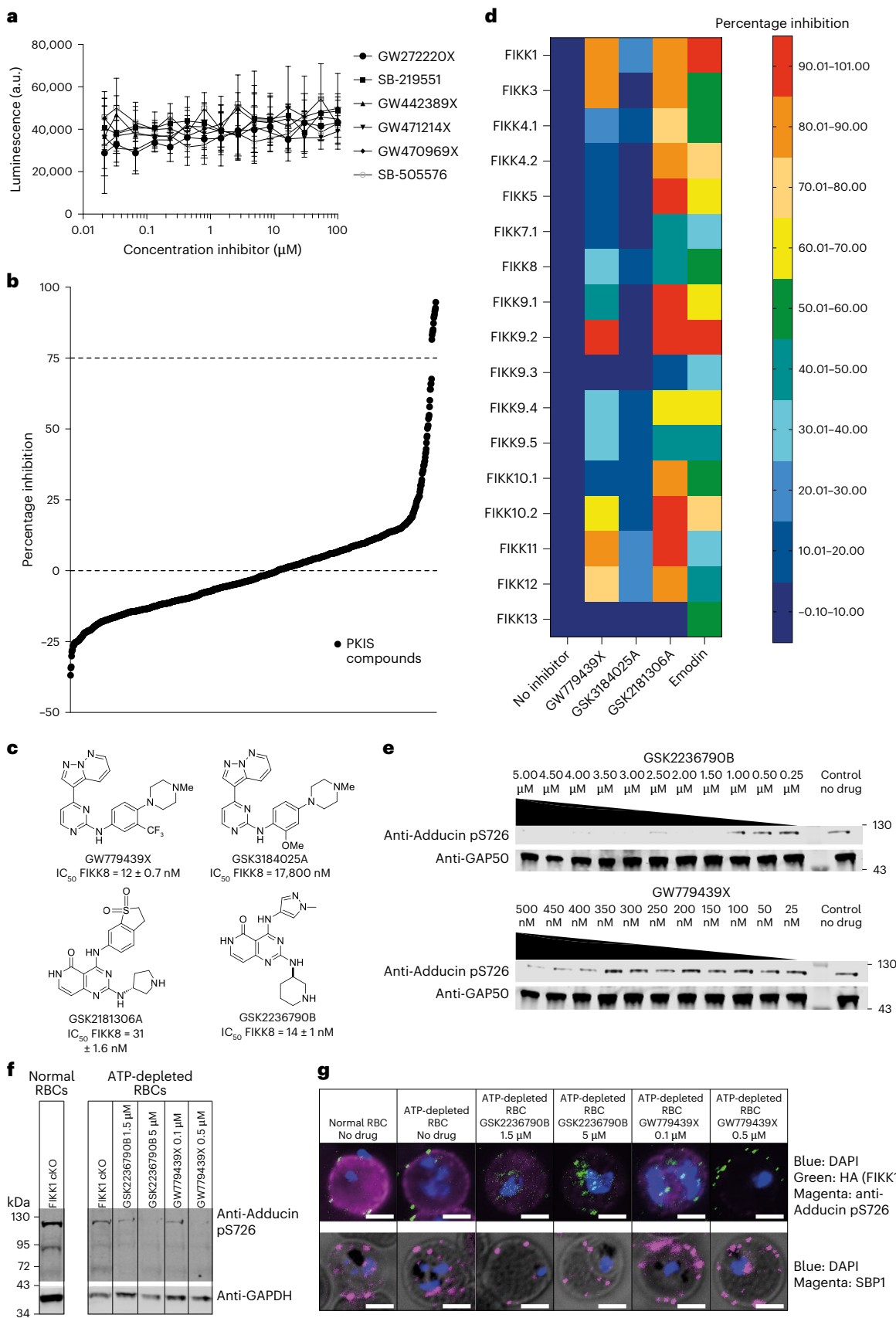

FIKK9.3 and FIKK13 were not inhibited by any compounds. The S/T/Y kinase inhibitor Emodin, known to inhibit *P. vivax* FIKK[24] and *P. falciparum* FIKK8 (ref. 25) showed potency against all *Pf*FIKKs (Fig. 6d). These results support the feasibility of pan-FIKK inhibition and provide further support that FIKK13 is indeed a tyrosine kinase.

## Pan-FIKK inhibitors inhibit FIKK1 in cell culture

In live parasite cultures, GW779439X, GSK2177277A and GSK2181306A inhibited parasite growth (half maximal effective concentration (EC$_{50}$) values of 0.31 ± 0.01 μM, 0.40 ± 0.01 μM and 0.16 ± 0.01 μM respectively) (Supplementary Fig. 9a), but not phosphorylation of serine 726 on human adducin (Supplementary Fig. 10), which depends on FIKK1 (ref. 16). As none of the exported *Pf*FIKKs were previously found to be individually essential for parasite growth[16], we hypothesized that the compounds engage one or multiple kinases other than the FIKKs at the concentrations used, preventing us from demonstrating their activity on the FIKKs in cell culture. Indeed, *P. knowlesi*, which only expresses one non-exported FIKK not expected to be essential, is equally susceptible to all the compounds (Supplementary Fig. 9b).

High ATP concentrations in the RBC (2–5 mM)[61] may compete with ATP analogues and prevent FIKK inhibition in cell culture. As we could not increase drug concentrations to test on-target activity without killing the parasites, we reduced the ATP concentration in the RBC[62] by pretreating RBCs with iodoacetamide and inosine. This resulted in substantial ATP depletion without preventing adducin S726 phosphorylation or parasite development (Supplementary Fig. 11). Under these conditions, adducin S726 phosphorylation, but not protein export (SBP1 (ref. 63)), was inhibited by the compounds (Fig. 6e–g), suggesting that they impair FIKK1 and potentially other FIKKs activity. Taken together, these data show that the compounds identified as pan-recombinant FIKK inhibitors are active on at least one of the parasite FIKKs in live cell culture. This sets a framework for screening better inhibitors working at physiological levels of ATP in RBCs.

## Discussion

*Laverania Plasmodium* species evolved ~1 million years ago from bird-infecting *Plasmodiae*[12]. *P. falciparum* emerged ~50,000 years ago, as a human parasite[14], with a severe population bottleneck in the last 5,000–10,000 years[12]. Several gene families important for host–pathogen interaction evolved specifically in the *Laverania* but their function remains largely elusive. Pseudogenization within these families in different *Laverania* species suggests that some genes may be remnants of their evolutionary past or indicate a level of redundancy that relaxes selection on current gene copies. Here, we provide strong evidence that most FIKKs in *P. falciparum* have diversified in function, are probably essential in human infections and appear under stringent selection within the *Laverania* clade. However, a few kinases are found as pseudogenes in patient isolates, indicating these may be remnants from an ancestor not required for infection of modern humans. We observe notable differences in pseudogenization between geographical backgrounds, suggesting plasticity in FIKK importance. This is interesting considering a recent study which found an association between an SNP in the *fikk4.2* gene and sickle-cell trait, which protects from severe *P. falciparum* malaria and is highly prevalent in people of African descent[64].

The expansion and functional diversification of the FIKK family required specialization of each kinase. This was achieved by different expression timing, subcellular localization and, as we show here, the evolution of highly specific phosphorylation motifs. By combining molecular docking and mutational analyses, we identified strong specificity determinants for FIKK12, and residues with more moderate effects for FIKK1 and FIKK9.1. Although we could not model specificity for all FIKKs, most of these map to rapidly evolving loop regions on the kinase domain, perhaps explaining why the FIKK family was able to functionally diversify rapidly (~1 million years) in terms of its phosphorylation motif specificity.

Strikingly, we show that FIKK13 is a bona fide tyrosine kinase in *P. falciparum* and *P. gaboni*, and therefore probably in other *Laverania* species. Additionally, several FIKKs show dual specificity. This suggests that tyrosine phosphorylation of host and/or exported parasite proteins by *Laverania* secreted kinases is not exclusively mediated by hijacking human kinases, as believed so far[65,66]. Thus, some exported FIKKs have probably evolved to specifically interfere with critical host signalling pathways relying on tyrosine phosphorylation. This could be important for the infection of nucleated cells such as erythroid precursors[67] and/or liver cells[68]. While a secreted dual specificity kinases (S/T and Y) has been described in the related *Toxoplasma* parasite[69], the evolution of apparently exclusive tyrosine specificity in FIKK13 from a S/T kinase family has not previously been observed. This is an important finding as it implies that in other species, bona fide tyrosine kinases might have evolved from a recent S/T kinase ancestor, as seen here for the FIKK family. However, predicting tyrosine kinase activity solely based on sequence or structure remains elusive. The determinants of tyrosine specificity of the FIKKs appear to be different than for canonical kinases[49,70] and it remains challenging to combine computational and experimental approaches to understand the precise molecular relationship between kinase sequence and specificity for all FIKKs and all substrate positions. The co-crystal structure of FIKK13 with a substrate may be required to understand tyrosine specificity. Two recent studies highlighted the importance of pre-phosphorylated residues in human kinases' preferred phosphorylation motifs[46,71]. Though the OPAL libraries used in this study lack pre-phosphorylated residues, future research could investigate possible hierarchical phosphorylation by FIKKs that co-localize.

While the diversification of the substrate specificity is underpinned by evolution of the peptide binding area, several conserved features of the FIKKs required for ATP-binding may allow the generation of inhibitors targeting several or all FIKKs simultaneously. The strictly conserved Phe in the F-I-K-K motif appears to participate in the unusual coordination of ATP in the kinase active site. In combination with a small gatekeeper residue common across the FIKKs, this feature may allow the design of compounds specifically inhibiting the FIKKs. Here, we identify in vitro pan-specific FIKK inhibitors that can interfere with FIKK1 activity in live parasites upon reducing ATP levels in the RBC, demonstrating that pan-FIKK inhibition is an achievable goal if inhibitors more specific over human enzymes can be found. Since FIKKs are probably critical for parasite survival in the host, their collective inhibition represents an interesting strategy for combination therapies. Resistance via single-gene mutations is readily observed against most current drugs[72] but unlikely for compounds that inhibit a whole family of proteins. The crystal structure solved in this work will allow further investigation of the FIKK family chemical inhibition.

## Methods

### FIKK orthologues in *Laverania*

To assess the number of FIKK orthologues in each *Laverania* species, the word 'FIKK' was entered into the search engine of the PlasmoDB website (www.PlasmoDB.org) (release 66) selecting *Plasmodium adleri* G01, *Plasmodium billcollinsi* G01, *Plasmodium blacklocki* G01, *Plasmodium falciparum* 3D7, *Plasmodium gaboni* G01, *Plasmodium praefalciparum* G01 and *Plasmodium reichenowi* CDC genomes. To assess syntenicity, the JBrowse genome browser of PlasmoDB was used, selecting the 'Syntenic Sequences and Genes (Shaded by Orthology)' track. The *P. falciparum* 3D7 genome was used as a reference to evaluate whether the chromosome sequences from the other *Laverania* species were complete. 'No genome information' signifies that chromosome sequence was not available in the database, probably due to degradation of telomeric regions.

## Field genomes FIKK pseudogenization analysis

To identify genetic variants in *fikk* genes, a global dataset of clinical *P. falciparum* samples was examined with bcftools, using the Pf3K project release 5 (ref. 29). Out of the 2,483 *P. falciparum* clinical samples of diverse geographical origin, 2,085 with high-quality data were selected (>80% of the genome covered with ten or more reads). For each isolate genome, *fikk* pseudogenes were defined by the presence of at least one internal STOP codon variant with an alternative allele frequency greater than 0.5 (Alt reads divided by total number of reads of that position). To identify natural genomic deletions that include *fikk* genes, deleted genes were defined as 95% of the gene sequence with coverage under three reads. The large majority of deleted FIKK kinases had zero reads over the entire length of the gene, with the rest of the genome being over 10× coverage (typically ~50×). As a complementary approach, we made use of the microarray transcriptomic data from Mok et al.[73]. From the 2,085 genome samples, 693 transcriptomes from the same isolates were also available. A negative $\log_2$ value was defined as 'no expression'.

## FIKK percentage of expression heat map

Expression data were taken from Hoeijmakers et al.[74] RNA-sequencing dataset available on PlasmoDB release 66 (www.PlasmoDB.org). The dataset gives a transcript per kilobase millions (TPM) value for eight different time windows throughout *P. falciparum* 48 h asexual replication cycle ((40–5 hours post-infection (hpi)), (2–10 hpi), (7–15 hpi), (12–20 hpi), (17–25 hpi), (22–30 hpi), (27–35 hpi) and (32–40 hpi)). Percentage of expression was calculated for each *P. falciparum* FIKK kinase using the following formula $\frac{\text{(TPM for a time point)}}{\text{(Highest TPM value)}} \times 100$. Percentage of expression values were then plotted in GraphPad Prism10 and represented as a heat map with dark blue cells representing no expression and yellow cells representing 100% expression.

## Human cells

Human RBCs were acquired from the National Health Service Blood and Transplant service.

## In vitro maintenance and synchronization of *Plasmodium* parasites

Human erythrocytes infected with *P. falciparum* asexual stages were cultured at 37 °C in complete medium. Complete medium consists of 1 l RPMI-1640 medium supplemented with 5 g Albumax II (Thermo Fisher Scientific) to act as a serum substitute, 0.292 g L-glutamine, 0.05 g hypoxanthine, 2.3 g sodium bicarbonate, 0.025 g gentamycin, 5.957 g HEPES and 4 g dextrose. A haematocrit of 1–5% was used and the blood was from anonymous donors provided through the UK Blood and Transfusion service. According to standard procedures, parasites were grown in a gas atmosphere consisting of 90% $N_2$, 5% $CO_2$ and 5% $O_2$ (ref. 75). Thin blood smears fixed in 100% methanol, air dried and stained with Giemsa were routinely used to assess parasitaemia and developmental stages by light microscopy. *P. knowlesi* parasites in the asexual RBC stages were cultured in complete medium supplemented with 10% human serum as described previously[76]. Parasite cultures were synchronized by Percoll (GE Healthcare) for isolation of mature schizont stages parasites. Purified schizonts were incubated in complete medium at 37 °C with fresh RBCs for 4 h in a shaking incubator. Any remaining schizonts were removed with a second Percoll purification leaving only tightly synchronized ring-stage parasites in the flask.

## Gametocyte induction, culture, FIKK gene excision and collection

An adapted version of previously described techniques was used to obtain synchronous gametocytes[77]. Briefly, highly synchronous ring-stage parasites at 8–10% parasitaemia were stressed by retaining half the spent culture medium and replenishing the rest with fresh complete medium. The following day, the stressed cultures were spun and the spent culture medium was replaced with complete media. Cultures were left shaking until the following day when all the schizonts had ruptured and reinvaded. A certain proportion of the reinvaded rings should have then committed to gametocytogenesis. These committed parasites were then split into two flasks and treated for 4 h at 37 °C with either 100 nM rapamycin (Sigma) or dimethyl sulfoxide (DMSO) (0.1% (v/v)) as described previously[78]. Parasites were then washed three times with complete medium and cultured in complete medium supplemented with 10% human serum. From this point onwards, parasite culture medium was exchanged daily with prewarmed complete medium supplemented with 10% human serum and heparin at 20 units per ml to prevent asexual growth. When a majority of stage III gametocytes could be seen on Giemsa smears, the cultures were submitted to Percoll purification allowing isolation of sexual stages which were lysed in 5× SDS sample buffer for western blot analysis of FIKK kinases expression.

## Immunoblotting

Parasites submitted to western blot analysis were first collected by Percoll purification. Then, 1 µl of parasite pellets were resuspended in 15 µl PBS, lysed with 5× SDS sample buffer (25 mM Tris–HCl pH 6.8, 10% SDS, 30% glycerol, 5% β-mercaptoethanol and 0.02% bromophenol blue) and denatured at 95 °C for 5 min. Samples were then subjected to SDS–PAGE, transferred onto a Transblot Turbo mini-size nitrocellulose membrane (Bio-Rad) and blocked overnight in 5% skimmed milk in PBS with 0.2% Tween 20 at 4 °C. For FIKK kinases expression in gametocytes (Fig. 1c), the membranes were probed with rat anti-HA high affinity (clone 3F10, 11867423001, Roche, 1:1,000) and rabbit anti-GAP50 (ref. 79) (a gift from Julian Rayner, 1:2,000) antibodies. For western blots investigating Adducin S726 phosphorylation (Fig. 6 and Supplementary Figs. 10 and 11), the membranes were probed with rabbit anti-Adducin pS726 (Abcam, ab53093, 1:1,500), rabbit anti-MAHRP1 (a gift from J. Rayner and L. Parish, 1:2,000), rabbit anti-GAP50 (1:2,000) or mouse anti-GAPDH (1:10,000) (monoclonal antibody 7.2 (anti-GAPDH), which was obtained from The European Malaria Reagent Depository (http://www.malariaresearch.eu), source Dr. Jana McBride[80]). For western blots assessing protein biotinylation by FIKK4.1 and FIKK4.2::TurboID (Supplementary Fig. 1b), the membranes were probed with rabbit anti-MAHRP1 (1:2,000) and mouse anti-V5 (Abcam, ab27671, 1:1,000). Following primary antibody staining, the membranes were incubated with the relevant secondary fluorochrome-conjugated antibodies (LI-COR, 1:20,000) or IRDye 800CW Streptavidin (926-32230, 1:2,000). The antibody reactions were carried out in 5% skimmed milk in PBS with 0.2% Tween 20 for 1 h in the dark and membranes were washed three times between each antibody staining in PBS with 0.2% Tween 20. After a final wash with PBS, the antigen–antibody reactions were visualised using the Odyssey infrared imaging system (LI-COR Biosciences).

## Transcription evidence of *Pf*FIKK kinases in sexual and mosquito stages

Data were obtained from the Malaria Cell Atlas (www.malariacellatlas.org)[31]. The SmartSeq2 cell view was used and an FIKK kinase was considered expressed if at least two sample analysed showed an expression above 0.

## Generation of FIKK::TurboID parasite lines

FIKK::TurboID parasite lines were generated using CRISPR–Cas9. Briefly, suitable gRNAs for FIKK4.1 and FIKK4.2 were identified using the Eukaryotic Pathogen CRISPR guide RNA/DNA Design Tool[81]. A pair of complementary oligonucleotides corresponding to the 19 nucleotides closest to the identified PAM sequence was synthesized (Integrated DNA Technologies (IDT)), phosphorylated using T4 polynucleotide kinase, annealed and ligated into pDC_Cas9_hDHFRyFCU[82] digested with BbsI. To generate compatible, sticky ends between the annealed primer pairs encoding the gRNAs and the BbsI digested vector, the forward oligonucleotide had 5′-ATTG added to the 19 nucleotides

corresponding to the gRNAs, whereas the compatible oligonucleotide had a 5′-AAAC overhang added (Supplementary Table 13). This way, gRNAs targeting *fikk4.1* and *fikk4.2* genes were assembled using oligonucleotide pairs gRNA_4.1_310For/Rev and gRNA_4.2_235For/Rev, respectively. Repair templates containing a 5′HR, a recodonized sequence, a linker, a TurboID-coding sequence, a V5-tag and a 3′HR flanked by two XhoI restriction sites were ordered from GeneArt (Supplementary Table 13). For transfection, 60 μg repair template plasmid was linearized with XhoI for 4 h at 37 °C before inactivation at 80 °C for 20 min. Next, 20 μg of gRNA plasmid was added and the plasmid mixture was ethanol precipitated, washed and resuspended in 10 μl sterile TE buffer (10 mM Tris and 1 mM EDTA). In parallel, highly synchronized segmented schizonts (48 hpi) of NF54::DiCre parasites[83] were collected by Percoll enrichment and washed once with complete medium. The DNA constructs in TE buffer were mixed with 90 μl P3 primary cell solution (Lonza) and used to resuspend 20 μl segmented schizonts, which were subsequently transferred to a transfection cuvette. Transfections were performed by electroporation using the FP158 program from an Amaxa 4D Electroporator machine (Lonza). Following transfection, the parasites were transferred to prewarmed flasks containing 2 ml complete medium and 300 μl fresh uninfected RBCs (uRBCs). After 40 min of gentle shaking at 37 °C, 8 ml complete medium were added to the flask. Transfected parasites were incubated for 24 h, then selection was performed with 2.5 nM WR99210 (Jacobus Pharmaceuticals) for 4 days. Following establishment of the transgenic lines, correct modification of the parasite genome was confirmed by PCR using the primers described in Supplementary Fig. 1a and Supplementary Table 13.

#### Phylogenetic tree

All FIKK amino acid sequences were retrieved from the UniProtKB[84]. Heavily truncated sequences (<200 amino acids) were removed manually. The full-length protein sequences were then aligned using the MAFFT L-INS-i algorithm[85]. Alignment positions where more than 20% of sequences contain a gap (-gt 0.8) were removed from the multiple sequence alignment using trimAl software[86].

A maximum-likelihood estimate of the FIKK phylogeny was generated with IQ-TREE2 software[87], using the ModelFinder parameter (-m MFP) to automatically detect the best evolutionary model[88]. Branch support for the maximum-likelihood phylogeny was assessed using 100 replicates of the Felsenstein bootstrap (-b 100)[89]. The phylogenetic tree was visualized using the ggtree package in R[90] after removing homologues to the FIKK7.2 and FIKK14 pseudogenes.

The dN/dS analysis was performed in CodeML using a published protocol[91,92]. Codon alignments were generated using TranslatorX[93]. The phylogenetic tree was generated as described above. A global dN/dS ratio (M0 model) was first estimated across all branches and sites. A branch model was then performed to compare dN/dS ratios between exported and non-exported kinases.

#### Immunofluorescence assays

Air-dried blood films were fixed for 5 min in ice-cold methanol and subsequently rehydrated in PBS for 5 min. Slides were blocked in 3% (w/v) bovine serum albumin (BSA) in PBS containing kanamycin (50 μg ml⁻¹) for 1 h and subsequently incubated with primary antibodies in 1% (w/v) BSA in PBS containing kanamycin (50 μg ml⁻¹) for 1 h at room temperature. Primary antibodies dilutions were as follow: high affinity rat anti-HA (clone 3F10, 11867423001, Roche; 1:1,000), mouse anti-FIKK4.2 (1:1,000) (monoclonal antibody 126 (anti-FIKK4.2) obtained from The European Malaria Reagent Depository (http://www.malariaresearch.eu) source, Dr. Odile Mercereau-Puijalon[17]), rabbit anti-phosphoAdducin S726 (1:1,500) (ab53093, Abcam), rabbit anti-SBP1 (1:10,000) (gift from T. Spielmann[94]). After three washes with PBS, the coverslips were incubated with the relevant Alexa Fluor secondary antibodies (1:2,000 in PBS with 1% BSA) at room temperature for 1 h in the dark. After three final washes with PBS, the slides were

mounted with Prolong Gold antifade reagent (Invitrogen) containing the DNA dye 4,6-diamidino-2-phenylindole (DAPI), covered with a coverslip and sealed with nail polish. Images were taken using a Ti-E Nikon microscope using a 100× objective at room temperature equipped with a light-emitting diode illumination and Orca-Flash4 camera. The images were processed using Nikon Elements software (Nikon).

#### Proximity labelling experiments

We first performed a NF54-FIKK4.2::TurboID comparison at the peptide level. We then repeated the assay with NF54, FIKK4.1::TurboID and FIKK4.2::TurboID. Both experiments were performed following the same protocol and data from both experiments were combined.

**Cell culture and lysis.** For all experiments, NF54 wild-type (WT) parasites used as controls, FIKK4.1::TurboID and FIKK4.2::TurboID parasites were tightly synchronized to a 4 h window using Percoll. For each line, parasites were grown in biological triplicate in 200 ml of complete medium containing biotin (0.2 mg l⁻ ≈819 nM) at at least 10% parasitaemia in 2 ml of blood, each replicate being cultured in blood coming from different donors. iRBCs were collected at late schizont stage (44–48 hpi) using Percoll. Subsequently, parasites were washed three times with 50 ml of complete medium and five times with 5 ml PBS. Parasites were then lysed in 8 M urea in 50 mM HEPES pH 8.0 containing protease inhibitors (cOmplete, Roche). Samples were further solubilized by sonication with a microtip sonicator on ice for three rounds of 30 s at an amplitude of 30%. Lysates were then clarified by centrifugation at 21,130g for 30 min at 4 °C. The protein concentrations were then calculated using a BCA protein assay kit (Pierce), first diluting 20 μl aliquots from all lysates 1:25 in H₂O to reduce the concentration of urea and then following the instructions provided in the kit.

**Protein digestion.** Then, 4 mg of each lysate was then reduced with 5 mM dithiothreitol (DTT) for 1 h at room temperature and subsequently alkylated in the dark with 10 mM iodoacetamide for 30 min at room temperature. Following alkylation, the lysates were diluted with 50 mM HEPES pH 8.0 to <2 M urea and digested overnight with trypsin (Promega) at 1:50 (enzyme:protein) at 37 °C.

**Sep-Pak desalting.** Samples were cooled on ice for 10 min before being acidified with trifluoroacetic acid (TFA; Thermo Fisher Scientific) to a final concentration of 0.4% (v/v) and left on ice for a further 10 min. All insoluble material was removed by centrifugation (21,130g, 10 min, 4 °C) and the supernatants were desalted on Sep-Pak C18 1cc Vac cartridges (Waters) in conjunction with a vacuum manifold. The columns were first washed with 3 ml acetonitrile, conditioned with 1 ml of 50% acetonitrile and 0.5% acetic acid in H₂O, and then equilibrated with 3 ml of 0.1% TFA in H₂O. The acidified samples were loaded, desalted with 3 ml of 0.1% TFA in H₂O, washed with 1 ml of 0.5% acetic acid in H₂O and finally eluted in 1.3 ml of 50% acetonitrile and 0.5% acetic acid in H₂O. Each sample was then dried by vacuum centrifugation.

**Charging protein G agarose beads with anti-biotin antibodies.** A total of 60 μl of protein G agarose bead slurry (Thermo Fisher Scientific) were taken per sample. Beads were washed three times with 10 bead volumes of Biosite buffer[95] (50 mM Tris–HCl pH 8.0, 150 mM NaCl, 0.5% Triton x100, pH 7.2–7.5) at 4 °C. According to the supplier's recommendations, protein G agarose beads were functionalized with 100 μg antibodies per 100 μl slurry with two different anti-biotin antibodies (150–109 A, Bethyl Laboratories; ab53494, Abcam) by adding 300 μg of each antibody to the beads which were incubated rotating overnight at 4 °C.

**Immunoprecipitation.** Samples were dissolved in 1.5 ml biosite buffer on ice and the pH adjusted with 1–5 μl 10 M NaOH to 7–7.5 at 4 °C. Any undissolved material was removed by spinning at 21,130g, for 10 min

at 4 °C and the peptide BCA assay (Pierce) was performed on the supernatant to determine the peptide concentration in each sample. Protein G agarose beads functionalized with anti-biotin antibodies were washed three times with 10 bead volumes (3 ml). Biosite buffer and equal amount of peptides per sample was added onto the antibody loaded beads (60 μl slurry per sample). The mixture was incubated rotating for 2 h at 4 °C. Beads were pelleted at 1,500$g$ for 2 min at 4 °C and washed three times with 500 μl biosite buffer, once with 500 μl of 50 mM Tris–HCl pH 8.0 and three times with 500 μl H$_2$O. Peptides were eluted from the beads by adding 50 μl of 0.2% TFA, gently shaken and spun at 1,500$g$ for 2 min at 4 °C. Elution was repeated four times for a total volume of 200 μl.

**Stage-tip desalting.** All samples were desalted before liquid chromatography–tandem mass spectrometry (LC–MS/MS) using Empore C18 discs (3 M). Briefly, each stage tip was packed with one C18 disc, conditioned with 100 μl of 100% methanol, followed by 200 μl of 1% TFA. The samples were loaded onto the stage tip in 200 μl of 0.2% TFA, washed twice with 300 μl of 1% TFA and eluted with 40 μl of 40% acetonitrile + 0.1% TFA. The desalted peptides were vacuum dried in preparation for LC–MS/MS analysis.

**LC–MS/MS.** Samples were loaded onto Evotips according to the manufacturer's instructions. After a wash with 0.1% formic acid in H$_2$O, samples were loaded onto an Evosep One system coupled to an Orbitrap Fusion Lumos (Thermo Fisher Scientific). A PepSep 15 cm column was fitted onto the Evosep One and a predefined gradient for a 44 min method was used. The Orbitrap Fusion Lumos was operated in data-dependent mode with a 1 s cycle time, acquiring IT HCD (higher-energy collisional dissociation) MS/MS scans in rapid mode after an OT MS1 survey scan ($R = 60,000$). The target used for MS1 was 4E5 ions whereas the MS2 target was 1E4 ions. The maximum ion injection time utilized for MS2 scans was 300 ms, the HCD normalized collision energy was set at 32 and the dynamic exclusion was set at 15 s.

**Data processing.** Acquired raw files were processed with MaxQuant v2.0.3.1 (ref. [96]). The Andromeda[97] search engine was used to identify peptides from the MS/MS spectra against *Plasmodium falciparum* (PlasmoDB_v46)[13] and *Homo sapiens* (UniProt, 2020)[84]. Acetyl (Protein N-term), Biotin (K), Oxidation (M) were selected as variable modifications whereas Carbamidomethyl (C) was selected as a fixed modification. The enzyme specificity was set to Trypsin with a maximum of three missed cleavages. The minimum peptide length was set to six amino acids. Biotinylated peptides search in MaxQuant was enabled by defining a biotin adduct (+226.0776) on lysine residues as well as three diagnostic ions: fragmented biotin ($m/z$ 227.0849), immonium ion harbouring biotin with a loss of NH$_3$ ($m/z$ 310.1584) and an immonium ion harbouring biotin ($m/z$ 327.1849).

The precursor mass tolerance was set to 20 ppm for the first search (used for mass recalibration) and to 4.5 ppm for the main search. The datasets were filtered on posterior error probability to achieve a 1% false discovery rate on protein, peptide and site level. Other parameters were used as pre-set in the software. 'Unique and razor peptides' mode was selected to allow identification and quantification of protein in groups (razor peptides are uniquely assigned to protein groups and not to individual proteins). Intensity-based absolute quantification in MaxQuant was performed using a built-in quantification algorithm[96] enabling the 'Match between runs' option (time window 0.7 min) within replicates.

**Data analysis.** The MaxQuant output files were processed with Perseus v1.5.0.9 (ref. [98]). Modified peptides data were filtered to remove contaminants and IDs originating from reverse decoy sequences. Intensity-based absolute quantification intensities were log$_2$ transformed and peptides with less than one valid value in total were removed. Non-biotinylated peptides (background) were also removed

from the datasets. Additionally, peptides with intensities only in the NF54 samples were removed as they are likely to represent background binding to the beads. Replicates were grouped for each condition (NF54 and FIKK4.2::TurboID for the first experiment and NF54, FIKK4.1::TurboID and FIKK4.2::TurboID for the second experiment) and only peptides with at least two valid values in at least one group were conserved for further analysis. Data for the first experiment (NF54 – FIKK4.2::TurboID) and the second experiment (NF54 – FIKK4.1::TurboID – FIKK4.2::TurboID) are available in Supplementary Table 14.

### Network
The network representation of the TurboID data (Fig. 2c) was generated using Cytoscape v3.10.1 (ref. [99]). Proximal proteins were included in the network if they contained at least one peptide that was biotinylated in two or more of the three biological replicates from either the FIKK4.1 or FIKK4.2 TurboID assays. All proteins in the vicinity of FIKK4.1 or FIKK4.2 were annotated as potential kinase targets if they were found to be less phosphorylated upon knock out of the respective kinase, using data published in ref. [16]. Regulated phosphosites on candidate substrates were scored against the FIKK4.1 or FIKK4.2 kinase specificity models shown in Supplementary Fig. 3, using a simple scoring function that outputs a normalized summation between 0 (minimum) and 1 (maximum)[100]. Data on protein proximity, target status and motif scores are given in Supplementary Table 7.

### Recombinant protein expression and purification
The DNA sequences coding for *P. falciparum* 3D7 and *P. gaboni* G01 FIKK kinase domains were obtained from PlasmoDB (https://plasmodb.org/plasmo/)[13] and were codon optimized for *E. coli* expression (IDT)[101] (see Supplementary Table 15 for recodonized FIKK kinase sequences). For FIKK4.2, blocks of low-complexity repeat sequences and the short low-complexity downstream sequence (amino acids 403–928) were removed as per ref. [17]. Sequences were subsequently inserted into a pET-28a vector (Novagen) to produce a N-terminal thrombin cleavage His$_6$ tag fusion (MGSS**HHHHHH**SSG_LVPRGSH_*MASMTGGQQMG*RGS, where the sequence in bold is the His$_6$ tag, the underlined sequence is the thrombin site and the sequence in italics is the T7 tag). The insert sequence was verified by DNA sequencing. For expression in *E. coli*, BL21-Gold (DE3) cells (Stratagene) were transformed with pET-28a-FIKK vectors, grown over 2 days at 18 °C in ZYM-5052 media supplemented with 50 μg ml$^{-1}$ kanamycin and collected by centrifugation. In a typical preparation, 10 g of cells were resuspended in 100 ml lysis buffer (50 mM Tris–HCl pH 7.5, 500 mM NaCl, 1 mM TCEP, 20 mM imidazole, 10 mM MgSO$_4$, 10% glycerol and two protease inhibitor cocktail tablets (cOmplete, EDTA free, Roche)), lysed by sonication and clarified by centrifugation at 20,000$g$ for 30 min at 4 °C. The supernatant was loaded into a 1 ml HisTrap column (GE Healthcare) and the bound proteins were eluted in 50 mM Tris–HCl pH 7.5, 500 mM NaCl, 1 mM TCEP, 300 mM imidazole and 10% glycerol. After concentration, the samples were loaded on a Hi-Load Superdex 200 16/600 column (GE Healthcare) equilibrated with 50 mM Tris–HCl pH 7.5, 250 mM NaCl, 1 mM TCEP and 10% glycerol. The fractions containing the different recombinant FIKK kinase domains were analysed by SDS–PAGE stained by Coomassie (Supplementary Fig. 2).

### Peptide arrays
OPAL and phosphoproteome peptide libraries synthesis was performed by the Francis Crick Institute Peptide Chemistry Science Technology platform as described previously[37,102]. Briefly, peptide arrays were synthesized on an Intavis ResResSL automated peptide synthesiser (Intavis Bioanalytical Instruments) by cycles of N(a)-Fmoc amino acids coupling via activation of the carboxylic acid groups with diisopropyl-carbodiimide in the presence of ethylciano-(hydroxyamino)-acetate (Oxyma pure) followed by removal of the temporary α-amino protecting group by piperidine treatment. Subsequent to chain assembly, side

chain protection groups are removed by treatment of membranes with a deprotection cocktail (20 ml 95% trifluoroacetic acid, 3% triisopropylsilane and 2% $H_2O$) for 4 h at room temperature, then washing (4× dichloromethane, 4× ethanol, 2× $H_2O$ and 1× ethanol) before being air dried. For the phosphoproteome peptide libraries, the final product is a cellulose membrane containing a library of 11-mer peptides. Sequences of the peptides can be found in Supplementary Table 9. For the OPAL libraries, the final product is a cellulose membrane containing a library of 9-mer peptides with the general sequences: A-X-X-X-X-S-X-X-X-X-A, A-X-X-X-X-T-X-X-X-X-A or A-X-X-X-X-Y-X-X-X-X-A. For each peptide, 1 of the 20 naturally occurring proteogenic amino acids was fixed at each of the 8 positions surrounding the phosphorylated residue (S, T or Y), with the remaining positions, represented by X, degenerate (approximately equimolar amount of the 16 amino acids excluding cysteine, serine, threonine and tyrosine). Cellulose membranes were placed in an incubation trough and moisten with 5 ml ethanol. They were subsequently washed twice with 50 ml kinase buffer (20 mM MOPS, 10 mM magnesium chloride and 10 mM manganese chloride, pH 7.4; Alfa Aesar) and incubated overnight in 100 ml reaction buffer (kinase buffer + 0.2 mg ml$^{-1}$ BSA (BSA Fraction V, Sigma) + 50 µg ml$^{-1}$ kanamycin). The next day, the kinase buffer was removed and the membranes were incubated at 30 °C for 1 h in 30 ml blocking buffer (kinase buffer + 1 mg ml$^{-1}$ BSA + 50 µg ml$^{-1}$ kanamycin). After incubation, the blocking buffer was replaced with 30 ml reaction buffer supplemented with 300 µl 10 mM ATP and 125 µCi [γ-32P]-ATP (Hartmann Analytics). The reaction was started by adding 100 nM of the recombinant FIKK kinase domain studied and left to incubate for 20 min at 30 °C with gentle agitation. After incubation, the reaction buffer was removed and the membranes were washed 10× 15 min with 100 ml 1 M NaCl, 3× 5 min with 100 ml $H_2O$, 3× 15 min with 5% $H_3PO_4$, 3× 5 min with 100 ml $H_2O$ and 2× 2 min with 100 ml ethanol. The membranes were left to air dry before being wrapped up in plastic film and exposed overnight to a PhosphorScreen. The radioactivity incorporated into each peptide was then determined using a Typhoon FLA 9500 phosphorimager (GE Healthcare) and quantified with the program ImageQuant (version 8.2, Cytiva LifeScience). Data corresponding to the 'signal above background' was used.

## PWM generation from OPAL data

PWMs were constructed from the raw OPAL data using a standard approach presented in refs. 103,104. First, raw OPAL values for S, T and Y amino acids were replaced with average (median) values for each corresponding peptide position to control for the possibility of spurious phosphorylation in flanking region. The OPAL values were then normalized per position to give a mean PWM score of 1 per amino acid and a total score of 20 per position. The raw OPAL data from S, T and Y libraries were then combined to generate a S/T/Y PWM. This was achieved by summing OPAL scores (after correcting flanking S/T/Y scores) from each of the peptide libraries. The OPAL data were then normalized as before to yield a mean PWM score of 1 per amino acid and a total PWM score of 20 per position. The relative scores between S, T and Y at position were calculated by taking the ratio of the total OPAL scores for the S, T and Y libraries. For ease of visualization, the PWM logos display only amino acids with the scores above the arbitrary threshold of 2.5 using the software package ggseqlogo[105]. These PWM scores were then log$_2$ transformed to generate heat maps of the matrix specificity scores.

## FIKK13 peptide RaPID selection

In vitro selections were carried out with Bio-His-FIKK13 following previously described protocols. Briefly, initial DNA libraries (including 6–12 degenerate NNK codons) were transcribed to mRNA using T7 RNA polymerase (37 °C, 16 h) (Thermo Scientific) and ligated to a puromycin linker primer ([5′Phos]CTCCCGCCCCCGT CC[SP18][SP18][SP18][SP18][SP18]CC[Puromycin]) using T4 RNA ligase

(30 min, 25 °C) (New England Biolabs). First round translation was performed on a 150 µl scale, with subsequent rounds performed on a 5 µl scale. Translations were carried out (30 min, 37 °C then 12 min, 25 °C) using a custom methionine(−) Flexible In vitro Translation system composed by PURExpress (ΔRF123) kit (New England Biolabs) solution B, an in-house solution A (50 mM HEPES–KOH pH 7.6, 2 mM ATP, 2 mM GTP, 1 mM CTP, 1 mM UTP, 20 mM creatine phosphate, 100 mM potassium acetate, 2 mM spermidine, 6 mM magnesium acetate, 1.5 mg ml$^{-1}$ E. coli tRNA mix (Roche) and 14 mM DTT) and additional ClAc-D-Tyr-tRNA$^{fMet}_{CAU}$ (25 µM). Ribosomes were then dissociated by addition of EDTA (18 mM final concentration, pH 8) and the library mRNA reverse transcribed using MMLV RTase, Rnase H Minus. The reaction mixture was buffer exchanged into selection buffer (50 mM Tris pH 7.5, 50 mM NaCl, 2 mM DTT, 10 mM MgCl$_2$, 1.5 µM ADP and 0.1% Tween) using 1 ml homemade columns containing pre-equilibrated Sephadex resin (Cytiva) before the addition of 2× blocking buffer (50 mM Tris pH 7.5, 250 mM NaCl, 2 mM DTT, 10 mM MgCl$_2$, 1.5 µM ADP, 0.1% Tween, 4 mg ml$^{-1}$ sheared salmon sperm DNA (Invitrogen) and 0.1% acetyl-BSA final (Invitrogen)). Libraries were incubated with negative selection beads (Dynabeads M280 streptavidin (Life Technologies)) (3× 30 min, 4 °C) followed by incubation with bead-immobilized His-bio-FIKK13 (200 nM, 4 °C, 30 min) before washing (3× 1 bead volume selection buffer, 4 °C) and elution of retained mRNA/DNA/peptide hybrids in PCR buffer (95 °C, 5 min). Library recovery was assessed by quantitative real-time PCR relative to a library standard, negative selection and the input DNA library. Recovered library DNA was used as the input library for the subsequent round. Following six rounds of selection, double-indexed libraries (Nextera XT indices) were prepared and sequenced on a MiSeq platform (Illumina) using a v3 chip as single 151 cycle reads. Sequences were ranked by total read numbers and converted into their corresponding peptides sequences for subsequent analysis (Supplementary Table 16).

The library DNA was 5′-TAATACGACTCACTATAGGGTT AACTTTAAGAAGGAGATATACATATG(NNK)$_n$TGCGGCAGCGGCAGCGG CAGCTAGGACGGGGGGGCGGAAA.

Bead preparation was carried out as follows: To assess the binding capacity of biotinylated FIKK13 to streptavidin, Bio-His-FIKK13 was incubated with different quantities of magnetic streptavidin beads (Invitrogen) for 30 min. Beads were then washed three times with cold selection buffer and protein elution was performed by boiling the beads at 95 °C for 5 min. Samples were then run in an SDS–PAGE gel and stained with Coomassie. Bead capacity was calculated quantifying the gel bands with FIJI.

## FIKK13 cyclic peptide synthesis

Peptides were synthesized using NovaPEG Rink Amide resin as C-terminal amides by standard Fmoc-based solid-phase synthesis as previously described, using a Liberty Blue Peptide Synthesis System (CEM), a SYRO I (Biotage) or a Activotec P-11 peptide synthesizer. Following synthesis, the N-terminal amine was chloroacetylated by reaction with 0.5 M chloromethylcarbonyloxysuccinimide (ClAc-NHS) in dimethylformamide (1 h, room temperature). The resin was washed (5× dimethylformamide and 5× dichloromethane) and dried in vacuo.

Peptides were cleaved from the resin and globally deprotected with TFA/triisopropylsilane/1,2-ethanedithiol/$H_2O$ (92.5:2.5:2.5:2.5) for 3 h at room temperature. Following filtration, the supernatant was concentrated by centrifugal evaporation and precipitated with cold diethyl ether. Crude peptides were resuspended in DMSO/$H_2O$ (95:5) and, following basification with triethylamine to pH 10, were incubated with rotation for 1 h at room temperature. Peptides were then acidified with TFA and purified by HPLC (Shimadzu) using a Merck Chromolith column (200 × 25 mm) with a 10–50% gradient of $H_2O$/acetonitrile containing 0.1% TFA. Pure peptides were lyophilized and dissolved in DMSO for further use. Peptide stock concentrations were determined by absorbance at 280 nm based on their predicted extinction coefficients.

## Surface plasmon resonance

Single-cycle kinetics analysis by SPR was carried out using Biacore S200 and a Biotin CAPture kit, series S (Cytiva). Bio-His-FIKK13 was immobilized on the chip to yield a response of approximately 1,400 RU. The running buffer contained 50 mM Tris pH 7.5, 250 mM NaCl, 2 mM DTT, 10 mM $MgCl_2$, 1.5 µM ADP, 0.02% Tween and 0.1% DMSO and experiments were performed at 25 °C. Samples were run with 100 s contact time and data were analysed using the Biacore S200 analysis software. Data represent the average ± s.d. of at least two independent replicates. SPR data are available in Supplementary Table 9.

## ADP-Glo assay

Recombinant FIKK kinase domains activity was measured using the ADP-Glo kinase assay (Promega), which quantifies the amount of ADP produced during the kinase reaction. Briefly, the kinase reactions were conducted at room temperature for 1 h by mixing 100 nM recombinant FIKK kinase domain with 10 µM ATP and 10 µM substrate when specified, in 40 µl kinase buffer (20 mM MOPS, 10 mM magnesium chloride and 10 mM manganese chloride, pH 7.4 (Alfa Aesar)). When kinase inhibition by ATP analogues was assessed, compounds (diluted in DMSO, final concentration ≤1%) were tested at 10 µM, or otherwise specified, by incubation for 15 min with the recombinant kinase domain before the addition of ATP ± substrate. ADP-Glo reagent (40 µl) was added to stop the kinase reaction and deplete the unconsumed ATP. After incubation at room temperature for another hour, 80 µl kinase detection reagent was added and incubated for 30 min at room temperature. Luminescence was measured using the multimode microplate reader FLUOstar Omega (BMG Labtech).

## Protein sequence identity matrix

As described above, *P. falciparum* FIKK amino acid sequences were retrieved from UniProt and aligned using the MAFFT L-INS-i algorithm[85].

Heavily gapped alignment positions (more than 20% gapped) were filtered out of the multiple sequence alignment using the trimAl software[86]. The sequence identity of this processed alignment was then calculated using seqidentity (normalise=TRUE) function in the R package bio3d[106].

## Phosphoproteome libraries analysis

PWMs were calculated for each FIKK kinase as described above. These data were then cross-referenced with the phosphoproteome peptides presented in Fig. 4. Each peptide in the array was scored for its match to the FIKK preferred phosphorylation motif, using a simple matrix similarity score (MSS) of the PWM against the peptide sequence[100]. This function outputs a normalized score that has a minimum of 0 and a maximum of 1. In each case, a Pearson's correlation coefficient is calculated between the motif score ($x$) and a $\log_{10}$ transformation of the phosphorylation signal from the phosphoproteome peptide array ($y$), from curves of the form $y = \log(x)$.

Peptides from the library were divided into a motif 'match' and 'no match' with respect to the FIKK specificity matrix. This was based on a null distribution of randomized peptide sequences for phosphosites not affected by the FIKK knockout tested previously[16]. Peptides with a motif score (MSS) below an empirical $P$ value of 0.05 were considered a 'match'. Peptides with an MSS above 0.05 in $P$ value were considered 'no match'.

The sequence logo of phosphoproteome targets (for example in Fig. 4c) represents the relative frequency of amino acids among peptides phosphorylated above background levels ($\log_{10}$(signal >4.0)) for the FIKK kinase of interest. Sequence logos were generated using ggseqlogo[105].

## Expression and purification of *P. falciparum* FIKK13 kinase domain proteins for crystallization

Codon-optimized DNA encoding the kinase domain of *Pf*FIKK13 residues 149–561 (PlasmoDB PF3D7_1371700) was cloned into pET-47b to produce an HRV 3C cleavable $His_6$ N-terminal fusion (MA**HHHHHH**SAA<u>LEVLFQ</u>⇓GPG) with the HRV 3C cleavage site underlined. The kinase inactive D379N mutant was generated by site directed mutagenesis using the pET-47-*Pf*FIKK13[149–561] construct as a template. Both constructs were verified by DNA sequencing.

The *Pf*FIKK13[149–561] and *Pf*FIKK13[149–561_D379N] were expressed in *E. coli* strain BL21 (DE3) Gold (Agilent). Bacterial cultures were grown in TB at 30 °C to an $OD_{600}$ of 1.2–1.5 and isopropyl-β-D-thiogalactoside was added to a final concentration of 0.5 mM to the culture grown at 25 °C overnight. Cell pellets were collected, resuspended in lysis buffer A (50 mM HEPES pH 7.5, 20 mM imidazole, 0.5 M NaCl, 10 mM $MgCl_2$, 10% (v/v) glycerol and 1 mM TCEP) supplemented with 1 U ml⁻¹ Universal nuclease (Pierce) and 1 Protease inhibitor tablet (cOmplete, Roche) per 50 ml solution, and lysed by sonication. The bacterial lysate was centrifuged for 30 min at 80,000$g$. The supernatant was applied to a 5 ml HisTrap column (Cytiva) and washed with 10 column volumes of buffer A. Fractions containing *Pf*FIKK13[149–561] and *Pf*FIKK13[149–561_D379N] were pooled separately and incubated overnight at 4 °C with HRV 3C protease to remove the 6xHis-tag. The next day, *Pf*FIKK13[149–561] and *Pf*FIKK13[149–561_D379N] were concentrated and purified by size-exclusion chromatography using buffer B (50 mM HEPES pH 7.5, 250 mM NaCl, 2 mM $MgCl_2$, 1 mM TCEP and 5% (v/v) glycerol) as running buffer.

## Generation of nanobodies recognizing *Pf*FIKK13[149–561]

A healthy llama (Arla) was immunized with three doses of 200 µg of purified *Pf*FIKK13[149–561] using GERBU as adjuvant following an established protocol[107] with animal handling carried out by trained personnel under the Home Office Project Licence PA1FB163A. The three immunizations took place on days 0, 28 and 56 and a 150 ml blood sample was collected 10 days after the third immunization. The peripheral blood mononuclear cells were isolated and total RNA extracted using described methods[108]. Total RNA was reverse transcribed using dT18-oligos. The VHH was PCR-amplified using primers CALL01 and CALL02 and a band of ~700 bp excised. The 700 bp band was used as template to re-amplify the VHH using primers VHH-Sfil2 (5′-GTCCTCGCAACTGCGGCCCAGCCGGCCATGGCTCAGGTGCAGCTGGTGGA-3′) and VHH-Not2 (5′-GGACTAGTGCGGCCGCTGAGGAGACGGTGACCTGGGT-3′). The PCR product was digested with *Sfi*I and *Not*I enzymes (NEB) and ligated into a pHEN2 vector modified with a triple c-Myc tag[109], which was used to transform electrocompetent TG1 cells (Lucigen). The resulting Nb library consisted of $5 \times 10^7$ independent colonies. Phage particles were prepared by super-infection with VCS13 helper phage (Agilent). The phage preparation was concentrated by adding 1/5 the volume of 20%(w/v) PEG 6000, 2.5 M NaCl and subsequent centrifugation at 4,000$g$ for 30 min.

Nanobodies specific for *Pf*FIKK13[149–561] were selected against biotinylated *Pf*FIKK13[149–561] immobilized on either streptavidin-coated magnetic beads (Dynabeads M280-Streptavidin, Thermo Fisher Scientific) or Pierce NeutrAvidin Coated plates (15123 Thermo Fisher Scientific). Individual clones were isolated by ELISA using soluble Nbs as primary antibody and detecting with the anti-C-myc antibody 9E10 (1;2,000), followed by anti-mouse-HRP conjugated antibodies (Agilent). The chromogenic TMB substrate (Thermo Fisher Scientific) was added, and colour development was quenched with 1 M HCl. The absorbance was read at 450 nm with 620 nm used as baseline. ELISA-positive Nb clones were sequenced. A total of seven different families of Nbs were isolated. Further work was performed with Nb2G9 and Nb9F10.

## Expression and purification of Nb2G9 and Nb9F10

Nb2G9 (protein sequence QVQLVESGGGSVQAGGSLRLSCAASGRTFSSYSMAWFRQAPGKERENVAVISWSGSTSYYAESVKGRFTISRDNAKNTVYLQMNSLKPEDTAVYYCAAGPRTTPQAMGAVEYDYWGQGTQVTVSS) was found to be compatible with Nb9F10 (protein sequence QEQLVESGGGLVQAGGSLTLSGASSGGTFETYAMGWFRQAPGKEREFAAAVSWSGGSAHYADSVKGRFTISRDKVKNTVYLQMNSLKPEDTAVYY

CAADRSYGSSWYHYPEDALDAWGQGTQVTVSS) in simultaneously binding *Pf*FIKK13[149–561] (data not shown). The DNA encoding Nb2G9 and Nb9F10 were cloned into a modified pET-21b vector to produce a N-term pelB secretion signal fusion (MKYLLPTAAAGLLLLAAQPA⇓MA) with a C-term TEV cleavable Avi•Tag/His$_8$ (AA*ENLYFQ*⇓*GLNDIFEAQKIE WHE***HHHHHHHH** where the underlined sequence is the TEV cleavage site, the Avi•Tag in italics and the 8xHis-tag in bold).

Nb2G9 and Nb9F10 were expressed in *E. coli* strain Rosetta2(DE3). Bacterial cultures were grown in TB at 37 °C to a density of $OD_{600}$ of 1.2–1.5 and protein expression was induced with 0.5 mM isopropyl-β-d-thiogalactoside at 30 °C overnight. The bacteria were pelleted by centrifugation at 5,000$g$ for 30 min. The clarified TB medium containing nanobodies was adjusted to pH 8.0, 20 mM NaCl and loaded onto a 5 ml HiTrap Excel column (Cytiva) pre-equilibrated in PBS. Bound proteins were eluted with PBS containing 400 mM imidazole pH 8.0. The C-terminal Avi/His$_8$ tag was removed by TEV protease cleavage and subsequent size-exclusion chromatography.

### FIKK13/Nb2G9/Nb9F10 complex formation

Purified *Pf*FIKK13[149–561_D379N] and Nb2G9 and 9F10 were mixed in a 1:1.2:1.2 molar ratio and loaded on a Superdex 75 10/300 Increase column (Cytiva) equilibrated in 20 mM HEPES pH 7.5, 150 mM NaCl, 2 mM $MgCl_2$ and 0.5 mM TCEP to remove the excess nanobodies. The fractions corresponding to the FIKK13[149–561_D379N]/Nb2G9/Nb9F10 complex peak were pooled, concentrated to 7 mg ml$^{-1}$ and used for crystallization experiments. Additionally, the FIKK13[149–561_D379N]/Nb2G9/Nb9F10/ATPγS samples were prepared by mixing purified FIKK13[149–561_D379N]/Nb2G9/Nb9F10 and ATPγS in a 1:1.1 molar ratio.

### Crystallization of *Pf*FIKK13[149–561_D379N] with Nb2G9, Nb9F10 and ATPγS

Crystallization trials were set up using samples at ~7 mg ml$^{-1}$. Initial crystals of the complex between *Pf*FIKK13[149–561_D379N] and Nb2G9 and Nb9F10 nanobodies were grown in 20% (w/v) PEG 3350 and 0.2 M sodium thiocyanate (Peg Ion HT screen condition B1, Hampton Research) and further optimized. Crystals were grown in sitting drops by vapour diffusion at 20 °C, cryoprotected by stepwise addition of PEG 4000 or ethylene glycol to a final concentration of 25% (v/v) and flash-cooled to 100 K by direct immersion in liquid nitrogen.

The initial apo (without ATPγS) structure was solved at low resolution by molecular replacement with PHASER using an AlphaFold search ensemble generated at Diamond Light Source using data obtained from crystals grown in 18% (w/v) PEG 3350, 150 mM sodium thiocyanate and 10 mM calcium chloride as an additive. The model was rebuilt and refined before molecular replacement into a higher resolution dataset obtained from crystals grown in 21% (w/v) PEG 3350 and 0.1 M Sodium thiocyanate, but is not reported here.

Crystals of *Pf*FIKK13[149–561_D379N] bound to ATPγS and in complex with Nb2G9 and Nb9F10 were obtained from a condition containing 0.1 M lithium chloride, 10% (v/v) ethylene glycol, 20% (w/v) PEG 6000 and 0.1 M HEPES pH 7.0 (ligand friendly screen condition C9, Molecular Dimensions) and seeding with apo crystals. Data were processed with the XIA2/DIALS pipeline at the Diamond Light Source. Molecular replacement with PHASER, in the resultant orthorhombic (P2$_1$2$_1$2$_1$) space group and utilizing the apo crystal complex coordinates yielded a poorly refinable solution and the ATPγS complex data were therefore reprocessed with the same pipeline in space group P2$_1$ with cell dimensions $a$ = 82.4 Å, $b$ = 121.7 Å, $c$ = 151.1 Å, $a$ = 90.0°, $b$ = 90.02° and $g$ = 90.0°. Model building and refinement were undertaken with COOT version 0.9.8 and REFMAC version 5.8.0430 within CCP4CLOUD. See Supplementary Table 17 for data collection and refinement statistics.

### Kinase-peptide models generation

Kinase-peptide models were generated using the HADDOCK 2.4 webserver[110] applied to AlphaFold2 predictions of the FIKK kinase domain[111,112]. Docking was executed using default parameters with the following alterations. First, residues with a minimum relative solvent accessibility of 5% could be considered as accessible. Second, the peptide sequence was designated as the molecule type 'Peptide' and defined to be fully flexible at every position. No 'active' or 'passive' residues were chosen as ambiguous interactions restraints[113], but an unambiguous interaction restraint was specified between the phosphoacceptor (S, T or Y) side-chain oxygen and the hydroxyl oxygen of the catalytic aspartate residue (D166 in PDB: 1ATP), at a maximum distance of 5.5 Å.

### PKIS screen and SAR assays

PKIS and SAR compounds at 1 mM in 60 nl DMSO were plated in white, opaque, flat-bottomed, 384-well microplates (Greiner Bio-One). Columns 6 and 18 of the microplates served as controls. Column 6 was a positive control (recombinant FIKK8 kinase domain + $P_o$ peptide (RRRAPSFYRK)[22] + ATP without compounds). Column 18 was a negative control (recombinant FIKK8 kinase domain + $P_o$ peptide − ATP). First, 3 µl kinase at 40 nM in kinase reaction buffer (20 mM MOPS, 10 mM magnesium chloride and 10 mM manganese chloride, pH 7.4 (Alfa Aesar)) was dispensed in each well using a Multidrop Combi Reagent dispenser (Thermo Fisher Scientific). Recombinant FIKK8 kinase domain was left to incubate in the presence of compounds for 15 min at room temperature. Then, 3 µl $P_o$ peptide + ATP at 20 µM each, diluted in kinase reaction buffer, was dispensed in each well except in column 18 in which 3 µl of peptide without ATP was dispensed. The kinase reaction was left to occur for 1 h and was stopped with 6 µl ADP-Glo reagent. Kinase activity was assessed after addition of 12 µl kinase detection reagent by measuring luminescence on a multimode microplate reader FLUOstar Omega (BMG Labtech).

### In vitro measurement of compounds IC$_{50}$

In vitro IC$_{50}$ values of PKIS and SAR compounds were determined by testing recombinant FIKK8 kinase domain activity in the presence of increasing concentrations of compounds. White, opaque, flat-bottomed 384-well microplates (Greiner Bio-One) containing compounds in a range of concentrations starting from 25 µM to 0.4 nM (1 in 3 serial dilutions) were ordered from GSK in Stevenage. Compounds were dispensed in the microplates at the required concentrations in 60 nl DMSO. Kinase activity in the presence of the compounds was measured as described above. Briefly, 3 µl recombinant FIKK8 kinase domain at 40 nM in kinase reaction buffer was dispensed in each well using a Multidrop Combi Reagent dispenser (Thermo Fisher Scientific). Recombinant FIKK8 kinase domain was left to incubate in the presence of compounds for 15 min at room temperature. Then, 3 µl $P_o$ peptide + ATP at 20 µM each, diluted in kinase reaction buffer, was dispensed in each well. Kinase reaction was left to occur for 1 h and was stopped with 6 µl ADP-Glo reagent. Kinase activity was assessed after addition of 12 µl kinase detection reagent by measuring luminescence on a multimode microplate reader FLUOstar Omega (BMG Labtech). Data were analysed using GraphPad Prism version 10 and IC$_{50}$ values were calculated from a four-parameter logistical fit of the data.

### FIKK inhibitors EC$_{50}$ determination

EC$_{50}$ values of the different compounds tested were determined by flow cytometry. Twofold dilutions of the compounds were plated in triplicate in 96-well plates. Then, 200 µl parasite solution containing 1% NF54 parasitaemia and 2% haematocrit was added to each well and plates were incubated for 72 h at 37 °C in a sealed gassed chamber. After incubation, parasite growth was assessed by flow cytometry. Samples (20 µl) were fixed in 2% paraformaldehyde + 0.2% glutaraldehyde in PBS for 1 h in the dark at 4 °C. Fixative was subsequently washed out with PBS and samples were stained with SYBR Green for 30 min in the dark at 37 °C. After a final wash, parasitaemia was counted by flow cytometry on a BD LSR Fortessa flow cytometer (Becton Dickinson) using the FACS

Diva software. Data were analysed using the FlowJo 10 analysis software (Becton Dickinson) and are available in Supplementary Table 18.

## ATP depletion

Irreversible depletion of ATP in RBCs was carried out by incubating uRBCs for 2 h at room temperature in PBS containing various concentrations of inosine and iodoacetamide[62] (see Supplementary Fig. 11 for the inosine and iodoacetamide concentrations used). ATP-depleted uRBCs were then washed three times with PBS and ATP depletion was evaluated for each dilution using the CellTiter-Glo→ Luminescent Cell Viability assay (Promega) following the instructions provided in the kit (see Supplementary Fig. 11 for ATP-depletion assessment). ATP-depleted uRBCs were put in the presence of Percoll-purified mature schizont stage parasites at 1% haematocrit in complete medium in a shaking incubator at 37 °C for 4 h. Parasites were allowed to grow for 48 h before samples were taken for immunofluorescence, western blot and flow cytometry assessment of parasite growth (Supplementary Table 19) as described above.

## Reporting summary

Further information on research design is available in the Nature Portfolio Reporting Summary linked to this article.

## Data availability

The mass spectrometry proteomics data have been deposited to the ProteomeXchange Consortium via the PRIDE[114] partner repository with the dataset identifier PXD048966. The crystal structure of *Pf*FIKK13[149−561_D379N] with Nb2G9, Nb9F10 and ATPγS is available from the Protein Data Bank under the accession code 9EMY. Gene sequences and annotations for *P. falciparum* 3D7 were acquired from PlasmoDB. org (v46)[13] and human sequences were acquired from Uniprot.org (2023)[84]. RNA-sequencing data from Hoeijmakers et al. (accession number GSE66185) available on PlasmoDB (www.PlasmoDB.org) was also used. The Pf3K project dataset used to identify genetic variants in fikk genes is available at ref. 29. Source data in the form of unprocessed gels and western blots corresponding to Figs. 1c and 6d–f and Supplementary Figs. 1a,b, 2 and 11c are available with the article. Source data are provided with this paper.

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

## Acknowledgements

We thank members of the Treeck, Blackman, Knuepfer and Sateriale labs for critical discussions. We also thank the Crick Science Technology Platforms (Proteomic and Flow cytometry) for their outstanding technical support and training. We thank J. Rayner and L. Parish for the MAHRP1 and GAP50 antibodies, T. Spielmann for the SBP1 antibody and the European Malaria Reagent Depository for the FIKK4.2 and GAPDH antibodies. We thank E. Yeh for sharing insights on the proximity labelling experiments in malaria-infected RBCs. We thank GSK for its commitment to support fundamental discovery research through the establishment of the Crick–GSK LinkLabs partnership. We would also like to thank H. Lin, B. Jones and G. Stephens, University of Reading, for expert help with generation of nanobodies under the authority of PA1FB163A. We thank D. Chen and E. Winzeler for sharing dN/dS ratios. We thank T. Holder and M. Higgins for their valuable insights and constructive discussions regarding the paper. Special thanks to PlasmoDB and VEuPathDB for providing critical resources. M.T. received funding from the ERC (ERC grant no. 101044428), the Francis Crick Institute (grant nos. CC2132 and CR2023/030/2132) and FCT–Fundação para a Ciência e Tecnologia, I.P. (grant no. 2023.06167.CEECIND). M.T.B. and L.W. were supported by the Francis Crick Institute (CC2030). The Francis Crick Institute and its Science Technology platforms receive core funding from Cancer Research UK, the UK Medical Research Council and the Wellcome Trust (grant no. CC0199). S.D.N. is funded by an Early-Career Award Wellcome Trust grant (no. 225686/Z/22/Z). C.R.L. and D.B. were supported by a Canadian Institutes of Health Research Foundation grant (no. 387697) and a Human Frontier Science Program research grant (no. RGP34/2018) to C.R.L. C.R.L. holds the Canada Research Chair in Cellular Systems and Synthetic Biology. D.B. was supported by an EMBO Long-Term Fellowship (grant no. ALTF 1069-2019).

## Author contributions

H.B. performed the parasite genetic manipulations and phenotypic analysis, and the kinase activity assays on peptides and membranes. D.B. performed the bioinformatics analysis with input from C.R.L. S.D.N. and H.B. performed the gametocyte experiments. H.B., H.D. and M.B. processed the proteomic samples. H.B., D.B. and M.B. analysed the proteomic data. H.B., E.C. and D. Jones expressed and purified the recombinant proteins. H.B. and D.B. analysed the substrate specificity data. M.T.B. performed the cyclic peptide screen and SPR under supervision from L.W. D. Joshi generated the peptide libraries and synthetic peptides under the supervision of N.O. E.C., A.G.P., R.W.O. and S.K. generated protein crystals and solved the protein crystal structure. D.B designed the FIKK mutants. A.C. performed the field isolates genome analysis. H.B. performed the inhibitor screens under the supervision of A.P. and D.H. H.B. and M.T. conceived the study. H.B., D.B., S.D.N., H.D., M.T.B., S.K. and M.T. designed figures. H.B., D.B., S.D.N., M.T.B., A.C., S.K., C.R.L and M.T. wrote the original manuscript. All authors were involved in critically reviewing and editing the paper.

## Funding

## Competing interests

The authors declare no competing interests.

## Additional information

**Extended data** is available for this paper at https://doi.org/10.1038/s41564-025-02017-4.

**Correspondence and requests for materials** should be addressed to Moritz Treeck.

¹Signalling in Apicomplexan Parasites Laboratory, The Francis Crick Institute, London, UK. ²Gulbenkian Institute for Molecular Medicine, Lisbon, Portugal. ³Département de Biochimie, de Microbiologie et de Bio-informatique, Faculté des Sciences et de Génie, Université Laval, Québec, Quebec, Canada. ⁴Institut de Biologie Intégrative et des Systems, Université Laval, Québec, Quebec, Canada. ⁵PROTEO, Le Groupement Québécois de Recherche sur la Function, l'Ingénierie et les Applications des Proteins, Université Laval, Québec, Quebec, Canada. ⁶Structural Biology Science Technology Platform, The Francis Crick Institute, London, UK. ⁷Protein–Protein Interaction Laboratory, The Francis Crick Institute, London, UK. ⁸Chemical Biology Science Technology Platform, The Francis Crick Institute, London, UK. ⁹CrickGSK Biomedical LinkLabs, GSK, Stevenage, UK. ¹⁰LPHI, MIVEGEC, INSERM, CNRS, IRD, University of Montpellier, Montpellier, France. ¹¹These authors contributed equally: Hugo Belda, David Bradley. ✉e-mail: moritz.treeck@gimm.pt

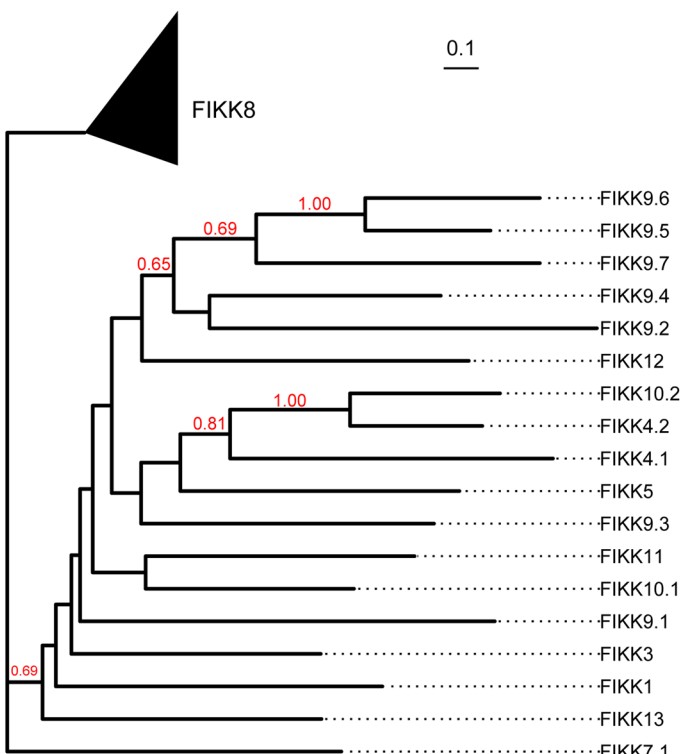

**Extended Data Fig. 1 | Phylogenetic tree of *Pf*FIKK kinases rooted on FIKK8 sequences.** Maximum-likelihood phylogenetic tree of *P. falciparum* FIKK kinase sequences (see Methods). The tree was rooted using known FIKK8 sequences across *Plasmodium* species. Branch support was assessed using 100 bootstrap replicates[89] and is shown for branches with support > 0.5. The scale bar represents the number of substitutions per site, that is two sequences separated by this distance have diverged by 0.1 substitutions per site on average.

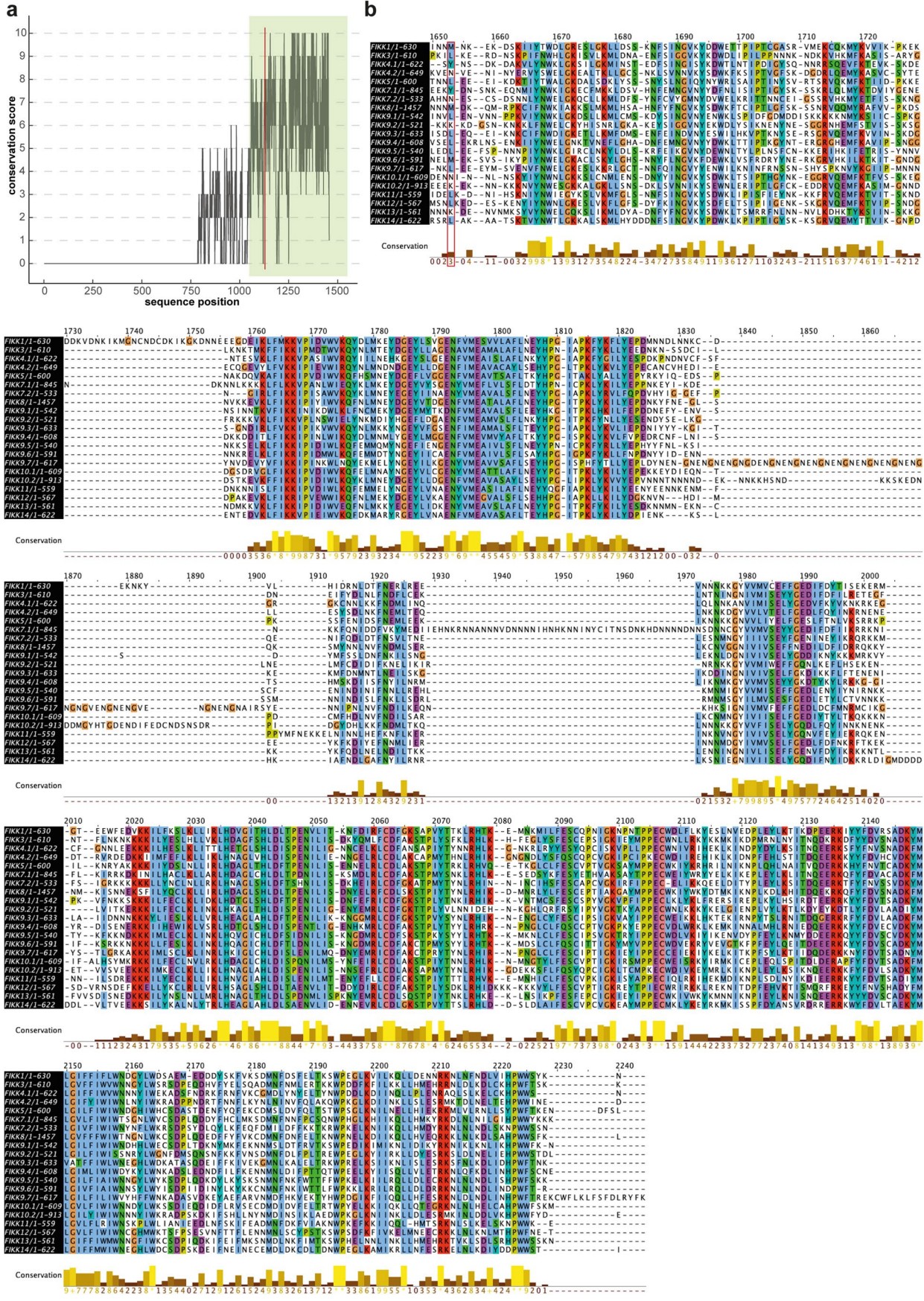

**Extended Data Fig. 2 | See next page for caption.**

**Extended Data Fig. 2 | Alignment of *P. falciparum* FIKK protein sequences allows for accurate determination of the FIKK kinase domain starting amino acid. a**, Amino acid sequence conservation in *P. falciparum* FIKKs was assessed using the PRALINE Multiple Sequence Alignment Software[116]. Conservation values reflect the normalised average of BLOSUM62 scores for each alignment column and range from 0 (low conservation) to 10 (high conservation). Sequence position is with respect to *P. falciparum* FIKK8 as the reference sequence. Green shading illustrates the FIKK kinase domain. The eponymous F-I-K-K motif is represented in red. **b**, Alignment of all FIKK sequences from *P. falciparum* including pseudokinases FIKK7.2 and FIKK14 using the T-Coffee multiple sequence alignment program[117] available in the Jalview software[118]. Shown is the alignment for the FIKK kinase domain. FIKK4.2 insertion (residues 381-953)

has been removed to simplify visualisation. Encircled in red are the amino acids chosen as a starting point for recombinant expression of *P. falciparum* FIKK kinase domains. The ClustalX colour scheme was used to assign colour to amino acids with the following criteria: Blue – Hydrophobic (A, I, L, M, F, W, V, C); Red – Positively charged (K, R); Magenta – Negatively charged (D, E); Green – Polar (N, Q, S, T); Orange – Glycine (G); Yellow – Proline (P); Cyan – Aromatic (H, Y); White – unconserved amino acids. Below the alignment is indicated the conservation score which measures the number of physicochemical properties conserved for each column of the alignment. Its calculation is based on[119]. Conserved columns are indicated by * (score of 11), conservation score then ranges between 10 (high conservation) and 0 (no conservation). Hyphens denote gaps.

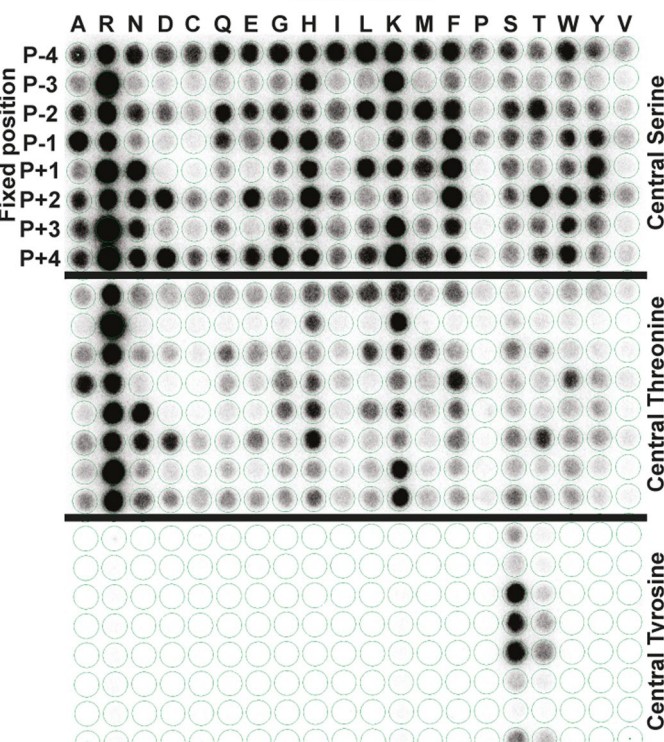

**Extended Data Fig. 3 | FIKK8 OPAL membrane.** An OPAL membrane constituted of 9-mer peptides with the general sequences A-X-X-X-X-S-X-X-X-X-A (top panel), A-X-X-X-X-T-X-X-X-X-A (middle panel) or A-X-X-X-X-Y-X-X-X-X-A (bottom panel) was used to assess FIKK8 preferred phosphorylation motif. X represents any natural amino acid except for S, T, Y or C. For each peptide, one of the 20 naturally occurring amino acids is fixed at each one of the 8 positions surrounding the phosphorylatable residue (S, T or Y). The membrane was incubated in the presence of recombinant FIKK8 kinase domain and [γ-32P]-ATP. After several washes, the membrane was exposed overnight to a phosphorscreen. The radioactivity incorporated into each peptide was determined by scanning the phosphorscreen with a phosphorimager giving the radiograph visible in this figure. Plotting the intensity pattern of the array enables the identification of preferred phosphorylation motifs (Fig. 3a(ii)) and reveals amino acids that are less favoured in a peptide sequence.

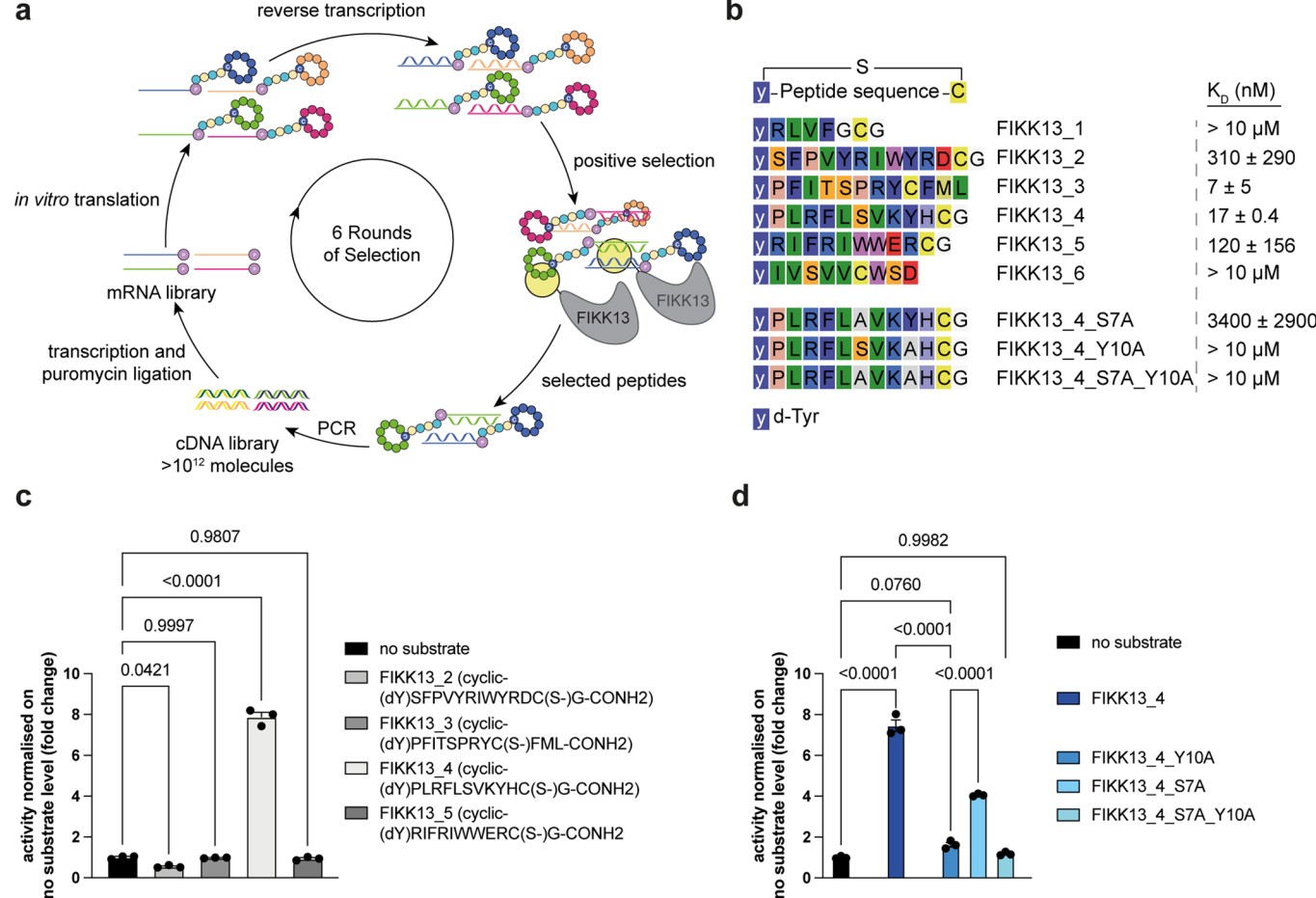

**Extended Data Fig. 4 | Identification of a tyrosine-based cyclic peptide as a substrate for FIKK13. a**, Scheme of the FIKK13 RaPID selection. **b**, Sequences and binding affinities of the different peptides recovered after 6 rounds of selection and different variants of the parent peptides. Peptides were initialised with d-Tyr and cyclised via a thioether bond between the N-terminus and the cysteine side chain. Binding affinities were measured by SPR (Supplementary Table 9) and show average ± standard deviation of at least 2 independent replicates. **c**, FIKK13 kinase domain phosphorylating activity on cyclic peptide identified in panel **b**. The results are represented as the mean ± SEM fold change compared to the no substrate luminescent signal obtained using the ADP-Glo assay. Statistical significance was determined using a one-way ANOVA followed by Dunnett's multiple comparison post-test (FIKK13_2 versus no substrate,

p = 0.0421; FIKK13_3 versus no substrate, p = 0.9997; FIKK13_4 versus no substrate, p < 0.0001; FIKK13_5 versus no substrate, p = 0.9807). n = 3 biological independent replicates. **d**, FIKK13 kinase domain phosphorylating activity on FIKK13_4 mutant peptides. The results are represented as the mean ± SEM fold change compared to the no substrate luminescent signal obtained using the ADP-Glo assay. Statistical significance was determined using a one-way ANOVA followed by Šidák's's multiple comparison post-test (FIKK13_4 versus no substrate, p < 0.0001; FIKK13_4_Y10A versus no substrate, p = 0.0760; FIKK13_4_S7A_Y10A versus no substrate, p = 0.9982; FIKK13_4_Y10A versus FIKK13_4, p < 0.0001; FIKK13_4_S7A versus FIKK13_4_Y10A, p < 0.0001). n = 3 biological independent replicates.

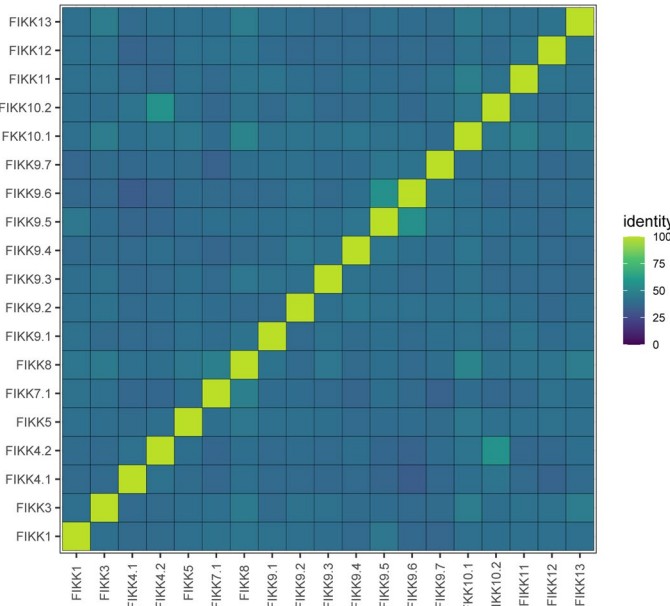

**Extended Data Fig. 5 | Protein sequence identity matrix of *P. falciparum* FIKK kinases.** Amino acid sequence identity on the basis of the *P. falciparum* multiple sequence alignment after removing poorly aligned regions.

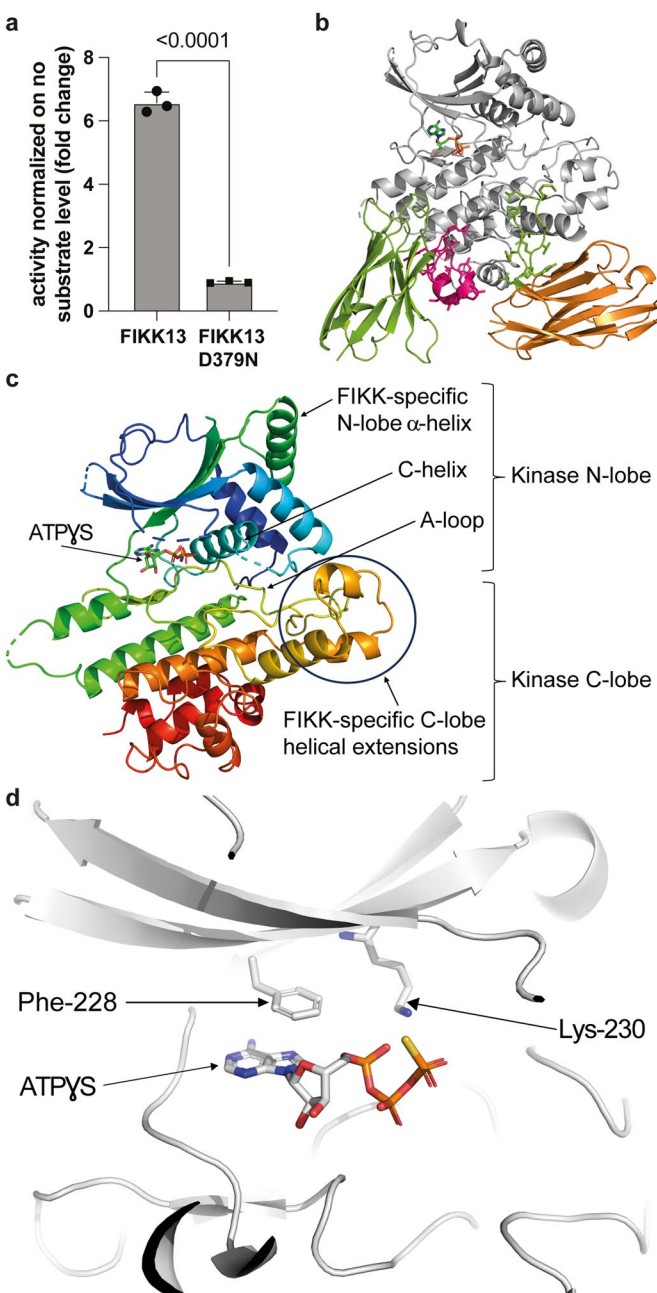

**Extended Data Fig. 6 | See next page for caption.**

**Extended Data Fig. 6 | FIKK13 D379N dead mutant crystal structure informs on ATP binding. a**, FIKK13 WT and FIKK13 D379N phosphorylating activity on cyclic peptide FIKK13_4. Results are represented as mean ± SEM fold change compared to the no substrate luminescent signal. Statistical significance was determined using a two-tailed t-test (FIKK13 D379N versus FIKK13, p < 0.0001). n = 3 biological replicates. **b**, FIKK13 kinase domain – in grey – bound to ATPɣS and complexed with Nb9F10 (olive, CDR3 in magenta) and Nb2G9 (orange, CDR3 in olive wrapping around the kinase C-lobe) **c**, FIKK13 D379N crystal structure with ATPɣS. The FIKKs N-lobe is compact with more features than ePKs including two α-helices packed above the conserved C-helix. The A-helix, rarely observed in kinase structures apart from the defining cAMP-dependent kinase PKA[47], marks the beginning of the N-lobe with the conserved Trp-162 (Extended Data Fig. 10) buried in a pocket between the narrow ends of the aligned A and B-helices positioned above the C-helix. The arrangement is capped by an FIKK-specific α-helix between the β4 and β5 strands. The mainly α-helical C-lobe contains, compared to ePKs, three additional α-helices inserted after the activation loop (A-loop). These helices directly interact with the A-loop,

potentially limiting its conformational flexibility upon phosphorylation, as seen in various ePKs[120]. The FIKK13 kinase domain catalytic machinery is conserved from ePKs with notable changes; the HRD motif where Asp acts as a general base during phospho-transfer, is conserved as [377]HLD[379]. The DFG motif, which can switch between active "DFG-in" and inactive "DFG-out" conformations[121], is present in FIKK13 as [398]DLS[400], although conserved as DFG in FIKK1 and FIKK9.1 (Extended Data Fig. 10) and adopts the "DFG-in" conformation in the FIKK13 kinase domain structure. **d**, Close-up representation of FIKK13 kinase domain ATP-binding pocket containing ATPɣS, focusing on the F-I-K-K motif. The size and hydrophobicity of Phe-228 restrict the ATP-binding pocket volume while the Lys-230 coordinates the nucleotide α- and β-phosphates and forms a salt bridge with Glu-261 of the C-helix, characteristic of active ePKs[47]. Taken together, the first experimentally determined FIKK kinase structure reveals strong resemblance to ePKs with conservation of the essential elements for catalysis. However, FIKK-specific features, such as additional α-helices in both the N- and C-lobe suggest differences in regulation.

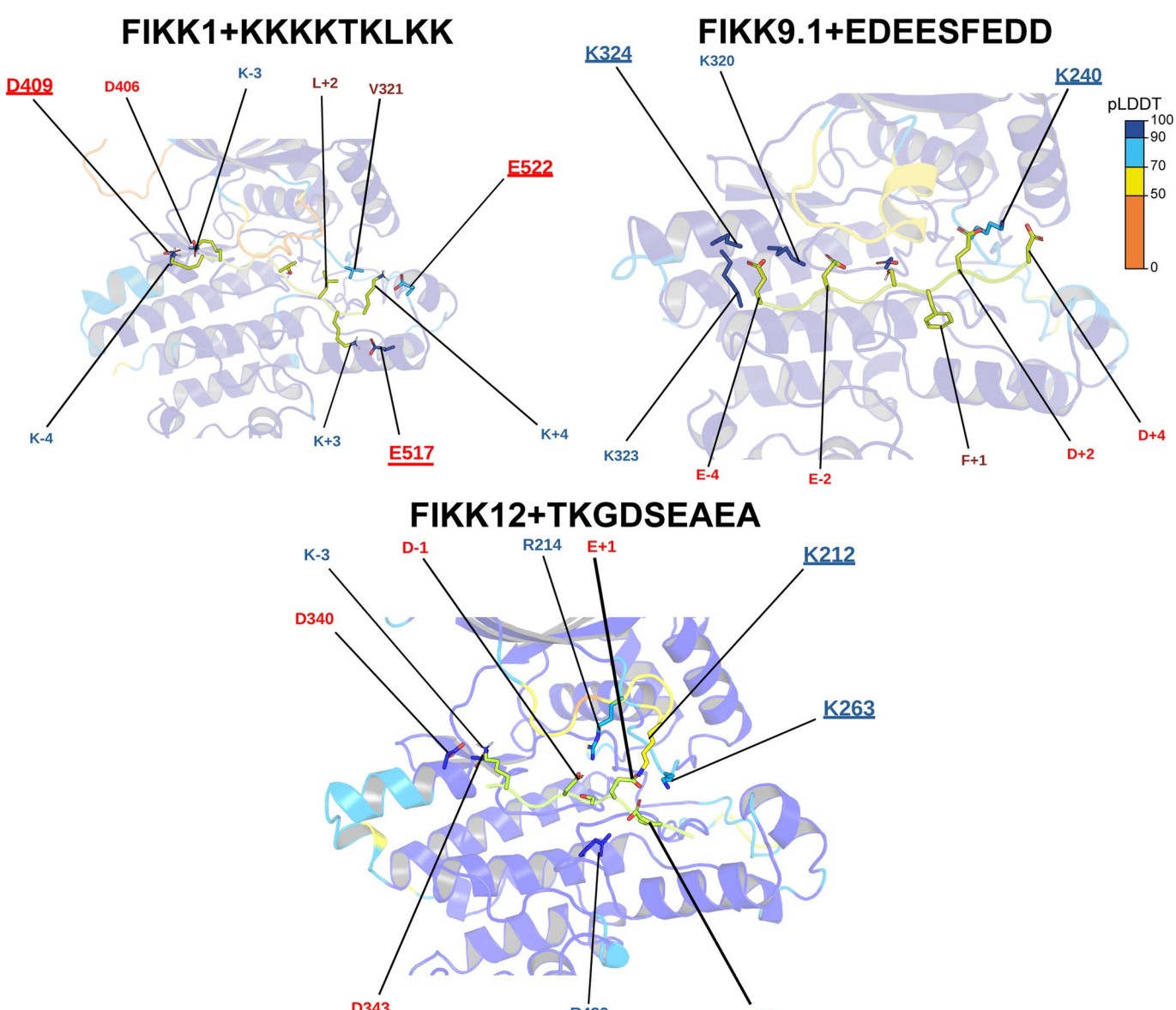

**Extended Data Fig. 7 | Target peptides of FIKK1, FIKK9.1, or FIKK12 modelled into the substrate-binding groove of the FIKK AlphaFold structures (see Methods).** Peptides may correspond to a likely target peptide of the kinase, or idealised targets based on the results of the OPAL arrays. FIKK kinase domain coloured according to the residues pLDDT score and the substrate peptide is coloured in green. Negatively charged amino acids are labelled in red, positively charged amino acids are labelled in blue, hydrophobic amino acids are labelled in brown.

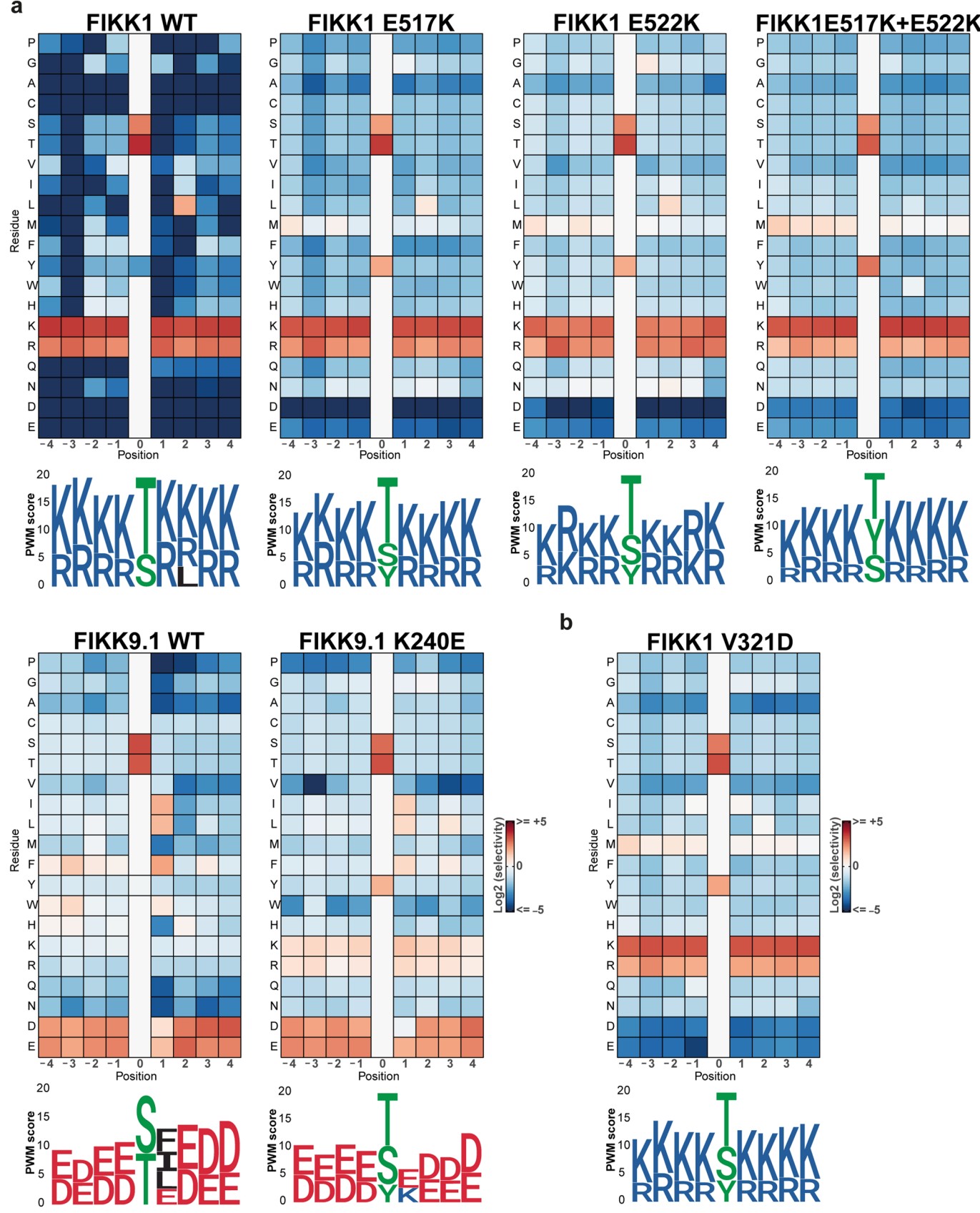

**Extended Data Fig. 8 | Substrate specificity assessment of FIKK1 and FIKK9.1 kinase mutants using OPAL arrays. a**, FIKK1 and FIKK9.1 wild type and mutant phosphorylation activity on OPAL membranes represented as heatmaps (see Fig. 3a(i) caption). Below is represented the PWM logos (see Fig. 3a(ii) caption). **b**, FIKK1 V321D mutant phosphorylation activity on OPAL membranes represented as a heatmap with PWM logo below.

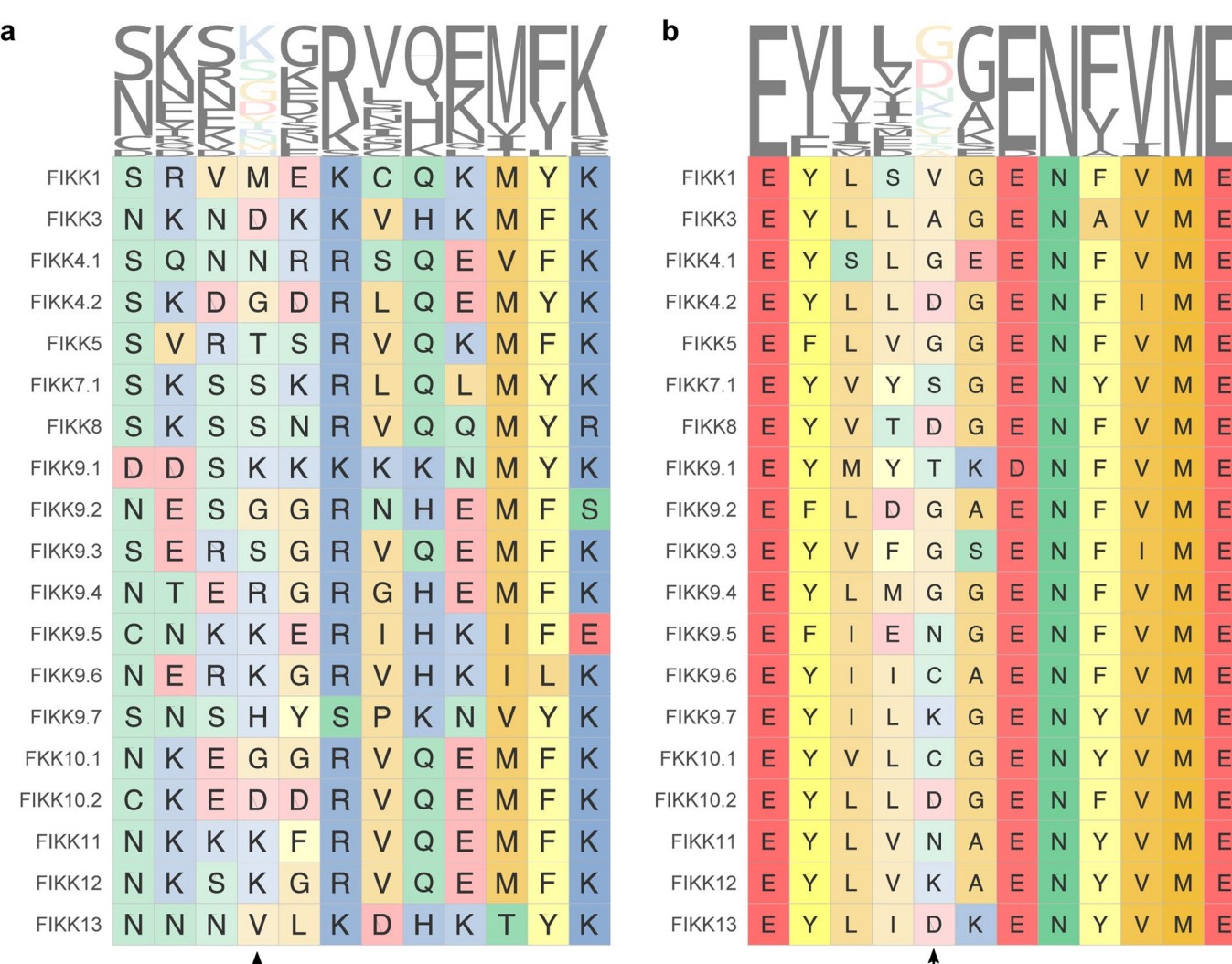

**Extended Data Fig. 9 | Sequence conservation of FIKK specificity determinants. a**, Conservation of the region surrounding FIKK12 K212 between FIKK paralogues in *P. falciparum*. The position containing FIKK12 K212 is labelled with an arrow. **b**, Conservation of the region surrounding FIKK12 K263 between FIKK paralogues in *P. falciparum*. The position containing FIKK12 K263 is labelled with an arrow.

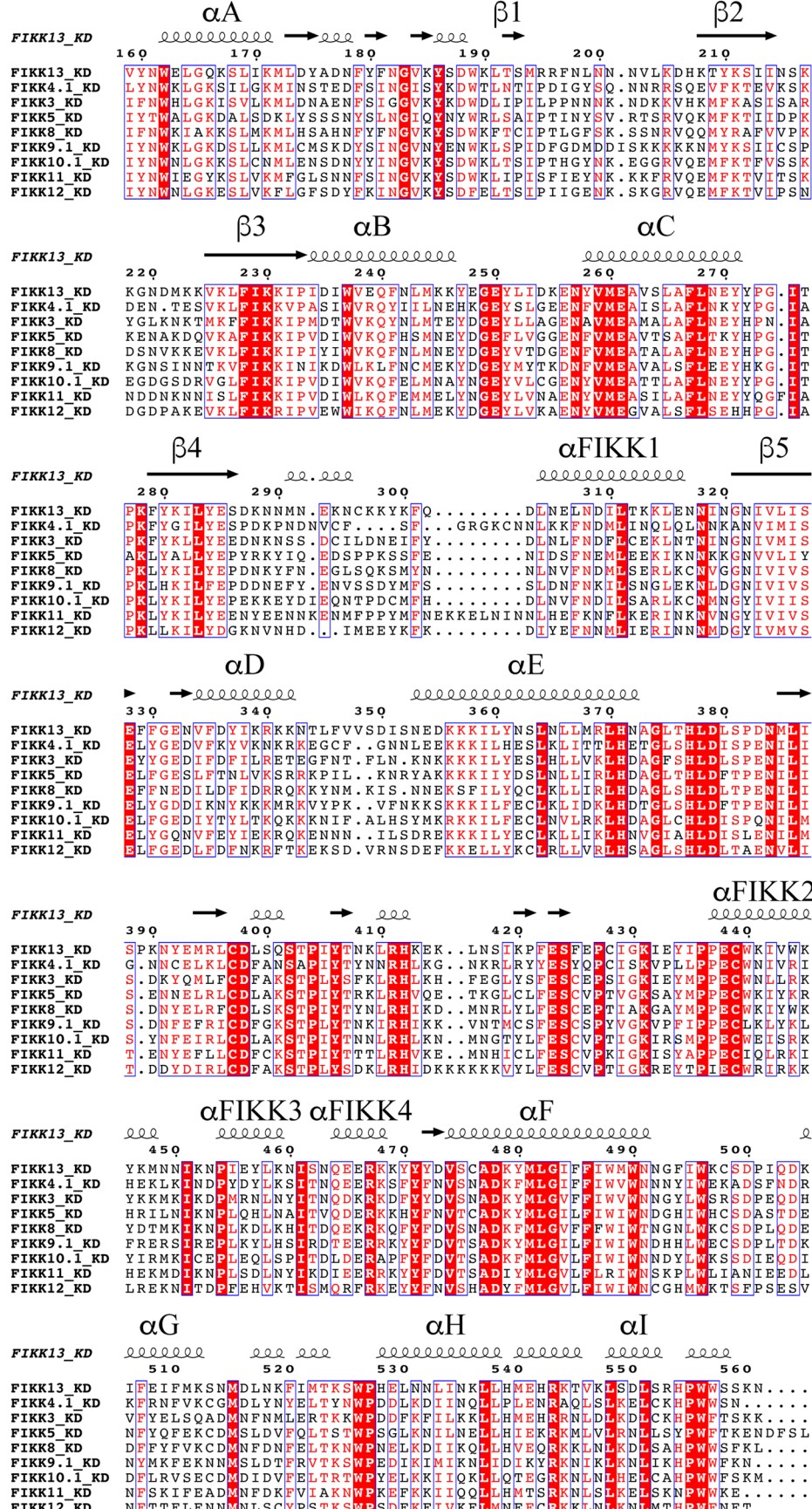

**Extended Data Fig. 10 | Multiple sequence alignment of various kinase domains.** Alignment generated using ESPript 3.0[122]. The secondary elements in FIKK13 are shown above the alignment. The α-helices and β-strands corresponding to ePKs are labelled. The αFIKK (1–4) are additional alpha-helices found in the FIKK family of kinases, but not in ePKs.

| | |
|---|---|

# Reporting Summary

## Statistics

For all statistical analyses, confirm that the following items are present in the figure legend, table legend, main text, or Methods section.

| n/a | Confirmed | |
|---|---|---|
| ☐ | ☒ | The exact sample size (*n*) for each experimental group/condition, given as a discrete number and unit of measurement |
| ☐ | ☒ | A statement on whether measurements were taken from distinct samples or whether the same sample was measured repeatedly |
| ☐ | ☒ | The statistical test(s) used AND whether they are one- or two-sided *Only common tests should be described solely by name; describe more complex techniques in the Methods section.* |
| ☒ | ☐ | A description of all covariates tested |
| ☐ | ☒ | A description of any assumptions or corrections, such as tests of normality and adjustment for multiple comparisons |
| ☐ | ☒ | A full description of the statistical parameters including central tendency (e.g. means) or other basic estimates (e.g. regression coefficient) AND variation (e.g. standard deviation) or associated estimates of uncertainty (e.g. confidence intervals) |
| ☐ | ☒ | For null hypothesis testing, the test statistic (e.g. *F*, *t*, *r*) with confidence intervals, effect sizes, degrees of freedom and *P* value noted *Give P values as exact values whenever suitable.* |
| ☒ | ☐ | For Bayesian analysis, information on the choice of priors and Markov chain Monte Carlo settings |
| ☒ | ☐ | For hierarchical and complex designs, identification of the appropriate level for tests and full reporting of outcomes |
| ☐ | ☒ | Estimates of effect sizes (e.g. Cohen's *d*, Pearson's *r*), indicating how they were calculated |

*Our web collection on statistics for biologists contains articles on many of the points above.*

## Software and code

Policy information about availability of computer code

| Data collection | FACSDiva v8.0.1, Nikon Elements v4.30.02, Xcalibur v4.2.28.14 |
|---|---|
| Data analysis | Cytoscape v3.10.1, GraphPad Prism V10, FlowJo v10.8.2, MaxQuant v2.0.3.1, Perseus v1.5.0.9, Jalview 2.11.3.2, FIJI v2.1.0/1.53c, PyMOL 2.5.4, ImageLab 6.1, Haddock 2.4, IQ-TREE2 v2.0.7, ComplexHeatmap v2.12.1, ggtree v3.5.1.902, ggseqlogo v0.1, ggmsa v1.2.3, bio3d v2.4-4, bcftools v1.15.0, trimAI v2.0. CCP4 version 8, XIA2 3.17.0, DIALS 3.17.0, PHASER 2.8.3, REFMAC 5.8.0430, COOT 0.9.8 SPR data were analysed using the Biacore Insight Evaluation Software. |

For manuscripts utilizing custom algorithms or software that are central to the research but not yet described in published literature, software must be made available to editors and reviewers. We strongly encourage code deposition in a community repository (e.g. GitHub). See the Nature Portfolio guidelines for submitting code & software for further information.

## Data

Policy information about availability of data

All manuscripts must include a data availability statement. This statement should provide the following information, where applicable:
- Accession codes, unique identifiers, or web links for publicly available datasets
- A description of any restrictions on data availability
- For clinical datasets or third party data, please ensure that the statement adheres to our policy

The mass spectrometry proteomics data have been deposited to the ProteomeXchange Consortium via the PRIDE113 partner repository with the dataset identifier PXD048966. The crystal structure of PfFIKK13149-561_D379N with Nb2G9, Nb9F10 and ATPγS is available from the Protein Data Bank under the accession code 9EMY. Gene sequences and annotations for P. falciparum 3D7 were acquired from PlasmoDB.org (v46)13 and human sequences were acquired from Uniprot.org (2023)86. RNA sequencing data from Hoeijmakers et al. (accession number GSE66185) available on PlasmoDB (www.PlasmoDB.org) was also used. The Pf3K project dataset used to identify genetic variants in fikk genes is available at the following address (www.malariagen.net/projects/parasite.pf3k)29. Source data in the form of unprocessed gels and western blots corresponding to Figs. 1c, 6d, 6e, 6f and Supplementary Figs. 1a, 1b, 2, 11c are available with the article.

## Research involving human participants, their data, or biological material

Policy information about studies with human participants or human data. See also policy information about sex, gender (identity/presentation), and sexual orientation and race, ethnicity and racism.

| | |
|---|---|
| Reporting on sex and gender | N/A |
| Reporting on race, ethnicity, or other socially relevant groupings | N/A |
| Population characteristics | N/A |
| Recruitment | N/A |
| Ethics oversight | N/A |

Note that full information on the approval of the study protocol must also be provided in the manuscript.

# Field-specific reporting

Please select the one below that is the best fit for your research. If you are not sure, read the appropriate sections before making your selection.

☒ Life sciences      ☐ Behavioural & social sciences      ☐ Ecological, evolutionary & environmental sciences

For a reference copy of the document with all sections, see nature.com/documents/nr-reporting-summary-flat.pdf

# Life sciences study design

All studies must disclose on these points even when the disclosure is negative.

| | |
|---|---|
| Sample size | Sample sizes are described in figure legends and methods. No statistical method was used to predetermine sample size.<br>In most instances we performed experiments in biological triplicates, unless otherwise stated.<br>Only one instance (Extended Data Fig. 4d, comparing "no substrate" and "FIKK13_4_Y10A") yielded a borderline p-value of close to 0.05, with a non-significant p-value of 0.0760. However, the data clearly show that FIKK13 loses its ability to phosphorylate peptide 4 when the tyrosine is mutated to alanine, confirming its tyrosine specificity. In all other cases, the results were definitive, with no indication of being overpowered or underpowered experiments.<br>For flow cytometry, parasitemia was determined by measuring the number of parasite-infected cells in a total of 100,000 red blood cells. |
| Data exclusions | As stated in the material and methods, for the proximity-labelling experiments we removed non-biotinylated peptides and biotinylated peptides present only in the NF54 sample (wild type) from the analysis as they represent background peptides binding to the beads. Other than that, no data were excluded. |
| Replication | The number of repeats for each experiments is given either in the figure legends or in the methods.<br>The proximity-labelling experiment has been performed in biological triplicate for each condition. Additionally, each replicate was cultured in a blood coming from different donors. The high consistence between replicates, and the agreement with our previously published dataset (PMID:32284562) provided high level of confidence to the sites which were observed to be significantly enriched in the TurboID-fusion samples.<br>Peptide libraries experiments were mostly performed once per kinase. Some kinases have been tested several times on the peptide libraries, each time giving consistent results. Additionally, the consistency of preferred phosphorylation motifs between FIKK kinases from two different species give a high degree of confidence in the data.<br>Binding affinity of cyclic peptides identified as binding to FIKK13 were measured in at least 2 independent replicates.<br>ADP-Glo assays measuring kinase activity in the presence of different substrates were always performed in biological triplicates, except for Fig. |

7a where each inhibitor was tested in 6 technical replicates.
Screening of the PKIS library against FIKK8 was performed in biological duplicates. Structural-Activity Relationship assay and IC50 determination of compounds were performed in biological triplicates.
ATP levels in the ATP-depletion optimisation steps were measured in biological triplicates.

| Randomization | Randomisation was not relevant to this study as no subjective judgments were required about which data to include, exclude, or measure. |
| --- | --- |
| Blinding | Investigators were not blinded during data collection and/or analysis as measurements were performed on quantitative endpoints that are not subject to investigator bias. |

# Reporting for specific materials, systems and methods

We require information from authors about some types of materials, experimental systems and methods used in many studies. Here, indicate whether each material, system or method listed is relevant to your study. If you are not sure if a list item applies to your research, read the appropriate section before selecting a response.

## Materials & experimental systems

| n/a | Involved in the study |
| --- | --- |
| ☐ | ☒ Antibodies |
| ☐ | ☒ Eukaryotic cell lines |
| ☒ | ☐ Palaeontology and archaeology |
| ☐ | ☒ Animals and other organisms |
| ☒ | ☐ Clinical data |
| ☒ | ☐ Dual use research of concern |
| ☒ | ☐ Plants |

## Methods

| n/a | Involved in the study |
| --- | --- |
| ☒ | ☐ ChIP-seq |
| ☐ | ☒ Flow cytometry |
| ☒ | ☐ MRI-based neuroimaging |

## Antibodies

| Antibodies used | Anti-HA high affinity antibodies (Clone 3F10, Roche, Lot number: 62572200, Cat. no: 11867431001). Dilution WB: 1 in 1,000, IFA: 1 in 1,000.<br>Anti-V5 (SV5-Pk1, Abcam, Cat. no: ab27671, Lot no: GR3337308-16). Dilution WB: 1 in 1,000<br>Anti-MAHRP1 (not commercially available, gift from Julian Rayner and Lindsay Parish). Dilution WB: 1 in 2,000.<br>Anti-GAP50 (not commercially available, gift from Julian Rayner). Dilution WB: 1 in 2,000.<br>Anti Adducin phospho-726 (Abcam, Cat. no ab53093, Lot no: GR81840-3). Dilution WB: 1 in 1,500, IFA: 1 in 1,000.<br>Anti-SBP1 (not commercially available, gift from Tobias Spielmann). Dilution IFA: 1 in 10,000.<br>Anti-GAPDH MAb 7.2 (European Malaria Reagent Repository (EMRR: www.malariaresearch.eu)). Dilution WB: 1 in 10,000.<br>Anti-FIKK4.2 MAb 126 (European Malaria Reagent Repository (EMRR: www.malariaresearch.eu)). Dilution IFA: 1 in 1,000.<br>Anti-Biotin Polyclonal Ab (Bethyl Laboratories, Cat. no: 150-109A)<br>Anti-Biotin Polyclonal Ab (Abcam, Cat. no: ab53494)<br>Anti-c-Myc MAb (Clone 9E10, ThermoFisher Scientific, Cat. no: MA1-980). Dilution ELISA: 1 in 2,000<br>Anti-FIKK13 nanobodies, described in this paper |
| --- | --- |
| Validation | Commercially available antibodies were validated by the suppliers. All commercially available antibody had validation statement available on the website of the suppliers.<br>Anti-HA high affinity antibodies (Clone 3F10, Roche, Lot number: 62572200, Cat. no: 11867431001) https://www.sigmaaldrich.com/GB/en/product/roche/roahaha?srsltid=AfmBOopff2YHT2mLNVsXDGFweR0ETGiame4ZWRznhmDod33MXgeDmqej<br>Anti-V5 (SV5-Pk1, Abcam, Cat. no: ab27671, Lot no: GR3337308-16) https://www.abcam.com/en-us/products/primary-antibodies/v5-tag-antibody-sv5-pk1-ab27671?srsltid=AfmBOor-BHxHZhU2Cq1LQr6hwJPI_4Q1w99na1YfilVUD52v8pmIIIcT<br>Anti Adducin phospho-726 (Abcam, Cat. no ab53093, Lot no: GR81840-3) https://www.abcam.com/en-us/products/primary-antibodies/alpha-adducin-phospho-s726-antibody-ab53093?srsltid=AfmBOoo6HoOf5t5Cj4LkjSprZ_g52JEEkWEvt2SarZ9ulfGeoOHpUn_V<br>Anti-c-Myc MAb (Clone 9E10, ThermoFisher Scientific, Cat. no: MA1-980) https://www.thermofisher.com/antibody/product/c-Myc-Antibody-clone-9E10-Monoclonal/MA1-980<br>Anti-MAHRP1 - Antibody validation unpublished. In a previous publication we saw a single band in Plasmodium infected cells only (not uninfected RBCs) by western blot and IFA probed with the antibody showed staining of maurer's clefts, where MAHRP1 is known to localise, and no staining of uninfected RBC, confirming specificity. In this publication, we observe the same single band at the expected size by western blot.<br>Anti-GAP50 has been characterised previously by immunoblot and IFA (doi: 10.1016/j.molbiopara.2006.01.009).<br>Anti-SBP1 has been characterised previously by immunoblot (PMCID: PMC4864081).<br>Anti-GAPDH MAb 7.2 has been characterised previously by immunoblot and IFA (doi: 10.1016/0166-6851(83)90025-7) http://www.malariaresearch.eu/reagents/monoclonal-antibody/72-anti-gapdh.<br>Anti-FIKK4.2 MAb 126 has been characterised previously by immunoblot and IFA (doi: 10.1016/j.ijpara.2014.01.003) http://www.malariaresearch.eu/reagents/monoclonal-antibody/126-anti-fikk42.<br>Both anti-GAPDH and anti-FIKK4.2 monoclonal antibodies were provided by the European Malaria Reagent Repository.<br><br>Anti-FIKK13 nanobodies were selected by ELISA and binding was confirmed by the fact that the complex FIKK13+nanobodies eluted at a higher molecular weight than FIKK13 alone or FIKK13 + 1 nanobody. Moreover, both nanobodies can be observed bound to |

FIKK13 in the crystal structure.

# Eukaryotic cell lines

Policy information about cell lines and Sex and Gender in Research

| Cell line source(s) | The NF54 DiCre line was made within our lab by inserting the DiCre locus into the transmissible lab-strain NF54 - ref -Tiburcio, M. et al. (2019). "A Novel Tool for the Generation of Conditional Knockouts To Study Gene Function across the Plasmodium falciparum Life Cycle." MBio 10(5). doi: 10.1128/mBio.01170-19

Plasmodium knowlesi parasites adapted for cell culture were obtained from Rob Moon - ref - Moon, R. W. et al. "Adaptation of the genetically tractable malaria pathogen Plasmodium knowlesi to continuous culture in human erythrocytes". Proc Natl Acad Sci U S A 110, 531-536, doi:10.1073/pnas.1216457110 (2013).

Commercial E. coli BL21-Gold (DE3) cells (Stratagene) were used for recombinant protein expression. |
| --- | --- |
| Authentication | None were authenticated |
| Mycoplasma contamination | All cell lines (Plasmodium and bacteria used for recombinant expression) were not tested for mycoplasma contamination. |
| Commonly misidentified lines (See ICLAC register) | No commonly misidentified lines were used. |

# Animals and other research organisms

Policy information about studies involving animals; ARRIVE guidelines recommended for reporting animal research, and Sex and Gender in Research

| Laboratory animals | Llama Glama, 6 years old |
| --- | --- |
| Wild animals | No |
| Reporting on sex | A single animal was immunised to generate nanobodies. The sex of the animal is not indicated. |
| Field-collected samples | No |
| Ethics oversight | An established protocol with animal handling carried out by trained personnel under the Home Office Project Licence PA1FB163A was followed. |

Note that full information on the approval of the study protocol must also be provided in the manuscript.

# Plants

| Seed stocks | N/A |
| --- | --- |
| Novel plant genotypes | N/A |
| Authentication | N/A |

# Flow Cytometry

## Plots

Confirm that:

☒ The axis labels state the marker and fluorochrome used (e.g. CD4-FITC).

☒ The axis scales are clearly visible. Include numbers along axes only for bottom left plot of group (a 'group' is an analysis of identical markers).

☒ All plots are contour plots with outliers or pseudocolor plots.

☒ A numerical value for number of cells or percentage (with statistics) is provided.

## Methodology

| | |
|---|---|
| Sample preparation | All Flow Cytometry samples in this study were Plasmodium infected red blood cells. The blood was obtained from the National Health Service Blood and Transplant (NHSBT) service.<br><br>For the half-maximal effective concentration (EC50) of FIKK inhibitors, samples were fixed in 2% paraformaldehyde (PFA) + 0.2% glutaraldehyde (GA) in PBS for 1 hour in the dark at 4 degrees. Fixative was subsequently washed out with PBS and samples were stained with SYBR Green for 30 minutes in the dark at 37 degrees. |
| Instrument | BD LSRFortessa flow cytometer (Becton Dickinson) |
| Software | Collected with FACSDiva, analysed with FlowJo10 |
| Cell population abundance | Cells were not sorted, just analysed.<br>For parasitemia measurement for EC50 assessment, one hundred thousand singlet events for each sample were measured.<br>For parasitemia measurement for ATP delpetion optimisation, three hundred thousand singlet events for each sample were measured. |
| Gating strategy | Samples were first gated for single cells by FSC-A and SSC-A.<br>Infected cells were clearly labelled by SYBR Green |

☒ Tick this box to confirm that a figure exemplifying the gating strategy is provided in the Supplementary Information.

