## [Peer Review File · Nature Microbiology]

The fast-evolving FIKK kinase family of *Plasmodium falciparum* can be inhibited by a single compound

Corresponding Author: Dr Moritz Treeck

Version 0:

Reviewer comments:

Reviewer #1

(Remarks to the Author)

This is a comprehensive study of the substrate preferences of the FIKKs, an intriguing family of kinases that is specific to the *Laverania* sub-taxon of malaria parasites. The FIKK family and its recent radiation and expansion in *Laverania* (to the exclusion of other taxons of malaria parasites) were discovered 20 years ago, but the important question of substrate specificity of its members has not been addressed comprehensively. Here, the authors implement a peptide array approach and of an almost complete panel recombinant FIKKs (only two enzymes out of the 20 *P. falciparum* FIKKs were refractory to heterologous expression) to fill this important gap. This yielded a convincing picture of distinct substrate preferences, including for FIKKs that share subcellular localisation. Surprisingly, one of the enzymes has exclusive tyrosine kinase activity.

The crystal structure of an FIKK allows the author to map to a specific loop in the catalytic domain the residues that mediate substrate preference. This is another unexpected result, as substrate binding in typical kinases is mediated by specific pockets (mostly located in the in the C-terminal lobe) of the kinase.

In addition to substrate preferences, that authors examined thousands of field isolates to detect the prevalence of pseudogenes in the FIKK family. This led to very interesting insights on the essentiality/relaxed selection of some of the FIKKs (e.g. FIKK14) and on the ongoing evolutionary history of this family.

Finally, the authors present data relating to a screen for ATP competitive inhibitors, and show that some molecules display broad-spectrum inhibitory activity (with respect to the FIKK family).

Overall, this is a well-conducted and comprehensive study that represents a significant contribution to our understanding of the evolutionary history and function of the FIKK family. Beyond that, the study will be of great interest to the fields of structural kinatology and of molecular evolution.

This reviewer does not have any major issue with the manuscript, but the authors may want to consider the following points:

Turbo ID studies: TurboID is a powerful method to assign the precise molecular environment of a given protein, and the data that FIKK4.1 and FIKK4.2 overlap but also differ in this respect are compelling. It would be interesting to give some more information in Supplementary Table 6, for example indicate the full name of the interactor (when available). I suppose "N" in this Table means "non-downregulated"? please clarify in the legend. It would be beneficial to somehow link the information in this table S6 to that in Supplementary Table 13.

Substrate preference: It seems from Figure 3 that the peptide array did not include pre-phosphorylated residues. It may be of interest to investigate if FIKKs have a preference for motifs that contain pre-phosphorylated residues (like GSK3 or CK1, for example, which use primed phosphorylation as an important motif element). This might be of great interest with respect to possible hierarchical phosphorylation by FIKKs that share the same localisation. This is not required for this paper, but a line or two in the discussion might interest the reader.

Structural studies: "the determinants identified here map to kinase loop regions". This is a bit vague --please indicate the boundaries of this loop(s), and label them on the 3D structure. With respect to the Tyr specificity, the authors say "We were unable to predict the basis for the tyrosine specificity of FIKK13". However, the peptide used for modelling in Fig. 6b has a serine residue. One of the determinants for Tyr specificity in other cases is the depth of the cleft; for example, the Insulin Receptor kinase IRK (a tyrosine kinase) does not phosphorylate serines because the serine side-chain is too short to reach the catalytic site; unlike Ser, Tyr extends far enough into the catalytic cleft to be efficiently phosphorylated (see Figure 2 of PMID:17585314). It would be of interest to model a Tyr-carrying peptide in FIKK13, and in serine-directed FIKKs, and to determine if the size of the Tyr/Ser side chain can be part of the determinant for the discrimination between Tyr-targeted (FIKK13) and Ser-targeted FIKKs (all the other FIKKs). Finally, it would be of interest to compare the data for the FIKK13 structure with those from other published FIKKs (e.g. PMID:

Drug discovery: It would be good to display the structure of the best compounds "from different chemical series" (line 562) in an additional panel of Fig. 7. To address the problem "that the compounds engage one or multiple kinases other than the FIKKs" in live parasite assays, it would be good to mention if these compounds (or related compounds from the same series) have been

profiled on the human kinome (data on Plasmodium non-FlKK kinases are likely not available), and how promiscuous they may be. A comparison with another published FlKK inhibitor (target the Cryptosporidium FlKK, PMID: 28162900) would be of interest --could this be modelled into the Plasmodium FlKK structures?

The fact that all FlKK have a small gatekeeper residue (line 548) opens the possibility to conduct reverse genetics/chemical genetics studies to investigate the exact role of each FlKK individually, by replacing the gatekeeper with a large residue that can prevent inhibition by specific compounds (as had been done with PfPKG, for example). This is of course not required for this paper, but a mention of this approach in the Discussion might be of interest to the reader.

Minor/editorial points

Line 59-61: "The flkk genes are conserved in syntenic loci across the Laverania, arguing for a rapid expansion controlling important functions in host cell remodelling and pathogenesis ". This is misleading. One cannot predict function from chromosomal location.

Line 103: "6 Plasmodium species known to infect humans" (there may be more than 6).

Line 120: explain the name "FlKK" at the first occurrence.

Line 126 "but no other human-infecting species". This is true for all species, not only those infecting humans.

Line 185: A bit confusing. "FlKks that may have lost functions during infection of their human host" ?

Line 298: "P. falciparum kinases phosphorylate S and T residues within acidic and basic motifs. Phosphorylated Y residues and proline-directed motifs are rarely found". This may be because a bias to the most abundant/detectable sites. The plasmodium kinome includes many enzymes from the CMGC group (notably CDK/MAPK) which are proline-directed.

Line 300: "predicted tyrosine kinases are lacking from the genome" The proper way to say this is that the Plasmodium kinome does not include enzyme from the TyrK group.

Line 344" : "...the rapid evolution of this relatively young protein family likely due to selection pressure to subvert the host machinery" I do not follow the argument about subverting the host machinery. Please develop.

Reviewer #2

(Remarks to the Author)

Summary: In this study, the authors have used a combination of structural characterization and AlphaFold2 predictions and mutation studies to first identify and then confirm likely structural determinants of Plasmodial FlKK family kinase specificity. Using this information, the authors then demonstrated feasibility of developing pan-specific inhibitors of this kinase family unique to Apicomplexan parasites. This thorough and impactful work in a timely area presents a promising avenue for new anti-malarial treatments with reduced potential for development of resistance parasites.

Minor comments

1. It would be helpful if the AlphaFold2 predicted structures could be overlaid and included in a supplementary figure.

Reviewer #3

(Remarks to the Author)

In this paper, the authors describe the evolution and function of the expansion of the FlKK kinase gene family in Laverania. First, they focus on its evolution, then perform molecular docking and mutational analysis—finding that FlKK13 is a tyrosine kinase not yet reported in Laverania. In addition to other analyses, like co-localisation and determining the structures of a FlKK kinase, they propose an inhibitor.

Due to the sheer number of experiments performed, it felt very dense in places. However, I can imagine that the readers of Nature Microbiology will. However, I enjoyed how the discussion pulled all the different aspects of the study together.

My main criticism:

Why don't the authors apply a test for purifying selection? What are the dn/ds ratios in the different FlKK clusters? As evolution is a major point of the paper (including the title), is to use the vast MalariaGen datasets, combined with the Laverania to use more rigorous tests, with a P-value.

I understand that the FlKK evolved, but how well do they align? I can see from Extended Fig 3 and 15 that there are very conserved regions, also Fig 3a shows the conservation on the end. For the Extended figure 1 tree: how many aa were used for the alignments, like 1000? What is the unit (0.1) of the branch length?

Minor:

- Didn't the number of malaria death climbed again to 620,000?
- Figure 1a – why did you not add the gametocyte expression?
- Line 219 – evolved unique functions – I would argue that they are still FIKK kinases but phosphorylate different proteins. See also L 633 – please comment.
- Line 203 – why reference Fig 3a? The text talks about predicted active PfFIKK, however fig 3a is an AA conservation plot.
- In figure 4b you show genes from *P. gaboni* G01. But in the methods you mentioned that you used *Plasmodium gaboni* strain SY75.
- Where did you use the Mok et al data (Line 830)
- Line 1498 – don't forget to put the link for the code in the next submission.

Decision Letter:

15th May 2024

Dear Moritz,

Thank you for your patience while your manuscript "Evolution and inhibition of the FIKK effector kinase family in *P. falciparum*" was under peer-review at Nature Microbiology. It has now been seen by 3 referees, whose expertise and comments you will find at the of this email. You will see from their comments below that while they find your work of interest, some important points are raised. We are very interested in the possibility of publishing your study in Nature Microbiology, but would like to consider your response to these concerns in the form of a revised manuscript before we make a final decision on publication.

Please note that for a potential second round of review, we would also invite an additional referee with expertise in structural biology related to kinase structure and function. We would advise this referee to only assess these aspects of the study.

If you have not done so already please begin to revise your manuscript so that it conforms to our Article format instructions at <http://www.nature.com/nmicrobiol/info/final-submission/>

The usual length limit for a Nature Microbiology Article is six display items (figures or tables) and 3,000 words. We have some flexibility, and can allow a revised manuscript at 3,500 words, but please consider this a firm upper limit. There is a trade-off of ~250 words per display item, so if you need more space, you could move a Figure or Table to Supplementary Information.

Some reduction could be achieved by focusing any introductory material and moving it to the start of your opening 'bold' paragraph, whose function is to outline the background to your work, describe in a sentence your new observations, and explain your main conclusions. The discussion should also be limited. Methods should be described in a separate section following the discussion, we do not place a word limit on Methods.

Nature Microbiology titles should give a sense of the main new findings of a manuscript, and should not contain punctuation. Please keep in mind that we strongly discourage active verbs in titles, and that they should ideally fit within 90 characters each (including spaces).

Please include a data availability statement as a separate section after Methods but before references, under the heading "Data Availability". This section should inform readers about the availability of the data used to support the conclusions of your study. This information includes accession codes to public repositories (data banks for protein, DNA or RNA sequences, microarray, proteomics data etc...), references to source data published alongside the paper, unique identifiers such as URLs to data repository entries, or data set DOIs, and any other statement about data availability. At a minimum, you should include the following statement: "The data that support the findings of this study are available from the corresponding author upon request", mentioning any restrictions on availability. If DOIs are provided, we also strongly encourage including these in the Reference list (authors, title, publisher (repository name), identifier, year). For more guidance on how to write this section please see: <http://www.nature.com/authors/policies/data/data-availability-statements-data-citations.pdf>

To improve the accessibility of your paper to readers from other research areas, please pay particular attention to the wording of the paper's opening bold paragraph, which serves both as an introduction and as a brief, non-technical summary in about 150 words. If, however, you require one or two extra sentences to explain your work clearly, please include them even if the paragraph is over-length as a result. The opening paragraph should not contain references. Because scientists from other sub-disciplines will be interested in your results and their implications, it is important to explain essential but specialised terms concisely. We suggest you show your summary paragraph to colleagues in other fields to uncover any problematic concepts.

If your paper is accepted for publication, we will edit your display items electronically so they conform to our house style and will reproduce clearly in print. If necessary, we will re-size figures to fit single or double column width. If your figures contain several

parts, the parts should form a neat rectangle when assembled. Choosing the right electronic format at this stage will speed up the processing of your paper and give the best possible results in print. We would like the figures to be supplied as vector files - EPS, PDF, AI or postscript (PS) file formats (not raster or bitmap files), preferably generated with vector-graphics software (Adobe Illustrator for example). Please try to ensure that all figures are non-flattened and fully editable. All images should be at least 300 dpi resolution (when figures are scaled to approximately the size that they are to be printed at) and in RGB colour format. Please do not submit Jpeg or flattened TIFF files. Please see also 'Guidelines for Electronic Submission of Figures' at the end of this letter for further detail.

Figure legends must provide a brief description of the figure and the symbols used, within 350 words, including definitions of any error bars employed in the figures.

When submitting the revised version of your manuscript, please pay close attention to our [href="https://www.nature.com/nature-research/editorial-policies/image-integrity">Digital Image Integrity Guidelines. and to the following points below:](https://www.nature.com/nature-research/editorial-policies/image-integrity)

Please include a statement before the acknowledgements naming the author to whom correspondence and requests for materials should be addressed.

Finally, we require authors to include a statement of their individual contributions to the paper -- such as experimental work, project planning, data analysis, etc. -- immediately after the acknowledgements. The statement should be short, and refer to authors by their initials. For details please see the Authorship section of our joint Editorial policies at http://www.nature.com/authors/editorial_policies/authorship.html

* include a point-by-point response to any editorial suggestions and to our referees. Please include your response to the editorial suggestions in your cover letter, and please upload your response to the referees as a separate document.

* ensure it complies with our format requirements for Letters as set out in our guide to authors at www.nature.com/nmicrobiol/info/gta/

* state in a cover note the length of the text, methods and legends; the number of references; number and estimated final size of figures and tables

* resubmit electronically if possible using the link below to access your home page:

Link Redacted

*This url links to your confidential homepage and associated information about manuscripts you may have submitted or be reviewing for us. If you wish to forward this e-mail to co-authors, please delete this link to your homepage first.

Please ensure that all correspondence is marked with your Nature Microbiology reference number in the subject line.

Nature Microbiology is committed to improving transparency in authorship. As part of our efforts in this direction, we are now requesting that all authors identified as 'corresponding author' on published papers create and link their Open Researcher and Contributor Identifier (ORCID) with their account on the Manuscript Tracking System (MTS), prior to acceptance. This applies to primary research papers only. ORCID helps the scientific community achieve unambiguous attribution of all scholarly contributions. You can create and link your ORCID from the home page of the MTS by clicking on 'Modify my Springer Nature account'. For more information please visit www.springernature.com/orcid.

We hope to receive your revised paper within three weeks. If you cannot send it within this time, please let us know.

On a side note, I will also be attending BioMaIPar XX next week and am looking forward to meet you in person.

Yours sincerely,

Reviewer Expertise:

Referee #1: Phosphorylation in Plasmodium

Referee #2: P. falciparum structural biology

Referee #3: Plasmodium parasite evolution

Reviewers Comments:

Reviewer #1 (Remarks to the Author):

This is a comprehensive study of the substrate preferences of the FIKKs, an intriguing family of kinases that is specific to the Laverania sub-taxon of malaria parasites. The FIKK family and its recent radiation and expansion in Laverania (to the exclusion of other taxons of malaria parasites) were discovered 20 years ago, but the important question of substrate specificity of its members has not been addressed comprehensively. Here, the authors implement a peptide array approach and of an almost complete panel recombinant FIKKs (only two enzymes out of the 20 P. falciparum FIKKs were refractory to heterologous expression) to fill this important gap. This yielded a convincing picture of distinct substrate preferences, including for FIKKs that share subcellular localisation. Surprisingly, one of the enzymes has exclusive tyrosine kinase activity.

The crystal structure of an FIKK allows the author to map to a specific loop in the catalytic domain the residues that mediate substrate preference. This is another unexpected result, as substrate binding in typical kinases is mediated by specific pockets (mostly located in the in the C-terminal lobe) of the kinase.

In addition to substrate preferences, that authors examined thousands of field isolates to detect the prevalence of pseudogenes in the FIKK family. This led to very interesting insights on the essentiality/relaxed selection of some of the FIKKs (e.g. FIKK14) and on the ongoing evolutionary history of this family.

Finally, the authors present data relating to a screen for ATP competitive inhibitors, and show that some molecules display broad-spectrum inhibitory activity (with respect to the FIKK family).

Overall, this is a well-conducted and comprehensive study that represents a significant contribution to our understanding of the evolutionary history and function of the FIKK family. Beyond that, the study will be of great interest to the fields of structural kinasology and of molecular evolution.

This reviewer does not have any major issue with the manuscript, but the authors may want to consider the following points:

Turbo ID studies: TurboID is a powerful method to assign the precise molecular environment of a give protein, and the data that FIKK4.1 and FIKK4.2 overlap but also differ in this respect are compelling. It would be interesting to give some more information in Supplementary Table 6, for example indicate the full name of the interactor (when available). I suppose "N" in this Table means "non-downregulated"? please clarify in the legend. It would be beneficial to somehow link the information in this table S6 to that in Supplementary Table 13.

Substrate preference: It seems from Figure 3 that the peptide array did not include pre-phosphorylated residues. It may be of interest to investigate if FIKKs have a preference for motifs that contain pre-phosphorylated residues (like GSK3 or CK1, for example, which use primed phosphorylation as an important motif element). This might be of great interest with respect to possible hierarchical phosphorylation by FIKKs that share the same localisation. This is not required for this paper, but a line or two in the discussion might interest the reader.

Structural studies: "the determinants identified here map to kinase loop regions". This is a bit vague --please indicate the boundaries of this loop(s), and label them on the 3D structure. With respect to the Tyr specificity, the authors say "We were unable to predict the basis for the tyrosine specificity of FIKK13". However, the peptide used for modelling in Fig. 6b has a serine residue. One of the determinants for Tyr specificity in other cases is the depth of the cleft; for example, the Insulin Receptor kinase IRK (a tyrosine kinase) does not phosphorylate serines because the serine side-chain is too short to reach the catalytic site; unlike Ser, Tyr extends far enough into the catalytic cleft to be efficiently phosphorylated (see Figure 2 of PMID:17585314). It would be of interest to model a Tyr-carrying peptide in FIKK13, and in serine-directed FIKKs, and to determine if the size of the Tyr/Ser side chain can be part of the determinant for the discrimination between Tyr-targeted (FIKK13) and Ser-targeted FIKKs (all the other FIKKs). Finally, it would be of interest to compare the data for the FIKK13 structure with those from other published FIKKs (e.g. PMID:

Drug discovery: It would be good to display the structure of the best compounds "from different chemical series" (line 562) in an additional panel of Fig. 7. To address the problem "that the compounds engage one or multiple kinases other than the FIKKs" in live parasite assays, it would be good to mention if these compounds (or related compounds from the same series) have been profiled on the human kinome (data on Plasmodium non-FIKK kinases are likely not available), and how promiscuous they may be. A comparison with another published FIKK inhibitor (target the Cryptosporidium FIKK, PMID: 28162900) would be of interest --could this be modelled into the Plasmodium FIKK structures?

The fact that all FIKK have a small gatekeeper residue (line 548) opens the possibility to conduct reverse genetics/chemical genetics studies to investigate the exact role of each FIKK individually, by replacing the gatekeeper with a large residue that can prevent inhibition by specific compounds (as had been done with PfPKG, for example). This is of course not required for this paper, but a mention of this approach in the Discussion might be of interest to the reader.

Minor/editorial points

Line 59-61: "The fikk genes are conserved in syntenic loci across the Laverania, arguing for a rapid expansion controlling important functions in host cell remodelling and pathogenesis ". This is misleading. One cannot predict function from chromosomal location.

Line 103: "6 Plasmodium species known to infect humans" (there may be more than 6).

Line 120: explain the name "FIKK" at the first occurrence.

Line 126 "but no other human-infecting species". This is true for all species, not only those infecting humans.

Line 185: A bit confusing. "FIKKs that may have lost functions during infection of their human host" ?

Line 298: "P. falciparum kinases phosphorylate S and T residues within acidic and basic motifs. Phosphorylated Y residues and proline-directed motifs are rarely found". This may be because a bias to the most abundant/detectable sites. The plasmodium kinome includes many enzymes from the CMGC group (notably CDK/MAPK) which are proline-directed.

Line 300: "predicted tyrosine kinases are lacking from the genome" The proper way to say this is that the Plasmodium kinome does not include enzyme from the TyrK group.

Line 344": "...the rapid evolution of this relatively young protein family likely due to selection pressure to subvert the host machinery" I do not follow the argument about subverting the host machinery. Please develop.

Reviewer #2 (Remarks to the Author):

Summary: In this study, the authors have used a combination of structural characterization and AlphaFold2 predictions and mutation studies to first identify and then confirm likely structural determinants of Plasmodial FIKK family kinase specificity. Using this information, the authors then demonstrated feasibility of developing pan-specific inhibitors of this kinase family unique to Apicomplexan parasites. This thorough and impactful work in a timely area presents a promising avenue for new anti-malarial treatments with reduced potential for development of resistance parasites.

Minor comments

1. It would be helpful if the AlphaFold2 predicted structures could be overlaid and included in a supplementary figure.

Reviewer #3 (Remarks to the Author):

In this paper, the authors describe the evolution and function of the expansion of the FIKK kinase gene family in Laverania. First, they focus on its evolution, then perform molecular docking and mutational analysis—finding that FIKK13 is a tyrosine kinase not yet reported in Laverania. In addition to other analyses, like co-localisation and determining the structures of a FIKK kinase, they propose an inhibitor.

Due to the sheer number of experiments performed, it felt very dense in places. However, I can imagine that the readers of Nature Microbiology will. However, I enjoyed how the discussion pulled all the different aspects of the study together.

My main criticism:

Why don't the authors apply a test for purifying selection? What are the d_n/d_s ratios in the different FIKK clusters? As evolution is a major point of the paper (including the title), is to use the vast MalariaGen datasets, combined with the Laverania to use more rigorous tests, with a P-value.

I understand that the FIKK evolved, but how well do they align? I can see from Extended Fig 3 and 15 that there are very conserved regions, also Fig 3a shows the conservation on the end. For the Extended figure 1 tree: how many aa were used for the alignments, like 1000? What is the unit (0.1) of the branch length?

Minor:

- Didn't the number of malaria deaths climb again to 620,000?
- Figure 1a – why did you not add the gametocyte expression?
- Line 219 – evolved unique functions – I would argue that they are still FIKK kinases but phosphorylate different proteins. See also L 633 – please comment.
- Line 203 – why reference Fig 3a? The text talks about predicted active PfFIKK, however fig 3a is an AA conservation plot.
- In figure 4b you show genes from P. gaboni G01. But in the methods you mentioned that you used Plasmodium gaboni strain SY75.
- Where did you use the Mok et al data (Line 830)
- Line 1498 – don't forget to put the link for the code in the next submission.

Version 1:

Reviewer comments:

Reviewer #1

(Remarks to the Author)

The authors provided a comprehensive and satisfactory response to most comments and requests. They chose to decline to follow some of the reviewer's recommendations, but in these cases the justification they provided was perfectly adequate. I recommend publication of the paper -this will be a hallmark paper in the unfolding story of FIKK characterisation.

Reviewer #3

(Remarks to the Author)

Overall, the paper is nicely improved.

Concerning the evolutionary analysis: The dN/dS analysis taken from the cited paper just shows that nothing really happens in the FIKK family. It might be interesting to see that the two FIKK with the strongest signal of purifying selection are the FIKK4.x - which have known functions. But overall the authors provide a test and describe the evaluation, as indicated in the title.

However, they don't test the idea of geographical differences. There they fall back on the pseudogenisation. I wonder, if sequences should be curated carefully, as in SEA, many genomes have gone through recent bottlenecks and might influence the signal. Also, if the authors want to dive deeper into the pseudogenisation, it might be necessary to see if indels might have changed the open reading frame. On the other side this is not an in-depth evolutionary analysis.

So, the one thing I wonder, if for the following statement in the discussion:

We observe notable differences in pseudogenisation between geographical backgrounds, suggesting that the environment might impact FIKK relevance.

You want to have a test, to see if the difference is statistically significant

Minor:

Should the reference to the figure in line 386 be figure 3D?

Reviewer #4

(Remarks to the Author)

The work by Belda et al is a very nice and comprehensive study focusing on a subgroup of protein kinases that play important roles in Plasmodium and related protozoa.

I shall focus within the limits of my expertise, only to analyze some aspects of the structural biology work and analyses.

I believe some modifications are needed to warrant an accurate model to be reported and deposited in the PDB. The accuracy in processing and refining the crystallographic data and model might also improve the electron density maps (eventually useful to further confirm and/or extend biologically relevant conclusions).

Concerns and comments:

1. I believe the main conclusions concerning the crystallographic (and overall structural) data, are valid.

However, the crystal structure has been processed and refined inaccurately. Materials and Methods details with regards to how these were done are also extremely limited for any reader to understand and/or reproduce.

This has resulted in several significant errors, that once corrected, will probably improve statistics, and perhaps even locally, improve electron density features that may now be erroneous.

After careful analysis of the atomic coordinates and observed/refined structure factors that the authors kindly shared, it seems clear to me that the data were indexed in the wrong space group. I humbly believe this is not a monoclinic crystal with angle beta so close to 90°, it simply is an orthorhombic crystal.

This potential error is what likely explains that the PDB validation report detects a "significant" degree of twinning (see Section 4 within the report, with Xtriage detecting a twinning fraction of ~0.42 following twinning law h,-k,-l).

If instead the authors, for some valid reason—in which case they should clearly report about it in Materials and methods—, truly believe this is a twinned monoclinic crystal, then they should also of course take good consideration of it, to properly refine taking into account the twinning law.

This may all explain the rather high Rfree value and an important Rfree-Rwork gap for this resolution.

Several important geometry issues have also been left uncorrected (such as several cis peptide omega dihedrals for non pre-Pro residues, a pretty high number of bad fitting residues within density or RSRZ outliers, etc etc etc).

Also, several side chains (and sometimes individual atoms within some side chains) that the authors omit, are clearly interpretable, working through the difference Fourier maps: they should be added into the final model and refined.

I suggest the authors to reprocess the data after indexing in SG P212121 (instead of the monoclinic P21 that the authors chose).

And, to invest more careful work in manually rebuilding the atomic model, such that no major errors are left uncorrected. After what, corroborate any biologically relevant conclusions that may be drawn from the structural analysis.

2. Please use pLDDT scores (instead of TM) to quantify the reliability of AlphaFold-predicted structures. For instance, in Supplementary Table 11, and consistently throughout the text.

The Template Modelling (TM) score is generally considered to be relatively insensitive to local inaccuracies (Xu 2010 <https://doi.org/10.1093/bioinformatics/btq066>). Actually, higher than 1Å rmsd values (for some kinase pairs even >2Å) as the authors report for the superpositions, indicate that, while definitely reliable in terms of fold (which is actually an expected outcome when it comes to predicting ePKs), they will most certainly include pretty significant variations at the local level.

On the other hand, pLDDT (predicted local distance difference test) scores can be reported as an average value (to replace TM in Supp Table 11), but also convey greatest information when reported as a per residue index. A typical way to do this is to color the AF2-predicted models that you show in main and supplementary figures, with a ramp according to pLDDT (this score is saved in the AF2 models on the B-factor column, so that it is easy to color by this property). Doing so, any regions/loops that would happen to have pLDDTs ≤ 70 (which AlphaFold will classify as being low) should be considered cautiously.

3. Please do not include explicit hydrogen atoms at this resolution, as you have currently done. I guess this might have been the result of an unwanted "default" setting in your refinement program. It is certainly a good practice to add H atoms in riding positions while refining, but not to actually leave them and write them within the final mmCIF file (at 2.8Å resolution this doesn't make sense, and drives overfitting due to exceedingly high parameters/observables ratio).

4. The loop 194:204 (not included in the crystal model because of lack of electron density), is positioned such that it protrudes from the N-lobe towards the C-lobe of the ePK domain. Even though this loop might be mobile (hence the weak density; and, perhaps low pLDDT scores in AlphaFold predictions?), it will be covering the catalytic site due to its topologic position. It somehow resembles to an analogous N:C-lobe interaction as the one observed in Tyr-kinases like IRK (e.g. pdb 4IBM). Could this loop and/or its characteristics (length, properties) be linked to Tyr vs Ser/Thr specificity?

Following up on this point, and converging with a colleague reviewer's question, even though there is not an identical crevice as in IRK, it's not obvious to me that a deepened cleft can be ruled out. I don't know exactly how does FIKK12 look like (which the authors used in their response to compare with a S/T-specific variant; again what about pLDDT scores for relevant structural components when predictions are used?); but in any case it is very useful for these matters, to look at solvent accessible surface representations (instead of cartoons), getting a better grip of clefts, pockets or even enclosed cavities (which could open up by movement of loops/residues). I actually observe such a cavity in FIKK13, poised to open just beside the catalytic Asp. This cavity is delimited by N257, L252, E426 on one side, and Q401, I429, L378 on the other (and, Leu378, interestingly, is replacing the highly conserved Arg in the catalytic site-containing fragment known as 'HRD', here 'HLD' in FIKK13. Can you please comment (considering of course other S/T-specific ePKs)?

Extended Data Fig 2 does not allow to see all these sequence analyses/comparisons: it would be worth having a clearer and complete MSA of FIKKs as extended data figure. I did also read carefully the response to other reviewers, including an MSA with full length FIKKs: I admit fonts are extremely small there, and of course including the complete sequences makes it more difficult to focus on the ePK domain. Maybe my previous questions are already responded therein, I apologize if that's the case, yet I cannot be clear with the elements at hand.

5. You mention the 'HRD' motif substitution by HLD (as well as the modification of the conserved 'DFG' motif), in the caption to Ext Data Fig 6, but actually consider them to be "minor changes". I tend to disagree with that. HRD and DFG motifs play key roles, explaining why they are so well conserved among a vast number of ePKs.

6. Please discuss your findings in view of the recent publication by Gizzio et al 2024 (Nat Commun doi:10.1038/s41467-024-50812-0). Are there coincidences or contradictions? Coming back to the DFG motif analysis just referred to above (with a "locked active" configuration in the case of FIKK13), this discussion with an evolutionary context seems particularly useful.

7. Please consider the possibility that the APTgS moieties have been partially/fully hydrolyzed. Electron density seems weaker on the gamma P center; and, please remember that sulfur should be sitting on a Fourier electron density peak at least as large as the phosphorus atom, since they both have approx. same number of electrons. Another evidence that this group has been hydrolysed (at least partially: occupancy can be refined). ATPgS is a slowly hydrolyzable ATP variant; for actual inhibition you may want to prefer using AMP-PCP (a methylene replacing the bridging O between beta and gamma P). Again, any analyses such as these should be made once refinement is properly finished, as phases might improve quite a bit, and then maybe the thio-derivatized gamma phosphate is indeed clearly there at full height, I cannot be sure right now; also, the nucleotides show right now a rather weird discontinuity, which in the end may be corrected. Last but not least, normally ATP and derivatives are complexed with a Mg²⁺ cation: consider this in your refinement (or else, comment).

Decision Letter:

12th September 2024

Dear Moritz,

Thank you for your patience while your manuscript "Evolution and inhibition of the FIKK effector kinase family in *P. falciparum*" was under peer-review at Nature Microbiology, and please again accept our apologies for the delay in getting back to you as I have been out of the office the last two weeks. Your manuscript has now been seen by 3 referees, whose expertise and comments you will find at the end of this email. Although they find your work of some potential interest, they have raised a

number of concerns that will need to be addressed before we can consider publication of the work in Nature Microbiology.

In particular, while referee #1 is satisfied with the revised manuscript, referees #3 and #4 still have some important concerns that will need to be addressed. Specifically, referee #3 suggest to test the idea of geographical differences (however, this is optional), and to add a statistical test for the observation of differences in pseudogenisation between geographical backgrounds. Referee #4 (who was recruited for the second round of review to assess the structural biology aspects of the study) has some important concerns regarding the structural analysis, modeling, and presentation of the data. These concerns are comprehensive and will need to be carefully addressed in full. I should also note that we would go back to referee #4 to check a potentially revised manuscript.

Should further experimental data allow you to address these criticisms, we would be happy to look at a revised manuscript.

Please include a data availability statement as a separate section after Methods but before references, under the heading "Data Availability". This section should inform readers about the availability of the data used to support the conclusions of your study. This information includes accession codes to public repositories (data banks for protein, DNA or RNA sequences, microarray, proteomics data etc...), references to source data published alongside the paper, unique identifiers such as URLs to data repository entries, or data set DOIs, and any other statement about data availability. At a minimum, you should include the following statement: "The data that support the findings of this study are available from the corresponding author upon request", mentioning any restrictions on availability. If DOIs are provided, we also strongly encourage including these in the Reference list (authors, title, publisher (repository name), identifier, year). For more guidance on how to write this section please see: <http://www.nature.com/authors/policies/data/data-availability-statements-data-citations.pdf>

* If you have not done so already we suggest that you begin to revise your manuscript so that it conforms to our Article format instructions at <http://www.nature.com/nmicrobiol/info/final-submission>. Refer also to any guidelines provided in this letter.

When submitting the revised version of your manuscript, please pay close attention to our [href="https://www.nature.com/nature-portfolio/editorial-policies/image-integrity">Digital Image Integrity Guidelines](https://www.nature.com/nature-portfolio/editorial-policies/image-integrity) and to the following points below:

Link Redacted

Note: This url links to your confidential homepage and associated information about manuscripts you may have submitted or be reviewing for us. If you wish to forward this e-mail to co-authors, please delete this link to your homepage first.

Nature Microbiology is committed to improving transparency in authorship. As part of our efforts in this direction, we are now requesting that all authors identified as 'corresponding author' on published papers create and link their Open Researcher and Contributor Identifier (ORCID) with their account on the Manuscript Tracking System (MTS), prior to acceptance. This applies to primary research papers only. ORCID helps the scientific community achieve unambiguous attribution of all scholarly contributions. You can create and link your ORCID from the home page of the MTS by clicking on 'Modify my Springer Nature account'. For more information please visit [please visit www.springernature.com/orcid](http://www.springernature.com/orcid).

If you wish to submit a suitably revised manuscript we would hope to receive it within 2 months. If you cannot send it within this time, please let us know.

Yours sincerely,

Reviewer Expertise:

Referee #1: Phosphorylation in Plasmodium
Referee #2: [withdrawn after first round]
Referee #3: Plasmodium parasite evolution
Referee #4: Structural Biology, parasite kinases

Reviewer Comments:

Reviewer #1 (Remarks to the Author):

The authors provided a comprehensive and satisfactory response to most comments and requests. They chose to decline to follow some of the reviewer's recommendations, but in these cases the justification they provided was perfectly adequate. I recommend publication of the paper -this will be a hallmark paper in the unfolding story of FIKK characterisation.

Reviewer #3 (Remarks to the Author):

Overall, the paper is nicely improved.

Concerning the evolutionary analysis: The dN/dS analysis taken from the cited paper just shows that nothing really happens in the FIKK family. It might be interesting to see that the two FIKK with the strongest signal of purifying selection are the FIKK4.x - which have known functions. But overall the authors provide a test and describe the evaluation, as indicated in the title.

However, they don't test the idea of geographical differences. There they fall back on the pseudogenisation. I wonder, if sequences should be curated carefully, as in SEA, many genomes have gone through recent bottlenecks and might influence the signal. Also, if the authors want to dive deeper into the pseudogenisation, it might be necessary to see if indels might have changed the open reading frame. On the other side this is not an in-depth evolutionary analysis.

So, the one thing I wonder, if for the following statement in the discussion:

We observe notable differences in pseudogenisation between geographical backgrounds, suggesting that the environment might impact FIKK relevance.

You want to have a test, to see if the difference is statistically significant

Minor:

Should the reference to the figure in line 386 be figure 3D?

Reviewer #4 (Remarks to the Author):

The work by Belda et al is a very nice and comprehensive study focusing on a subgroup of protein kinases that play important roles in Plasmodium and related protozoa.

I shall focus within the limits of my expertise, only to analyze some aspects of the structural biology work and analyses.

I believe some modifications are needed to warrant an accurate model to be reported and deposited in the PDB. The accuracy in processing and refining the crystallographic data and model might also improve the electron density maps (eventually useful to further confirm and/or extend biologically relevant conclusions).

Concerns and comments:

1. I believe the main conclusions concerning the crystallographic (and overall structural) data, are valid.

However, the crystal structure has been processed and refined inaccurately. Materials and Methods details with regards to how these were done are also extremely limited for any reader to understand and/or reproduce.

This has resulted in several significant errors, that once corrected, will probably improve statistics, and perhaps even locally, improve electron density features that may now be erroneous.

After careful analysis of the atomic coordinates and observed/refined structure factors that the authors kindly shared, it seems clear to me that the data were indexed in the wrong space group. I humbly believe this is not a monoclinic crystal with angle beta so close to 90°, it simply is an orthorhombic crystal.

This potential error is what likely explains that the PDB validation report detects a "significant" degree of twinning (see Section 4 within the report, with Xtriage detecting a twinning fraction of ~0.42 following twinning law h,-k,-l).

If instead the authors, for some valid reason –in which case they should clearly report about it in Materials and methods–, truly believe this is a twinned monoclinic crystal, then they should also of course take good consideration of it, to properly refine taking into account the twinning law.

This may all explain the rather high Rfree value and an important Rfree-Rwork gap for this resolution.

Several important geometry issues have also been left uncorrected (such as several cis peptide omega dihedrals for non pre-Pro residues, a pretty high number of bad fitting residues within density or RSRZ outliers, etc etc etc).

Also, several side chains (and sometimes individual atoms within some side chains) that the authors omit, are clearly interpretable, working through the difference Fourier maps: they should be added into the final model and refined.

I suggest the authors to reprocess the data after indexing in SG P212121 (instead of the monoclinic P21 that the authors chose).

And, to invest more careful work in manually rebuilding the atomic model, such that no major errors are left uncorrected. After what, corroborate any biologically relevant conclusions that may be drawn from the structural analysis.

2. Please use pLDDT scores (instead of TM) to quantify the reliability of AlphaFold-predicted structures. For instance, in Supplementary Table 11, and consistently throughout the text.

The Template Modelling (TM) score is generally considered to be relatively insensitive to local inaccuracies (Xu 2010 <https://doi.org/10.1093/bioinformatics/btq066>). Actually, higher than 1 Å rmsd values (for some kinase pairs even >2 Å) as the authors report for the superpositions, indicate that, while definitely reliable in terms of fold (which is actually an expected outcome when it comes to predicting ePKs), they will most certainly include pretty significant variations at the local level.

On the other hand, pLDDT (predicted local distance difference test) scores can be reported as an average value (to replace TM in Supp Table 11), but also convey greatest information when reported as a per residue index. A typical way to do this is to color the AF2-predicted models that you show in main and supplementary figures, with a ramp according to pLDDT (this score is saved in the AF2 models on the B-factor column, so that it is easy to color by this property). Doing so, any regions/loops that would happen to have pLDDTs ≤ 70 (which AlphaFold will classify as being low) should be considered cautiously.

3. Please do not include explicit hydrogen atoms at this resolution, as you have currently done. I guess this might have been the result of an unwanted “default” setting in your refinement program. It is certainly a good practice to add H atoms in riding positions while refining, but not to actually leave them and write them within the final mmCIF file (at 2.8 Å resolution this doesn't make sense, and drives overfitting due to exceedingly high parameters/observables ratio).

4. The loop 194:204 (not included in the crystal model because of lack of electron density), is positioned such that it protrudes from the N-lobe towards the C-lobe of the ePK domain. Even though this loop might be mobile (hence the weak density; and, perhaps low pLDDT scores in AlphaFold predictions?), it will be covering the catalytic site due to its topologic position. It somehow resembles to an analogous N:C-lobe interaction as the one observed in Tyr-kinases like IRK (e.g. pdb 4IBM). Could this loop and/or its characteristics (length, properties) be linked to Tyr vs Ser/Thr specificity?

Following up on this point, and converging with a colleague reviewer's question, even though there is not an identical crevice as in IRK, it's not obvious to me that a deepened cleft can be ruled out. I don't know exactly how does FIKK12 look like (which the authors used in their response to compare with a S/T-specific variant; again what about pLDDT scores for relevant structural components when predictions are used?); but in any case it is very useful for these matters, to look at solvent accessible surface representations (instead of cartoons), getting a better grip of clefts, pockets or even enclosed cavities (which could open up by movement of loops/residues). I actually observe such a cavity in FIKK13, poised to open just beside the catalytic Asp. This cavity is delimited by N257, L252, E426 on one side, and Q401, I429, L378 on the other (and, Leu378, interestingly, is replacing the highly conserved Arg in the catalytic site-containing fragment known as 'HRD', here 'HLD' in FIKK13. Can you please comment (considering of course other S/T-specific ePKs)?

Extended Data Fig 2 does not allow to see all these sequence analyses/comparisons: it would be worth having a clearer and complete MSA of FIKKs as extended data figure. I did also read carefully the response to other reviewers, including an MSA with full length FIKKs: I admit fonts are extremely small there, and of course including the complete sequences makes it more difficult to focus on the ePK domain. Maybe my previous questions are already responded therein, I apologize if that's the case, yet I cannot be clear with the elements at hand.

5. You mention the 'HRD' motif substitution by HLD (as well as the modification of the conserved 'DFG' motif), in the caption to Ext Data Fig 6, but actually consider them to be “minor changes”. I tend to disagree with that. HRD and DFG motifs play key roles, explaining why they are so well conserved among a vast number of ePKs.

6. Please discuss your findings in view of the recent publication by Gizzio et al 2024 (Nat Commun doi:10.1038/s41467-024-50812-0). Are there coincidences or contradictions? Coming back to the DFG motif analysis just referred to above (with a “locked active” configuration in the case of FIKK13), this discussion with an evolutionary context seems particularly useful.

7. Please consider the possibility that the APTgS moieties have been partially/fully hydrolyzed. Electron density seems weaker on the gamma P center; and, please remember that sulfur should be sitting on a Fourier electron density peak at least as large as the phosphorus atom, since they both have approx. same number of electrons. Another evidence that this group has been hydrolysed (at least partially: occupancy can be refined). ATPgS is a slowly hydrolyzable ATP variant; for actual inhibition you may want to prefer using AMP-PCP (a methylene replacing the bridging O between beta and gamma P). Again, any analyses such as these should be made once refinement is properly finished, as phases might improve quite a bit, and then maybe the thio-derivatized gamma phosphate is indeed clearly there at full height, I cannot be sure right now; also, the nucleotides show right now a rather weird discontinuity, which in the end may be corrected. Last but not least, normally ATP and derivatives are complexed with a Mg²⁺ cation: consider this in your refinement (or else, comment).

Version 2:

Reviewer comments:

Reviewer #4

(Remarks to the Author)

I want to thank the authors for their much improved version. Several of my suggestions were considered helpful and either included in the revised version of the ms, or yet were very nicely discussed here within the reviewing process. All in all, I think this is a very nice paper that will make an important contribution in the field of apicomplexan kinase biology, extending the impact more broadly to the community interested in protein kinases in general (given the singularities of this ePK family).

Concerning disagreement points, I have been convinced in most of them based upon the authors' thoughtful and well explained arguments.

There are three points that I'm afraid I cannot agree upon. Given that the most important of these refers to the quality of the refinement and, what I still think was an involuntary mistake in the space group assignment, I also attach some files that I believe will greatly facilitate the correction of these issues/.

1- Concerning my request for crystallographic methods to be better explained.

The authors replied that they think it is OK what they presented.

My only job as a reviewer is just to indicate that, going through the Methods section ("Crystallisation of PfFIKK13149-561_D379N with Nb2G9, Nb9F10 and ATPγS") as it's written, one cannot actually reproduce what was done. In the revised version, the authors added the programs they used to do MR and data processing. Yet nothing about why they chose P21 (even though orthorhombic was indeed an option as they explain now), or how they proceeded to do the refinement and validation. I am not asking for specialized details, just the basics of the software AND protocols you used to process the raw diffraction and then to refine your proposed model.

In other words, it is not up to me to establish the editorial requirements. Just to indicate whether the results can be reproduced, that's all I did.

2- Concerning my strong suggestion to reprocess the data in space group 19, to see whether that would generate a more accurate refined model.

The authors disagreed and decided not to do it. They argue that the data are indeed monoclinic. The reason they put forward is that in P212121, refinement would stall with unacceptable statistics (probably referring to too high R factors, I would guess). I thank the authors for having shared data and coordinates. The best way to reply from my side is to attach a properly reprocessed data set (using the program Zanuda from CCP4, as a means to quantitatively test all groups compatible with the Laue assignment that derives from the unmerged intensities; Zanuda also performs a quick Refmac refinement in all subgroups that produce low-enough Rmerge statistics). I also did a very quick refinement with Phenix afterwards, with the reprocessed data in SG 19 (and maintaining the authors' choice of TLS refinement, although with slightly different TLS groups chosen: this is not critical anyhow, different TLS group choosing have marginal effect): quite standard protocol: xyz refinement, B factors by atom, local NCS restraints, automatic choosing of TLS groups, optimization of geometry/Rfacs scaling.

Rfree and Rwork, as well as model stereochemistry, were all significantly better in SG P212121, than the ones reported by the authors. This fact demonstrates the crystal is indeed orthorhombic.

Again, please do not consider my job as trying to raise obstacles. My intention is to serve as a double-check, so that models, as accurate as possible (using state of the art software) are the ones being published and deposited in the PDB.

The authors state "The twinning test mentioned is only one of many and not a good indicator when a crystallographic symmetry operation has been replaced by non-crystallographic symmetry." This is a mistake, or the authors misinterpreted what I was suggesting.

If I raised a potential twinning as the reason for their choosing a lower symmetry space group (as the PDB validation report actually brings in, signs of significant twinning: which is absolutely expected given the higher symmetry space group that we now know is true), it was as a means of being open to such a possibility.

If the higher symmetry I originally suggested to explore for, would be the consequence of NCS, it would never allow for smooth convergent refinement with good R factor residuals and geometrical quality. I don't know why the authors' refinement protocols stalled at unsatisfactory levels (they don't describe any refinement protocols whatsoever).

As a further proof, if NCS were to explain the additional 2-fold axes (just by chance falling into a perpendicular direction to the unique 2-fold), a self-rotation analysis of the data would not bring up 2-fold peaks that have the same height as the crystallographic 2-fold that the authors refined with:

I attach self-rotation plots and the beginning of the table with the signal/noise values (calculated with Molrep). The first plot (top right) represents the $\chi=180^\circ$ section of the rotation function, from the data processed in P21: the same level peaks are readily seen at the equator along the crystallographic a axis (parallel to x according to Molrep's convention), and of course a third 2-fold appears at the center perpendicular to the plane of the paper. This is pathognomonic of a 222 point group. If it were a monoclinic, non-crystallographic symmetry axes at 90° of it would be weaker in intensity (by the way, some 2-fold NCS axes can indeed be seen, but they have <6-fold less signal/noise compared to the crystallographic ones in this case).

All this analysis has nothing to do with intensities' distribution analyses (the so called "twinning tests"), but rather consequences

of intrinsic symmetry of the crystal.

Again, I do not believe there are major modifications to the biological conclusions because of this mistake. Yet, it is important that the authors, who have the biological expertise, double-check this with a properly solved and refined structure. Having said that, and not being myself a specialist in apicomplexa kinases, the re-refined model tends to indicate that the gamma phosphate is not present on the bound nucleotides (I am aware that the authors used a normally non-hydrolysable variant AGS, yet...perhaps slowly hydrolysable as they agree). The sulfur atoms of their derivative are excellent markers, since S atoms scatter X-rays significantly more strongly than oxygen atoms, so that S on g-P should give a larger Fourier peak if it were there.

- I would also add the Mg²⁺ cation complexed to the phosphates, it appears visible in density. Check links with neighbor atoms as I did only an automatic assignment with phenix.metal_coordination.
- Many side chains became visible in the maps, and I only did a very quick refinement.
- Please check whether any additional information of biological relevance can be extracted.

Last: the authors were right that there are no cis peptides, apologies for that mistake of mine (Coot 0.9.x now signals "cis" peptides on each first residue after a gap...which is certainly a bug due to the way it calculates it)

3- Concerning my advice to avoid depositing hydrogen atoms at this resolution.

The authors did not agree. I must insist.

What I absolutely agree with, is that, as the authors sensibly say, adding H atoms at riding positions is a very good practice (especially at lower resolutions). But that is done during refinement (as well as at validation time, particularly to calculate clash scores). You do not want to keep these atoms explicitly in the final pdb file. The authors actually avoided including side chains altogether if they are not seen in density: how would one keep H atoms? This leads to misuse afterwards by unaware PDB users.

The authors probably used an option "by default" of some sorts, that keep the H atoms in their riding positions: it is just not a good idea at this resolution.

Decision Letter:

28th November 2024

Dear Moritz,

Thank you for your patience while your manuscript "Evolution and inhibition of the FIKK effector kinase family in *P. falciparum*" was under peer-review at Nature Microbiology. It has now been seen again by referee #4, whose expertise and comments you will find at the of this email. You will see from their comments below that while they find your work of interest, some important points are raised. We are very interested in the possibility of publishing your study in Nature Microbiology, but would like to consider your response to these concerns in the form of a revised manuscript before we make a final decision on publication.

In particular, you will see that the referee has some ongoing concerns about the structure solution and refinement, and that they mention three points that will need to be addressed. The referee also provided some additional files which I am attaching to this message. The rest of the referees' reports are clear and the remaining issues should be straightforward to address. I should note that With this additional revision, we want to ensure that the data are as accurate as possible prior to publication, but please do let us know if you have any concerns. We do not feel the referee had opposing views but asks for best practice which should improve the accuracy of data reporting and analysis.

If you have not done so already please begin to revise your manuscript so that it conforms to our Article format instructions at <http://www.nature.com/nmicrobiol/info/final-submission/>

The usual length limit for a Nature Microbiology Article is six display items (figures or tables) and 3,000 words. We have some flexibility, and can allow a revised manuscript at 3,500 words, but please consider this a firm upper limit. There is a trade-off of ~250 words per display item, so if you need more space, you could move a Figure or Table to Supplementary Information.

Some reduction could be achieved by focusing any introductory material and moving it to the start of your opening 'bold' paragraph, whose function is to outline the background to your work, describe in a sentence your new observations, and explain your main conclusions. The discussion should also be limited. Methods should be described in a separate section following the discussion, we do not place a word limit on Methods.

Nature Microbiology titles should give a sense of the main new findings of a manuscript, and should not contain punctuation. Please keep in mind that we strongly discourage active verbs in titles, and that they should ideally fit within 90 characters each (including spaces).

We strongly support public availability of data. Please place the data used in your paper into a public data repository, if one

exists, or alternatively, present the data as Source Data or Supplementary Information. If data can only be shared on request, please explain why in your Data Availability Statement, and also in the correspondence with your editor. For some data types, deposition in a public repository is mandatory - more information on our data deposition policies and available repositories can be found at <https://www.nature.com/nature-research/editorial-policies/reporting-standards#availability-of-data>.

Please include a data availability statement as a separate section after Methods but before references, under the heading "Data Availability". This section should inform readers about the availability of the data used to support the conclusions of your study. This information includes accession codes to public repositories (data banks for protein, DNA or RNA sequences, microarray, proteomics data etc...), references to source data published alongside the paper, unique identifiers such as URLs to data repository entries, or data set DOIs, and any other statement about data availability. At a minimum, you should include the following statement: "The data that support the findings of this study are available from the corresponding author upon request", mentioning any restrictions on availability. If DOIs are provided, we also strongly encourage including these in the Reference list (authors, title, publisher (repository name), identifier, year). For more guidance on how to write this section please see: <http://www.nature.com/authors/policies/data/data-availability-statements-data-citations.pdf>

To improve the accessibility of your paper to readers from other research areas, please pay particular attention to the wording of the paper's opening bold paragraph, which serves both as an introduction and as a brief, non-technical summary in about 150 words. If, however, you require one or two extra sentences to explain your work clearly, please include them even if the paragraph is over-length as a result. The opening paragraph should not contain references. Because scientists from other sub-disciplines will be interested in your results and their implications, it is important to explain essential but specialised terms concisely. We suggest you show your summary paragraph to colleagues in other fields to uncover any problematic concepts.

If your paper is accepted for publication, we will edit your display items electronically so they conform to our house style and will reproduce clearly in print. If necessary, we will re-size figures to fit single or double column width. If your figures contain several parts, the parts should form a neat rectangle when assembled. Choosing the right electronic format at this stage will speed up the processing of your paper and give the best possible results in print. We would like the figures to be supplied as vector files - EPS, PDF, AI or postscript (PS) file formats (not raster or bitmap files), preferably generated with vector-graphics software (Adobe Illustrator for example). Please try to ensure that all figures are non-flattened and fully editable. All images should be at least 300 dpi resolution (when figures are scaled to approximately the size that they are to be printed at) and in RGB colour format. Please do not submit Jpeg or flattened TIFF files. Please see also 'Guidelines for Electronic Submission of Figures' at the end of this letter for further detail.

Figure legends must provide a brief description of the figure and the symbols used, within 350 words, including definitions of any error bars employed in the figures.

When submitting the revised version of your manuscript, please pay close attention to our [href="https://www.nature.com/nature-research/editorial-policies/image-integrity">Digital Image Integrity Guidelines.](https://www.nature.com/nature-research/editorial-policies/image-integrity) and to the following points below:

Please include a statement before the acknowledgements naming the author to whom correspondence and requests for materials should be addressed.

Finally, we require authors to include a statement of their individual contributions to the paper -- such as experimental work, project planning, data analysis, etc. -- immediately after the acknowledgements. The statement should be short, and refer to authors by their initials. For details please see the Authorship section of our joint Editorial policies at http://www.nature.com/authors/editorial_policies/authorship.html

* include a point-by-point response to any editorial suggestions and to our referees. Please include your response to the editorial suggestions in your cover letter, and please upload your response to the referees as a separate document.

* ensure it complies with our format requirements for Letters as set out in our guide to authors at www.nature.com/nmicrobiol/info/gta/

* state in a cover note the length of the text, methods and legends; the number of references; number and estimated final size of figures and tables

* resubmit electronically if possible using the link below to access your home page:

Link Redacted

*This url links to your confidential homepage and associated information about manuscripts you may have submitted or be

reviewing for us. If you wish to forward this e-mail to co-authors, please delete this link to your homepage first.

Please ensure that all correspondence is marked with your Nature Microbiology reference number in the subject line.

Nature Microbiology is committed to improving transparency in authorship. As part of our efforts in this direction, we are now requesting that all authors identified as 'corresponding author' on published papers create and link their Open Researcher and Contributor Identifier (ORCID) with their account on the Manuscript Tracking System (MTS), prior to acceptance. This applies to primary research papers only. ORCID helps the scientific community achieve unambiguous attribution of all scholarly contributions. You can create and link your ORCID from the home page of the MTS by clicking on 'Modify my Springer Nature account'. For more information please visit www.springernature.com/orcid.

We hope to receive your revised paper within three weeks. If you cannot send it within this time, please let us know.

Yours sincerely,

Reviewer Expertise:

Referee #4: Structural Biology, parasite kinases

Reviewers Comments:

Reviewer #4 (Remarks to the Author):

I want to thank the authors for their much improved version. Several of my suggestions were considered helpful and either included in the revised version of the ms, or yet were very nicely discussed here within the reviewing process. All in all, I think this is a very nice paper that will make an important contribution in the field of apicomplexan kinase biology, extending the impact more broadly to the community interested in protein kinases in general (given the singularities of this ePK family).

Concerning disagreement points, I have been convinced in most of them based upon the authors' thoughtful and well explained arguments.

There are three points that I'm afraid I cannot agree upon. Given that the most important of these refers to the quality of the refinement and, what I still think was an involuntary mistake in the space group assignment, I also attach some files that I believe will greatly facilitate the correction of these issues/.

1- Concerning my request for crystallographic methods to be better explained.

The authors replied that they think it is OK what they presented.

My only job as a reviewer is just to indicate that, going through the Methods section ("Crystallisation of PfFIKK13149-561_D379N with Nb2G9, Nb9F10 and ATP γ S") as it's written, one cannot actually reproduce what was done. In the revised version, the authors added the programs they used to do MR and data processing. Yet nothing about why they chose P21 (even though orthorhombic was indeed an option as they explain now), or how they proceeded to do the refinement and validation. I am not asking for specialized details, just the basics of the software AND protocols you used to process the raw diffraction and then to refine your proposed model.

In other words, it is not up to me to establish the editorial requirements. Just to indicate whether the results can be reproduced, that's all I did.

2- Concerning my strong suggestion to reprocess the data in space group 19, to see whether that would generate a more accurate refined model.

The authors disagreed and decided not to do it. They argue that the data are indeed monoclinic. The reason they put forward is that in P212121, refinement would stall with unacceptable statistics (probably referring to too high R factors, I would guess).

I thank the authors for having shared data and coordinates. The best way to reply from my side is to attach a properly reprocessed data set (using the program Zanuda from CCP4, as a means to quantitatively test all groups compatible with the Laue assignment that derives from the unmerged intensities; Zanuda also performs a quick Refmac refinement in all subgroups that produce low-enough Rmerge statistics). I also did a very quick refinement with Phenix afterwards, with the reprocessed data in SG 19 (and maintaining the authors' choice of TLS refinement, although with slightly different TLS groups chosen: this is not critical anyhow, different TLS group choosing have marginal effect): quite standard protocol: xyz refinement, B factors by atom, local NCS restraints, automatic choosing of TLS groups, optimization of geometry/Rfacs scaling.

Rfree and Rwork, as well as model stereochemistry, were all significantly better in SG P212121, than the ones reported by the authors. This fact demonstrates the crystal is indeed orthorhombic.

Again, please do not consider my job as trying to raise obstacles. My intention is to serve as a double-check, so that models, as accurate as possible (using state of the art software) are the ones being published and deposited in the PDB.

The authors state "The twinning test mentioned is only one of many and not a good indicator when a crystallographic symmetry operation has been replaced by non-crystallographic symmetry." This is a mistake, or the authors misinterpreted what I was suggesting.

If I raised a potential twinning as the reason for their choosing a lower symmetry space group (as the PDB validation report actually brings in, signs of significant twinning: which is absolutely expected given the higher symmetry space group that we now know is true), it was as a means of being open to such a possibility.

If the higher symmetry I originally suggested to explore for, would be the consequence of NCS, it would never allow for smooth convergent refinement with good R factor residuals and geometrical quality. I don't know why the authors' refinement protocols stalled at unsatisfactory levels (they don't describe any refinement protocols whatsoever).

As a further proof, if NCS were to explain the additional 2-fold axes (just by chance falling into a perpendicular direction to the unique 2-fold), a self-rotation analysis of the data would not bring up 2-fold peaks that have the same height as the crystallographic 2-fold that the authors refined with:

I attach self-rotation plots and the beginning of the table with the signal/noise values (calculated with Molrep). The first plot (top right) represents the $\chi=180^\circ$ section of the rotation function, from the data processed in P21: the same level peaks are readily seen at the equator along the crystallographic a axis (parallel to x according to Molrep's convention), and of course a third 2-fold appears at the center perpendicular to the plane of the paper. This is pathognomonic of a 222 point group. If it were a monoclinic, non-crystallographic symmetry axes at 90° of it would be weaker in intensity (by the way, some 2-fold NCS axes can indeed be seen, but they have <6-fold less signal/noise compared to the crystallographic ones in this case).

All this analysis has nothing to do with intensities' distribution analyses (the so called "twinning tests"), but rather consequences of intrinsic symmetry of the crystal.

Again, I do not believe there are major modifications to the biological conclusions because of this mistake. Yet, it is important that the authors, who have the biological expertise, double-check this with a properly solved and refined structure.

Having said that, and not being myself a specialist in apicomplexa kinases, the re-refined model tends to indicate that the gamma phosphate is not present on the bound nucleotides (I am aware that the authors used a normally non-hydrolysable variant AGS, yet...perhaps slowly hydrolysable as they agree). The sulfur atoms of their derivative are excellent markers, since S atoms scatter X-rays significantly more strongly than oxygen atoms, so that S on g-P should give a larger Fourier peak if it were there.

- I would also add the Mg²⁺ cation complexed to the phosphates, it appears visible in density. Check links with neighbor atoms as I did only an automatic assignment with phenix.metal_coordination.

- Many side chains became visible in the maps, and I only did a very quick refinement.

- Please check whether any additional information of biological relevance can be extracted.

Last: the authors were right that there are no cis peptides, apologies for that mistake of mine (Coot 0.9.x now signals "cis" peptides on each first residue after a gap...which is certainly a bug due to the way it calculates it)

3- Concerning my advice to avoid depositing hydrogen atoms at this resolution.

The authors did not agree. I must insist.

What I absolutely agree with, is that, as the authors sensibly say, adding H atoms at riding positions is a very good practice (especially at lower resolutions). But that is done during refinement (as well as at validation time, particularly to calculate clash scores). You do not want to keep these atoms explicitly in the final pdb file. The authors actually avoided including side chains altogether if they are not seen in density: how would one keep H atoms? This leads to misuse afterwards by unaware PDB users.

The authors probably used an option "by default" of some sorts, that keep the H atoms in their riding positions: it is just not a good idea at this resolution.

Version 3:

Decision Letter:

Our ref: NMICROBIOL-24030843C

12th February 2025

Dear Moritz,

Thank you for submitting your revised manuscript "Evolution and inhibition of the FIKK effector kinase family in *P. falciparum*" (NMICROBIOL-24030843C). We have now editorially assessed the revised manuscript and find that the paper has improved in revision, and therefore we'll be happy in principle to publish it in Nature Microbiology, pending minor revisions to comply with our editorial and formatting guidelines.

We are now performing detailed checks on your paper and will send you a checklist detailing our editorial and formatting

requirements in about two weeks. Please do not upload the final materials and make any revisions until you receive this additional information from us.

Thank you again for your interest in Nature Microbiology. Please do not hesitate to contact me if you have any questions.

Best wishes,

Version 4:

Decision Letter:

14th April 2025

Dear Moritz,

I am pleased to accept your Article "The fast-evolving FIKK kinase family of *Plasmodium falciparum* can be inhibited by a single compound" for publication in Nature Microbiology. Thank you for having chosen to submit your work to us and many congratulations.

Authors may need to take specific actions to achieve <https://www.springernature.com/gp/open-research/funding/policy-compliance-faqs> compliance with funder and institutional open access mandates. If your research is supported by a funder that requires immediate open access (e.g. according to <https://www.springernature.com/gp/open-research/plan-s-compliance>) Plan S principles) then you should select the gold OA route, and we will direct you to the compliant route where possible. For authors selecting the subscription publication route, the journal's standard licensing terms will need to be accepted, including <https://www.nature.com/nature-portfolio/editorial-policies/self-archiving-and-license-to-publish> self-archiving policies. Those licensing terms will supersede any other terms that the author or any third party may assert apply to any version of the manuscript.

We welcome the submission of potential cover material (including a short caption of around 40 words) related to your manuscript; suggestions should be sent to Nature Microbiology as electronic files (the image should be 300 dpi at 210 x 297 mm in either TIFF or JPEG format). Please note that such pictures should be selected more for their aesthetic appeal than for their scientific content, and that colour images work better than black and white or grayscale images. Please do not try to design a cover with the Nature Microbiology logo etc., and please do not submit composites of images related to your work. I am sure you

will understand that we cannot make any promise as to whether any of your suggestions might be selected for the cover of the journal.

With kind regards,

Emily

Emily White, PhD
Chief Editor
Nature Microbiology

4 Crinan Street, London, UK, N1 9XW
+44 207 418 5601
emily.white@nature.com

orcid.org/0000-0002-2314-5718

P.S. Click on the following link if you would like to recommend Nature Microbiology to your librarian
<http://www.nature.com/subscriptions/recommend.html#forms>

** Visit the Springer Nature Editorial and Publishing website at http://editorial-jobs.springernature.com?utm_source=ejP_NMicro_email&utm_medium=ejP_NMicro_email&utm_campaign=ejp_NMicro for more information about our career opportunities. If you have any questions please click [here](mailto:editorial.publishing.jobs@springernature.com).**

Open Access This Peer Review File is licensed under a Creative Commons Attribution 4.0 International License, which permits use, sharing, adaptation, distribution and reproduction in any medium or format, as long as you give appropriate credit to the original author(s) and the source, provide a link to the Creative Commons license, and indicate if changes were made. In cases where reviewers are anonymous, credit should be given to 'Anonymous Referee' and the source. The images or other third party material in this Peer Review File are included in the article's Creative Commons license, unless indicated otherwise in a credit line to the material. If material is not included in the article's Creative Commons license and your

intended use is not permitted by statutory regulation or exceeds the permitted use, you will need to obtain permission directly from the copyright holder.

We thank all reviewers for their helpful comments.

Reviewer Expertise:

Referee #1: Phosphorylation in Plasmodium

Referee #2: P. falciparum structural biology

Referee #3: Plasmodium parasite evolution

Reviewers Comments:

Reviewer #1 (Remarks to the Author):

This is a comprehensive study of the substrate preferences of the FIKKs, an intriguing family of kinases that is specific to the Laverania sub-taxon of malaria parasites. The FIKK family and its recent radiation and expansion in Laverania (to the exclusion of other taxons of malaria parasites) were discovered 20 years ago, but the important question of substrate specificity of its members has not been addressed comprehensively. Here, the authors implement a peptide array approach and of an almost complete panel recombinant FIKKs (only two enzymes out of the 20 P. falciparum FIKKs were refractory to heterologous expression) to fill this important gap. This yielded a convincing picture of distinct substrate preferences, including for FIKKs that share subcellular localisation. Surprisingly, one of the enzymes has exclusive tyrosine kinase activity.

The crystal structure of an FIKK allows the author to map to a specific loop in the catalytic domain the residues that mediate substrate preference. This is another unexpected result, as substrate binding in typical kinases is mediated by specific pockets (mostly located in the in the C-terminal lobe) of the kinase.

In addition to substrate preferences, that authors examined thousands of field isolates to detect the prevalence of pseudogenes in the FIKK family. This led to very interesting insights on the essentiality/relaxed selection of some of the FIKKs (e.g. FIKK14) and on the ongoing evolutionary history of this family.

Finally, the authors present data relating to a screen for ATP competitive inhibitors, and show that some molecules display broad-spectrum inhibitory activity (with respect to the FIKK family).

Overall, this is a well-conducted and comprehensive study that represents a significant contribution to our understanding of the evolutionary history and function of the FIKK family. Beyond that, the study will be of great interest to the fields of structural kinatology and of molecular evolution.

This reviewer does not have any major issue with the manuscript, but the authors may want to consider the following points:

Turbo ID studies: TurboID is a powerful method to assign the precise molecular environment of a give protein, and the data that FIKK4.1 and FIKK4.2 overlap but also differ in this respect are compelling. It would be interesting to give some more information in Supplementary Table 6, for example indicate the full name of the interactor (when available).

Full name of the interactors has been added to what is now Supplementary Table 7.

I suppose “N” in this Table means “non-downregulated”? please clarify in the legend.

This has been corrected with the sentence “‘N’ refers to proteins with no change in phosphorylation status upon FIKK4.1 or FIKK4.2 KO, as described in Davies et al., 2020.” now in the legend of Supplementary Table 7.

It would be beneficial to somehow link the information in this table S6 to that in Supplementary Table 13.

The sentence “Interactors are taken from proximity labelling data provided in Supplementary Table 14” has been added to the legend of Supplementary Table 7.

Substrate preference: It seems from Figure 3 that the peptide array did not include pre-phosphorylated residues. It may be of interest to investigate if FIKKs have a preference for motifs that contain pre-phosphorylated residues (like GSK3 or CK1, for example, which use primed phosphorylation as an important motif element). This might be of great interest with respect to possible hierarchical phosphorylation by FIKKs that share the same localisation. This is not required for this paper, but a line or two in the discussion might interest the reader.

This is a good point- we added the following short note in the discussion.

“Two recent studies highlighted the importance of pre-phosphorylated residues in the preferred phosphorylation motif of human kinases^{47,73}. The OPAL libraries used in this study do not include pre-phosphorylated residues but it will be interesting to investigate possible hierarchical phosphorylation by FIKKs that share the same localisation in the future.”

Structural studies: “the determinants identified here map to kinase loop regions”. This is a bit vague --please indicate the boundaries of this loop(s), and label them on the 3D structure.

We have now labelled the loop regions on Figure 5b, as follows:

K212 is found on a large loop on the N-terminal lobe. This residue lies between the $\beta 1$ and $\beta 2$ regions labelled on Extended data Fig. 10, and so we annotate the loop here as ' $\beta 1$ - $\beta 2$ loop'. K263 is found on the loop between the αB and αC helices. Secondary structure regions are referred to (for reference) in Extended Data Fig. 6 and Extended Data Fig. 10.

In the figure legend, we have labelled the boundaries of these regions. The substrate-facing region of the N-lobe loop lies between residues 202 to 217 (inclusive). The αB - αC loop is between residues 257 to 267 (inclusive).

With respect to the Tyr specificity, the authors say "We were unable to predict the basis for the tyrosine specificity of FIKK13". However, the peptide used for modelling in Fig. 6b has a serine residue. One of the determinants for Tyr specificity in other cases is the depth of the cleft; for example, the Insulin Receptor kinase IRK (a tyrosine kinase) does not phosphorylate serines because the serine side-chain is too short to reach the catalytic site; unlike Ser, Tyr extends far enough into the catalytic cleft to be efficiently phosphorylated (see Figure 2 of PMID:17585314). It would be of interest to model a Tyr-carrying peptide in FIKK13, and in serine-directed FIKKs, and to determine if the size of the Tyr/Ser side chain can be part of the determinant for the discrimination between Tryr-targeted (FIKK13) and Ser-targeted FIKKs (all the other FIKKs).

Yes indeed when comparing kinases of the TyrK group with canonical Ser/Thr kinases, the depth of the cleft is an important discriminant. We checked to see if this

would also be the case for the FIKKs by performing a superposition between FIKK13 and a S/T-specific FIKK (FIKK12):

FIKK12 is in grey whereas FIKK13 is in light blue. The catalytic Asp is in stick format. As can be seen, the substrate-binding grooves overlap very strongly between the two kinases. Therefore, differences in the depth of the catalytic cleft is an unlikely explanation in this case.

A sentence has now been added to the manuscript

As suggested, we also revisited the kinase-peptide docking analysis. Figure 5b represents an S peptide bound to the S/T kinase FIKK12. We tried mutating S->Y and then repeating the analysis, with the following results:

The Y peptide binds in an unusual, non-linear conformation but still contacts the K212 and K263 residues that we experimentally validated. It is worthwhile to note

that in these conditions the Y is “forced” to bind to the catalytic aspartate residue in the kinase active site. While this works well for the S-peptide which binds in a “normal” linear manner, the Y-containing peptide is forced into a conformation which is unlikely to be adopted under normal conditions.

We next attempted to model a Y peptide and an S peptide into the substrate-binding groove of FIKK13 (which has Y specificity). We should note that there is no known substrate peptide for FIKK13 and so to address this question we attempted modelling with an idealised (but unnatural) EEDEYLKKD peptide. A model was built with Y as the phosphoacceptor and then with S as the phosphoacceptor (i.e. EEDESLKKD). However, both models have a similar docking score for the method used (Haddock 2.4).

In conclusion, using these approaches, unfortunately, we still cannot determine why FIKK13 kinase phosphorylates Y peptides instead of S peptides. A natural Y-containing peptide - FIKK13 co-structure is likely needed to clarify the structural basis of the interaction.

We have not amended our original statement that we were unable to identify the basis for Tyrosine phosphorylation of FIKK13 and have not included the results shown above in the manuscript because of word limits.

Finally, it would be of interest to compare the data for the FIKK13 structure with those from other published FIKKs (e.g. PMID:

The reviewer missed to add the PMID (blank space). As our FIKK13 structure is the first published FIKK structure to our knowledge, we have no comparison.

Drug discovery: It would be good to display the structure of the best compounds “from different chemical series” (line 562) in an additional panel of Fig. 7.

We agree with the reviewer and have modified Fig. 6 (previously 7) accordingly.

To address the problem “that the compounds engage one or multiple kinases other than the FIKKs” in live parasite assays, it would be good to mention if these compounds (or related compounds from the same series) have been profiled on the human kinome (data on Plasmodium non-FIKK kinases are likely not available), and how promiscuous they may be.

Compounds in the PKIS set originate from human kinase drug discovery projects.

As ATP competitive inhibitors, these compounds will have varying levels of off-target activity on human kinases. Unfortunately, the compounds used in this study have not been profiled in concentration-response against a common panel of kinases which makes it difficult to make extensive comparisons of off-targets between the compounds. We recognise there are important limitations in working with probe compounds and this was a key consideration in adopting GSK3184025A as a weakly active control compound from the same chemical series as GW779439X.

A comparison with another published FIKK inhibitor (target the *Cryptosporidium* FIKK, PMID: 28162900) would be of interest --could this be modelled into the Plasmodium FIKK structures?

We thank the reviewer for highlighting the FIKK inhibitor series published by Osman *et al* for *cryptosporidium*. These compounds were discovered by screening a

targeted collection of approximately 2500 known (human) kinase inhibitors which incorporated the PKIS set used in our study. In the Osman paper, the two aminothiazoles hits **4a** (GW780159X) and **4j** (GW785804X) are constituents of the PKIS compound set and we can confirm that these same two compounds were present in our PKIS screen against *Pf*FIKK. Compound **4a** showed no inhibition and compound **4j** showed only 15% inhibition at 10 μ M (well below the 75% threshold we designated for hits). In the absence of comparable activity between *Cp*FIKK and *Pf*FIKK enzymes, we feel the proposed dockings would offer limited value.

The fact that all FIKK have a small gatekeeper residue (line 548) opens the possibility to conduct reverse genetics/chemical genetics studies to investigate the exact role of each FIKK individually, by replacing the gatekeeper with a large residue that can prevent inhibition by specific compounds (as had been done with *Pf*PKG, for example). This is of course not required for this paper, but a mention of this approach in the Discussion might be of interest to the reader.

This is indeed an interesting point- once we have identified an inhibitor that has less/ not off-target effects. In the absence of such an inhibitor, we feel the discussion in this manuscript is maybe not a priority given the world limits.

Minor/editorial points

Line 59-61:” The fikk genes are conserved in syntenic loci across the *Laverania*, arguing for a rapid expansion controlling important functions in host cell remodelling and pathogenesis “. This is misleading. One cannot predict function from chromosomal location.

This sentence has been removed. The sentence is now “~One million years ago, a single FIKK kinase conserved across *Plasmodium* species gained an export element and rapidly expanded into a family of ~20 atypical FIKK kinases exported into the host cell across the *Laverania* subgenus”

Line 103: “6 *Plasmodium* species known to infect humans” (there may be more than 6).

This has been corrected in the manuscript

Line 120: explain the name “FIKK” at the first occurrence.

This has been corrected

Line 126 “but no other human-infecting species”. This is true for all species, not only those infecting humans.

This has been corrected to “Only *Plasmodium* species from the *Laverania* subgenus (includes *P. falciparum* and *Plasmodium* species infecting great apes) possess the expanded FIKK kinase family”

Line 185: A bit confusing. “FIKKs that may have lost functions during infection of their human host” ?

Corrected to: “FIKKs that may have lost functions in the human host”

Line 298: “*P. falciparum* kinases phosphorylate S and T residues within acidic and basic motifs. Phosphorylated Y residues and proline-directed motifs are rarely found”. This may be because a bias to the most abundant/detectable sites. The

plasmodium kinome includes many enzymes from the CMGC group (notably CDK/MAPK) which are proline-directed.

While CMGC group kinases are indeed often proline directed, this has not been experimentally validated for Pf CMGC kinases to our knowledge. In phosphoproteome analysis of Pf infected RBCs, there is a significant reduction of prolines in phosphorylation motifs compared to human, or Toxoplasma phosphoproteome studies that were generated using the same workflows (Treeck et al., 2011, CHM). We therefore believe the statement remains true.

Line 300: “predicted tyrosine kinases are lacking from the genome” The proper way to say this is that the Plasmodium kinome does not include enzyme from the TyrK group.

This has now been corrected

Line 344” : “...the rapid evolution of this relatively young protein family likely due to selection pressure to subvert the host machinery” I do not follow the argument about subverting the host machinery. Please develop.

Changed to: ...the rapid evolution of this relatively young protein family likely due to selection pressure to subvert host cell functions specific to the great ape lineage of hosts by *Laverania Plasmodium* species.

Reviewer #2 (Remarks to the Author):

Summary: In this study, the authors have used a combination of structural characterization and AlphaFold2 predictions and mutation studies to first identify and the confirm likely structural determinants of Plasmodial FIKK family kinase specificity. Using this information, the authors then demonstrated feasibility of developing pan-specific inhibitors of this kinase family unique to Apicomplexan parasites. This thorough and impactful work in a timely area presents a promising avenue for new anti-malarial treatments with reduced potential for development of resistance parasites.

Minor comments

1. It would be helpful if the AlphaFold2 predicted structures could be overlaid and included in a supplementary figure.

We were not sure whether the reviewer meant to overlay the full FIKK kinase sequences, or just the kinase domains. We felt that overlays as a supplementary figure would not be highly informative, as details would likely not be easy to see. Instead, we calculated RMSD and TM-score values for each kinase superimposed to the non-exported FIKK8 (as a reference) and provided data in Supplementary Table 11. We feel that readers with a keen interest in this question can easily overlay the structures and look at the structure from all angles.

Reviewer #3 (Remarks to the Author):

In this paper, the authors describe the evolution and function of the expansion of the FIKK kinase gene family in *Laverania*. First, they focus on its evolution, then perform molecular docking and mutational analysis—finding that FIKK13 is a tyrosine kinase not yet reported in *Laverania*. In addition to other analyses, like co-localisation and

determining the structures of a FIKK kinase, they propose an inhibitor.

Due to the sheer number of experiments performed, it felt very dense in places. However, I can imagine that the readers of Nature Microbiology will. However, I enjoyed how the discussion pulled all the different aspects of the study together.

My main criticism:

Why don't the authors apply a test for purifying selection? What are the dn/ds ratios in the different FIKK clusters? As evolution is a major point of the paper (including the title), is to use the vast MalariaGen datasets, combined with the Laverania to use more rigorous tests, with a P-value.

Following the suggestion from the reviewer, we calculated dN/dS ratios across *Plasmodium* species using the phylogenetic tree presented in Figure 4b (N=131 sequences). Our analysis indicates a global dN/dS (across branches and sites) of 0.251, indicating strong purifying selection on FIKK kinases.

When the reviewer refers to MalariaGen datasets, we assume that they are alluding to dN/dS ratios that could be calculated from population genome data. Fortunately, dN/dS ratios were calculated recently from *P. falciparum* field isolates in the following paper: PMID 37750720. While the values were not included in the original manuscript, the authors kindly shared them with us. All FIKK dN/dS ratios calculated are <1 (range 0.31–0.94, mean=0.54) and indicate purifying selection. The only caveat of the MalariaGen datasets is that the pseudokinases FIKK7.2, FIKK14 and the active FIKK13- which was only recently annotated are not called. Therefore, we don't have dN/dS ratios for these. However, for FIKK7.2 and 14 we already know that pseudogenisation occurs in the field.

Finally, we performed a dN/dS branch test to determine changes in dN/dS ratios between the FIKK8 clade (non-exported) and all other FIKKs in the phylogenetic tree. We calculate a dN/dS of 0.0867 for the FIKK8 clade (non-exported kinases), and dN/dS of 0.342 for the other Laverania FIKKs (exported kinases). The difference is highly significant ($p=3.05 \times 10^{-135}$, $df=1$, likelihood-ratio test) and indicates a relaxation of purifying selection following gene duplication from the FIKK8 ancestor.

I understand that the FIKK evolved, but how well do they align? I can see from Extended Fig 3 and 15 that there are very conserved region, also Fig 3a shows the conservation on the end. For the Extended figure 1 tree: how many aa were used for the alignments, like 1000? What is the unit (0.1) of the branch length?

To address the first question, we show below the full alignment of all 19 *P. falciparum* kinases sequences. The first image represents sequences outside the kinase domain whereas the second image represents the kinase domain.

As can be seen, sequences outside the kinase domain align very poorly with the exception of a small number of alignment positions. Conversely, the kinase domain is strongly conserved with the exception of a large insertion in *P. falciparum* FIKK4.2.

The final alignment used for the generation of the phylogenetic tree (Figure 3) is a trimmed alignment where positions with more than 20% gap sequences were removed, resulting in 525 alignment positions. Across the *P. falciparum* FIKKs, this corresponds to 9834 non-gapped amino acids in total.

The scale bar in Extended Data Fig. 1 tree represents the number of substitutions per site i.e. two sequences separated by this distance have diverged by 0.1 substitutions per site on average. It was produced by the `geom_treescale()` function in `ggtree`.

This has now been indicated in Extended Data Fig. 1 legend.

Minor:

- Didn't the number of malaria death climbed again to 620,000?

Changed to "249 million infections and 608,000 deaths were observed in 2022"

- Figure 1a – why did you not add the gametocyte expression?

The transcriptomics data for *P. falciparum* gametocytes available on PlasmoDB is not reflective of the results we observed for Western blot analysis for FIKK kinases. Therefore, gametocyte expression data was taken from the Malaria Cell Atlas, which mirrors the data better. However, the Malaria Cell Atlas uses single cell RNA sequencing to investigate gene expression with significant variability between cells. We therefore preferred a binary classification of "expressed" or "not expressed". These data can be found in Supplementary Table 6.

- Line 219 – evolved unique functions – I would argue that they are still FIKK kinases but phosphorylate different proteins. See also L 633 – please comment.

We are not sure we understand the comment: Yes, they are all FIKK kinases which evolved to phosphorylate different targets, and therefore likely have unique functions.

- Line 203 – why reference Fig 3a? The text talks about predicted active PfFIKK, however fig 3a is an AA conservation plot.

In line 302, We reference now Extended Data Fig.2. The figure is still the conservation plot. This is to explain to the reader our strategy to decide where the kinase domain of each of the FIKK kinases starts.

- In figure 4b you show genes from P gaboni G01. But in the methods you mentioned that you used Plasmodium gaboni strain SY75.

Thank you for pointing this out. This has now been corrected by mentioning P gaboni G01 in the methods.

- Where did you use the Mok et al data (Line 830)

The Mok et al data is used in Supplementary Table 4.

- Line 1498 – don't forget to put the link for the code in the next submission.

Code 9EMY has now been added to the manuscript

Rebuttal:

We thank all reviewers for their valuable time and comments. Point-point responses below.

Reviewer Comments:

Reviewer #1 (Remarks to the Author):

The authors provided a comprehensive and satisfactory response to most comments and requests. They chose to decline to follow some of the reviewer's recommendations, but in these cases the justification they provided was perfectly adequate.

I recommend publication of the paper -this will be a hallmark paper in the unfolding story of FIKK characterisation.

Thank you very much!

Reviewer #3 (Remarks to the Author):

Overall, the paper is nicely improved.

Concerning the evolutionary analysis: The dN/dS analysis taken from the cited paper just shows that nothing really happens in the FIKK family. It might be interesting to see that the two FIKK with the strongest signal of purifying selection are the FIKK4.x - which have known functions. But overall the authors provide a test and describe the evaluation, as indicated in the title.

However, they don't test the idea of geographical differences. There they fall back on the pseudogenisation. I wonder, if sequences should be curated carefully, as in SEA, many genomes have gone through recent bottlenecks and might influence the signal. Also, if the authors want to dive deeper into the pseudogenisation, it might be necessary to see if indels might have changed the open reading frame. On the other side this is not an in-depth evolutionary analysis.

About geographical differences:

MalariaGEN P.f. genomes dataset is by far the best publicly available resource for P.falciparum population genomics. However, the main caveat is that, these genomes are not necessarily representative of the parasite population. Only the location and date of samplings are available. Sampling is not controlled for host age, sex, symptoms etc... so a sampling bias is unavoidable (e.g. a MalariaGEN partner might have sent 50 parasite isolates from Cerebral Malaria cases and 50 from Mild Malaria cases for sequencing, this might not be representative of P.f. population in that endemic country). Perhaps even more importantly, the heavy usage of ACT (Artemisinin Combined Therapy) in South East Asia has likely greatly reduced the P.f. genetic diversity in the area.

This is why the prevalence of stop codons and deletions we observe here is an indication of the *P. falciparum* genetic diversity worldwide, but a true comparison of two locations would require a carefully controlled study design.

We have made the following modifications in the Results (italics).

Interestingly, 11.44% (137/1197) and 12.05% (107/888) of African and SEA samples, respectively, have *fikk14* deletions, so the preponderance for stop codons in *fikk14* in African isolates is not observed for gene deletions (Supplementary Table 4).

However, prevalences from SEA should be interpreted with caution as anti-malarial drugs have recently reduced the P.f. population genetic diversity.

About deletion and open reading frames:

The deletion of *FIKK14* we are describing here is a complete deletion of the entire gene.

So, the one thing I wonder, if for the following statement in the discussion:

We observe notable differences in pseudogenisation between geographical backgrounds, suggesting that the environment might impact *FIKK* relevance.

You want to have a test, to see if the difference is statistically significant.

We have modified this sentence in the discussion:

*We observe notable differences in pseudogenisation between geographical backgrounds, suggesting plasticity in *FIKK* importance.*

Minor:

Should the reference to the figure in line 386 be figure 3D?

Thank you for spotting this mistake. This has now been corrected to figure 3d.

Reviewer #4 (Remarks to the Author):

We thank reviewer 4 for the thoughtful review of the structural part of the manuscript.

The work by Belda et al is a very nice and comprehensive study focusing on a subgroup of protein kinases that play important roles in *Plasmodium* and related protozoa.

I shall focus within the limits of my expertise, only to analyze some aspects of the structural biology work and analyses.

I believe some modifications are needed to warrant an accurate model to be reported and deposited in the PDB. The accuracy in processing and refining the crystallographic data and model might also improve the electron density maps (eventually useful to further confirm and/or extend biologically relevant conclusions).

Concerns and comments:

1. I believe the main conclusions concerning the crystallographic (and overall structural) data, are valid.

However, the crystal structure has been processed and refined inaccurately. Materials and Methods details with regards to how these were done are also extremely limited for any reader to understand and/or reproduce.

We believe that the level of information provided in the manuscript follows what multi-disciplinary journals generally show for space constraints. Only specialised

crystallography journals will have full details of the refinement strategies used. The relevant information is available in the downloadable pdb data and within the manuscript and supplemental files. We are of course happy to provide all potential missing information and data to specialists upon request.

This has resulted in several significant errors, that once corrected, will probably improve statistics, and perhaps even locally, improve electron density features that may now be erroneous.

After careful analysis of the atomic coordinates and observed/refined structure factors that the authors kindly shared, it seems clear to me that the data were indexed in the wrong space group. I humbly believe this is not a monoclinic crystal with angle beta so close to 90°, it simply is an orthorhombic crystal.

This potential error is what likely explains that the PDB validation report detects a “significant” degree of twinning (see Section 4 within the report, with Xtrriage detecting a twinning fraction of ~0.42 following twinning law h,-k,-l).

If instead the authors, for some valid reason –in which case they should clearly report about it in Materials and methods–, truly believe this is a twinned monoclinic crystal, then they should also of course take good consideration of it, to properly refine taking into account the twinning law.

This may all explain the rather high Rfree value and an important Rfree-Rwork gap for this resolution.

Several important geometry issues have also been left uncorrected (such as several cis peptide omega dihedrals for non pre-Pro residues, a pretty high number of bad fitting residues within density or RSRZ outliers, etc etc etc).

Also, several side chains (and sometimes individual atoms within some side chains) that the authors omit, are clearly interpretable, working through the difference Fourier maps: they should be added into the final model and refined.

I suggest the authors to reprocess the data after indexing in SG P212121 (instead of the monoclinic P21 that the authors chose). And, to invest more careful work in manually rebuilding the atomic model, such that no major errors are left uncorrected. After what, corroborate any biologically relevant conclusions that may be drawn from the structural analysis.

The data was originally processed automatically in P212121, but refinement in this space group stalled with unacceptable statistics. Compared to the chosen P21 space group, refining in P212121 leads to less of the polypeptide being visible and we therefore respectfully disagree with the reviewer’s view that “the crystal structure has been processed and refined inaccurately” as a space group is only a hypothesis and the data is better explained by relaxing from orthorhombic to monoclinic.

We added the sentence “Data was processed with the XIA2/DIALS pipeline in space group P21 with cell dimensions a=82.4Å b=121.7Å c=151.1Å, $\alpha=90.0^\circ$ $\beta=90.02^\circ$ $\gamma=90.0^\circ$.” to the material and methods.

The twinning test mentioned is only one of many and not a good indicator when a crystallographic symmetry operation has been replaced by non-crystallographic symmetry.

Xtrriage analysis states "One or more twin operators show a significant twin fraction but since the intensity statistics do not indicate twinning, you may have an NCS rotation axis parallel to a crystallographic axis" which we find to be the case here. It is important to note that this is not a twinned lattice, but some of the twinning tests will misreport in this sort of case and twinned refinement is therefore not the best approach.

The high R-free and R-work to Rfree gap are consequences of the considerable percentage of unordered polypeptide. Both statistics are worse rather than better when refinement is undertaken in the orthorhombic space group, rather than in monoclinic. Additionally, given the 2.8 Angstrom overall resolution neither the gap nor the absolute values are unexpectedly high. There is also considerable anisotropy in the data (which shows in both the monoclinic and orthorhombic space groups) and this will also not be helping with the Rfactor statistics. Refinement shouldn't be a case of chasing particular statistics but generating as good a model as possible given limitations in the data.

There are no non pre-PRO cis-peptides according to the analysis in Coot and with other tools.

Concerning the density for sidechains observed by the reviewer, it does not mean that the density remains after refinement. Given that the evidence for the position of these atoms is not strong after iterative refinement with them present and absent, they have (mostly) been removed. Likewise with the other geometrical outliers, attempting to resolve many of these leads to more issues with further refinement. With better data, many more of these could be properly resolved but the model submitted to the PDB is the best available dataset and combining data from multiple crystals has not helped.

2. Please use pLDDT scores (instead of TM) to quantify the reliability of AlphaFold-predicted structures. For instance, in Supplementary Table 11, and consistently throughout the text.

The Template Modelling (TM) score is generally considered to be relatively insensitive to local inaccuracies (Xu 2010 <https://doi.org/10.1093/bioinformatics/btq066>). Actually, higher than 1Å rmsd values (for some kinase pairs even >2Å) as the authors report for the superpositions, indicate that, while definitely reliable in terms of fold (which is actually an expected outcome when it comes to predicting ePKs), they will most certainly include pretty significant variations at the local level.

We agree with the reviewer and in Supplementary Table 11 have now replaced the TM-Score with the mean pLDDT for each FIKK kinase domain. The mean pLDDT is generally high (average of 88.8 across all FIKKs).

On the other hand, pLDDT (predicted local distance difference test) scores can be reported as an average value (to replace TM in Supp Table 11), but also convey greatest information when reported as a per residue index. A typical way to do this is to color the AF2-predicted models that you show in main and supplementary figures, with a ramp according to pLDDT (this score is saved in the AF2 models on the B-factor column, so that it is easy to color by this property). Doing so, any regions/loops that would happen

to have pLDDTs ≤ 70 (which AlphaFold will classify as being low) should be considered cautiously.

We agree with the reviewer recommendation and below we have recreated Figure 5A using the standard AlphaFold pLDDT colouring scheme:

Extended Data Figure 7 has also been updated in the same way, with all AlphaFold2 models coloured according to the residue pLDDT.

Finally, in the main text we have removed any mention of the TM-Score (as we now exclude this) and state that the average pLDDT (88.8) of the protein kinase domain was high. In the previous version of the manuscript, we had already mentioned that our validated specificity determinants map to low pLDDT regions of the AlphaFold models:

Lines 497-500:

‘the validated determinants FIKK12
K212 and K263 map to kinase loop regions. These loop regions are rapidly evolving (Extended Data Fig. 9) and likely flexible given their low pLDDT scores in the AlphaFold models’

With the new pLDDT color bar in Figure 5A, it is now clearly visible in Figure 5B that these determinants (K212 and K263) map to low-confidence regions.

We thank the reviewer for making these important suggestions.

3. Please do not include explicit hydrogen atoms at this resolution, as you have currently done. I guess this might have been the result of an unwanted “default” setting in your refinement program. It is certainly a good practice to add H atoms in riding positions while refining, but not to actually leave them and write them within the final mmcif file (at 2.8Å resolution this doesn’t make sense, and drives overfitting due to exceedingly high parameters/observables ratio).

Riding hydrogens are retained because this is the model that was refined. Given the fairly low-resolution, riding hydrogen positions are important to maintain geometry and minimise clashing between heavier atoms and retaining them between cycles gives more stability. We mention in the coordinate files that these are riding hydrogens meaning that their positions are only geometrically restrained and they are not used in refinement against the data.

4. The loop 194:204 (not included in the crystal model because of lack of electron density), is positioned such that it protrudes from the N-lobe towards the C-lobe of the ePK domain. Even though this loop might be mobile (hence the weak density; and, perhaps low pLDDT scores in AlphaFold predictions?), it will be covering the catalytic site due to its topologic position. It somehow resembles to an analogous N:C-lobe interaction as the one observed in Tyr-kinases like IRK (e.g. pdb 4IBM). Could this loop and/or its characteristics (length, properties) be linked to Tyr vs Ser/Thr specificity?

We thank the reviewer for their insightful comments. This loop is missing from the FIKK13 crystal model (as noted), and so we use AlphaFold models to address the reviewer question.

First, we confirm that the 194:204 loop is of low confidence (i.e. low pLDDT) in the FIKK13 AlphaFold model:

FIKK13
(kinase domain)

In line with the reviewer suggestion, the model is coloured by the pLDDT confidence score: orange is pLDDT < 50, yellow is 50 < pLDDT < 70, light blue is 70 < pLDDT < 90, and dark blue is pLDDT > 90.

The role of this loop on phospho-acceptor specificity (Y vs S/T) is difficult to determine structurally due to the low average pLDDT, implying flexibility in the loop position and conformation.

However, if it were a strong determinant of Y specificity, we would expect it to be present in FIKK13, but not in the other characterised FIKKs (which are mostly S/T-specific). In fact, we find similar predicted N-lobe protruding loops in other S/T FIKKs such as FIKK4.1 and FIKK7.1.

The N-lobe loop of FIKK1 (S/T) and FIKK13 (Y) are especially similar:

FIKK13

The FIKK13 kinase domain shown previously is in grey, and the FIKK1 kinase domain is now colored by the pLDDT confidence score. The length and conformation of the loops are similar but the composition is different (FIKK13: EKDSKIIYTWDL, FIKK1: DIWVEQFNLMK), albeit with a net charge of -1 in both cases.

Given these similarities between the FIKK13 (Y-specific) and FIKK1 (S/T-specific) loops in the N-lobe, at this point we do not infer strong evidence of this loop acting as a direct determinant of Y kinase specificity. However, we cannot rule out the possibility that the amino acid composition of the loops affects their phospho-acceptor specificity. The low confidence (low pLDDT) loop regions in the AlphaFold models are limiting for further interpretation. Further discussion is directly below.

Following up on this point, and converging with a colleague reviewer's question, even though there is not an identical crevice as in IRK, it's not obvious to me that a deepened cleft can be ruled out. I don't know exactly how does FIKK12 look like (which the authors used in their response to compare with a S/T-specific variant; again what about pLDDT scores for relevant structural components when predictions are used?); but in any case it is very useful for these matters, to look at solvent accessible surface representations (instead of cartoons), getting a better grip of clefts, pockets or even enclosed cavities (which could open up by movement of loops/residues).

We indeed agree that a surface representation of the relevant grooves would have been more informative in this instance.

First, we show a surface representation of **FIKK12**, which we compared with FIKK13 in the previous round of revisions:

The pLDDT colouring scheme is the same as described above, and the catalytic aspartate position (**HLD**) is coloured in purple.

We next show the same representation but for **FIKK13**:

The surfaces on the C-lobe are similar between FIKK12 and FIKK13, but the N-lobe protrusion (described above) may enclose a deeper cavity for the phosphoacceptor in FIKK13. The clear caveat to this observation is that the N-lobe loop in FIKK13 is modelled with low confidence (low pLDDT, orange colour).

As stated above, the N-lobe loop (194:204 in FIKK13) is more similar in FIKK1 and so below we give a surface representation of the **FIKK1** AlphaFold model:

In the conformations shown, the FIKK13 N-lobe loop may enclose a deeper cavity than the N-lobe loop of FIKK1, while the C-lobe surfaces are similar. However, since the FIKK13 and FIKK1 N-lobe loops are both modelled with low confidence, further extensive investigation would be required to determine if differential loop dynamics

(between FIKK13 and FIKK1) have an effect on phosphoacceptor specificity (Y vs. S/T). We agree that this question merits further investigation. We continue to try to identify a FIKK13 substrate for co-crystallisation, which we believe is ultimately needed to identify the molecular basis for Tyrosine specificity. Because we cannot confidently make statements about Tyr-specificity of FIKK13 using molecular modeling, we refrained from discussing these results in this paper, but hope to answer the question, with experimental validation, in the future.

I actually observe such a cavity in FIKK13, poised to open just beside the catalytic Asp. This cavity is delimited by N257, L252, E426 on one side, and Q401, I429, L378 on the other (and, Leu378, interestingly, is replacing the highly conserved Arg in the catalytic site-containing fragment known as 'HRD', here 'HLD' in FIKK13. Can you please comment (considering of course other S/T-specific ePKs)?

We thank the reviewer for this suggestion. Below we have highlighted (in red) the listed molecules on the FIKK13 crystal structure of the kinase domain:

For reference, the catalytic position is coloured in purple and the α B- α C loop is coloured in green. We observe from this visualisation that a large part of the cavity is occluded by the α B- α C loop. This is confirmed using a surface representation:

Here we use the standard pLDDT colouring scheme on the AlphaFold model of **FIKK13**. The residues listed by the reviewer are again coloured in red. The light blue surface covering the red residues is the α B- α C loop, suggesting that this occludes the cavity. Confidence in the α B- α C loop structural conformation is relatively high, implying limited flexibility of the loop.

In comparison to other S/T kinases, we observe a similar cavity in **FIKK1** that also seems to be occluded by the α B- α C loop:

Taken together, we therefore think it is unlikely that this cavity is contributing directly to phospho-acceptor specificity for tyrosine (given that we see similar cavities in FIKK13 and FIKK1, and both are occluded by the α B- α C loop).

Extended Data Fig 2 does not allow to see all these sequence analyses/comparisons: it would be worth having a clearer and complete MSA of FIKKs as extended data figure. I did also read carefully the response to other reviewers, including an MSA with full length FIKKs: I admit fonts are extremely small there, and of course including the complete sequences makes it more difficult to focus on the ePK domain. Maybe my previous questions are already responded therein, I apologize if that's the case, yet I cannot be clear with the elements at hand.

Extended Data Fig. 2 has now been changed to show a complete multiple sequence alignment focusing on the FIKKs' kinase domain. The font has been increased as well. We hope this makes the figure more readable.

5. You mention the 'HRD' motif substitution by HLD (as well as the modification of the conserved 'DFG' motif), in the caption to Ext Data Fig 6, but actually consider them to be "minor changes". I tend to disagree with that. HRD and DFG motifs play key roles, explaining why they are so well conserved among a vast number of ePKs.

We apologise for any confusion caused. By 'minor' we mean that these motifs do not involve mutations to the critical catalytic residues i.e. aspartate in HRD and aspartate in DFG and so the FIKK kinases are still (evidently) active. However, we see that this wording could be misleading and agree that the overall motifs are important for kinase regulation, as explained in Gizzio et al 2024 (Nat Commun doi:10.1038/s41467-024-50812-0). To avoid confusion, we have replaced the word 'minor' with 'notable changes'.

Inspecting a structure-based alignment of all human kinase domains (from Modi and Dunbrack, 2019, PMID: 31875044) reveals that the full HRD motif is found in 74.4% of all sequences after removing pseudokinases. The YRD motif (n=44) and HLD motif (n=18) can also be tolerated in some families. The DFG motif is found in 85.5% of all sequences after excluding pseudokinases. The DLG (n=34) and DYG (n=10) motifs can also be tolerated in some families.

We would also note that the FIKK kinases are an Atypical kinase family and highly sequence diverged from canonical ePKs. Differences in the overall kinase fold are described in the legend of Extended Data Figure 6. Therefore, the effect of any given mutation in a kinase motif may differ dramatically between the FIKK family and ePKs. In other words, the effect of any motif mutation will also depend on the background kinase domain sequence, due to epistasis and resulting co-evolution between substitutions in the critical motifs and the rest of the kinase sequences. Substitutions in the FIKK activation/catalytic loops that appear destabilising may therefore have been compensated by mutations elsewhere in the kinase domain during evolution. Further discussion is directly below.

6. Please discuss your findings in view of the recent publication by Gizzio et al 2024 (Nat Commun doi:10.1038/s41467-024-50812-0). Are there coincidences or contradictions? Coming back to the DFG motif analysis just referred to above (with a "locked active"

configuration in the case of FIKK13), this discussion with an evolutionary context seems particularly useful.

Indeed this is an interesting and relevant paper that was published recently. To summarise, Gizzio et al., used statistical models of sequence variation and molecular dynamic simulations to understand why the distribution of ‘active’ and ‘inactive’ conformational states generally differs between serine/threonine and tyrosine kinases, with tyrosine kinases tending towards auto-inhibition. They conclude that substitutions surrounding three structural motifs contribute to this divergence between ST kinases and Y kinases. These structural motifs are centred on the kinase activation loop and the catalytic loop containing the ‘HRD’ motif.

Overall, we find little consistency between the findings of this paper and what we observe for the FIKK family of kinases. Specifically:

For the N-terminal activation loop anchor, the observed ePK S/T kinase pattern at HRD-2 and DFG+2 is observed in only 16.7% of our FIKK S/T kinases, compared to in 77% of ePK S/T kinases.

For the RD-pocket structural motif, a small aliphatic side chain is found at position APE-7 in ePK S/T kinases but this is the case for only 38.9% of FIKK S/T kinases.

Finally, for the C-terminal activation loop, we observe no (0%) correspondence between the patterns observed for ePK S/T kinases at positions APE-5 (S or T in ePK S/T kinases) and HRD+2 (K in ePK S/T kinases), and what we observe in the S/T FIKKs.

To give context to these findings, we should note that the architecture of the FIKK activation loop has many differences from a canonical ePK activation loop, as noted in the legend of Extended Data Figure 6. In particular, the DFG motif has been mutated in most FIKKs (‘DLS’ in FIKK13), the APE motif is replaced with PPE, and the HRD motif of the catalytic loop replaced with HLD. Also, the FIKK13 activation loop is supported by three additional alpha helices not seen in canonical ePKs.

Connected to this point, the FIKK kinase family is Atypical and the kinase domain is highly sequence diverged from canonical ePKs. Also, the canonical tyrosine kinase family that emerged in the ancestors of animals and their unicellular relatives, are very distantly related (evolutionarily) to the FIKK family that is found exclusively in the Apicomplexan phylum. Taking these facts all together, the low correspondence that we observe between the Gizzio results and the FIKK sequences is expected.

Finally, we note that we have a sample size of 1 known FIKK Y kinase and 14 known S/T kinases (albeit some with Y dual specificity). A full statistical analysis of FIKK sequence variation is not currently possible with this sample size. We believe that a co-crystal is probably the only way to ultimately determine the basis for tyrosine specificity of FIKK13. We have now cited Gizzio et al in the manuscript, but refrain from a detailed discussion because we are already at the maximum space limit of the journal, and

believe that a detailed comparison with this manuscript will be better placed when we have an actual co-crystal in a later publication.

7. Please consider the possibility that the APTgS moieties have been partially/fully hydrolyzed. Electron density seems weaker on the gamma P center; and, please remember that sulfur should be sitting on a Fourier electron density peak at least as large as the phosphorus atom, since they both have approx. same number of electrons. Another evidence that this group has been hydrolysed (at least partially: occupancy can be refined). ATPgS is a slowly hydrolyzable ATP variant; for actual inhibition you may want to prefer using AMP-PCP (a methylene replacing the bridging O between beta and gamma P). Again, any analyses such as these should be made once refinement is properly finished, as phases might improve quite a bit, and then maybe the thio-derivatized gamma phosphate is indeed clearly there at full height, I cannot be sure right now; also, the nucleotides show right now a rather weird discontinuity, which in the end may be corrected. Last but not least, normally ATP and derivatives are complexed with a Mg²⁺ cation: consider this in your refinement (or else, comment).

It is indeed possible that some hydrolysis of ATPgS has occurred and we thank the reviewer for suggesting AMPCP for future experiments. However, we are confident the position of the ATPgS is accurate and does not lead to erroneous interpretations.

Rebuttal:

We thank the reviewer for clarifying their point-of-views, which we address below. We hope this clarifies the analysis pipeline and choices taken.

1- Concerning my request for crystallographic methods to be better explained.

The authors replied that they think it is OK what they presented.

My only job as a reviewer is just to indicate that, going through the Methods section ("Crystallisation of PFIKK13149-561_D379N with Nb2G9, Nb9F10 and ATPγS") as it's written, one cannot actually reproduce what was done. In the revised version, the authors added the programs they used to do MR and data processing. Yet nothing about why they chose P2₁ (even though orthorhombic was indeed an option as they explain now), or how they proceeded to do the refinement and validation.

I am not asking for specialized details, just the basics of the software AND protocols you used to process the raw diffraction and then to refine your proposed model.

In other words, it is not up to me to establish the editorial requirements. Just to indicate whether the results can be reproduced, that's all I did.

Response: To provide the reader and reviewer with a rationale for choosing the P2₁ Space group, we have added the sentence "Molecular replacement with PHASER, in the resultant orthorhombic (P2₁2₁2₁) space group and utilising the apo crystal complex coordinates yielded a poorly refinable solution and the ATPγS complex data was therefore reprocessed with the same pipeline in space group P2₁ with cell dimensions a=82.4Å b=121.7Å c=151.1Å, α=90.0°, β=90.02°, γ=90.0°" to the Material and Methods section.

We have also added further detail on the software used for processing and refinement "Model building and refinement were undertaken with COOT version 0.9.8 and REFMAC version 5.8.0430 within CCP4CLOUD".

To provide a bit of further background on the history of events that led to the space group decision; we had obtained, in addition to ATPγS diffraction data presented in the current manuscript, other diffraction data-sets (not included in the manuscript) from apo crystals and from crystals with other ligands (AMPPNP and a small molecular weight inhibitor - Emodin). These datasets happily refine in P2₁2₁2₁. Whilst, for the ATPγS, the apo co-crystal search model provided a solution, the data did not refine comparably and it was decided to reprocess the data in the lower space group P2₁, where the refinement was more stable. To us, this is evidence that the ATPγS form was slightly different and that the symmetry had been broken. This can be seen with some differences in the interfaces between macromolecules, as well as completeness of the polypeptides which would be identical in the orthorhombic cell as described in further detail below. We would also like to point out that processing in a lower space group is not wrong, but doing so may confuse some of the tests used to validate the space group. Modelling in a monoclinic space group has doubled the effort required during manual model building and wouldn't have been chosen had it not been deemed necessary given the data.

Both space groups do give similar statistics in the processing but, for example, Chain B is more 'complete' than the equivalent Chain H and when processed in P2₁2₁2₁ it is difficult to model this chain equally well.

For example, the interactions of the more complete loops of chain B (compared the equivalent chain H) with the symmetry related molecules.

Here, the distance between the CA of residue Thr 344 from FIKK to a symmetry partner nanobody are as follows:

Thr 344 CA Chain B – Thr 123 CA Chain I (symmetry copy) distance 5.47 Å and the following loop of chain B is complete.

Thr 344 CA Chain H – Thr 123 CA Chain C (symmetry copy) distance 6.18 Å and here the density for the following loop is weaker and so not modelled.

A 0.71 Å difference doesn't seem a lot but is present all along this interface and in our view, similar subtle shifts across the whole model are sufficient to break the crystallographic symmetry present in the Orthorhombic cell.

From the PISA analysis, this interface (B to symmetry I or H to symmetry C) has the most favourable ΔG of those involved in crystal contacts despite there being no hydrogen-bonds or salt bridges involved, so a subtle shift could be sufficient to reduce the symmetry.

The mention of the ATP γ S as being a potential cause is due to the well-known issue with the phosphate group(s) being hydrolysed even when stored as a solid. Modelling intact ATP γ S gives negative density at the phosphate positions, hence it has been modelled with partial occupancy, and this may also be affecting the packing, breaking the symmetry present. Given the resolution, it is hard to show this in an obvious way.

2- Concerning my strong suggestion to reprocess the data in space group 19, to see whether that would generate a more accurate refined model.

The authors disagreed and decided not to do it. They argue that the data are indeed monoclinic. The reason they put forward is that in P212121, refinement would stall with unacceptable statistics (probably referring to too high R factors, I would guess).

I thank the authors for having shared data and coordinates. The best way to reply from my side is to attach a properly reprocessed data set (using the program Zanuda from CCP4, as a means to quantitatively test all groups compatible with the Laue assignment that derives from the unmerged intensities; Zanuda also performs a quick Refmac refinement in all subgroups that produce low-enough Rmerge statistics). I also did a very quick refinement with Phenix afterwards, with the reprocessed data in SG 19 (and maintaining the authors' choice of TLS refinement, although with slightly different TLS groups chosen: this is not critical anyhow, different TLS group choosing have marginal effect): quite standard protocol: xyz refinement, B factors by atom, local NCS restraints, automatic choosing of TLS groups, optimization of geometry/Rfacs scaling.

Rfree and Rwork, as well as model stereochemistry, were all significantly better in SG P212121, than the ones reported by the authors. This fact demonstrates the crystal is indeed orthorhombic.

Again, please do not consider my job as trying to raise obstacles. My intention is to serve as a double-check, so that models, as accurate as possible (using state of the art software) are the ones being published and deposited in the PDB.

The authors state "The twinning test mentioned is only one of many and not a good indicator when a crystallographic symmetry operation has been replaced by non-crystallographic symmetry." This is a mistake, or the authors misinterpreted what I was suggesting.

If I raised a potential twinning as the reason for their choosing a lower symmetry space group (as the PDB validation report actually brings in, signs of significant twinning: which is absolutely expected given the higher symmetry space group that we now know is true), it was as a means of being open to such a possibility.

If the higher symmetry I originally suggested to explore for, would be the consequence of NCS, it would never allow for smooth convergent refinement with good R factor residuals and geometrical quality. I don't know why the authors' refinement protocols stalled at unsatisfactory levels (they don't describe any refinement protocols whatsoever).

As a further proof, if NCS were to explain the additional 2-fold axes (just by chance falling into a perpendicular direction to the unique 2-fold), a self-rotation analysis of the data would not bring up 2-fold peaks that have the same height as the crystallographic 2-fold that the authors refined with:

I attach self-rotation plots and the beginning of the table with the signal/noise values (calculated with Molrep). The first plot (top right) represents the $\chi=180^\circ$ section of the rotation function, from the data processed in P21: the same level peaks are readily seen at the equator along the crystallographic a axis (parallel to x according to Molrep's convention), and of course a third 2-fold appears at the center perpendicular to the plane of the paper. This is pathognomonic of a 222 point group. If it were a monoclinic, non-crystallographic symmetry axes at 90° of it would be weaker in intensity (by the way, some 2-fold NCS axes can indeed be seen, but they have <6-fold less signal/noise compared to the crystallographic ones in this case).

All this analysis has nothing to do with intensities' distribution analyses (the so called "twinning tests"), but rather consequences of intrinsic symmetry of the crystal.

Again, I do not believe there are major modifications to the biological conclusions because of this mistake. Yet, it is important that the authors, who have the biological expertise, double-check this with a properly solved and refined structure.

Having said that, and not being myself a specialist in apicomplexa kinases, the re-refined model tends to indicate that the gamma phosphate is not present on the bound nucleotides (I am aware that the authors used a normally non-hydrolysable variant AGS, yet...perhaps slowly hydrolysable as they agree). The sulfur atoms of their derivative are excellent markers, since S atoms scatter X-rays significantly more strongly than oxygen atoms, so that S on g-P should give a larger Fourier peak if it were there.

- I would also add the Mg²⁺ cation complexed to the phosphates, it appears visible in density. Check links with neighbor atoms as I did only an automatic assignment with phenix.metal_coordination.
- Many side chains became visible in the maps, and I only did a very quick refinement.
- Please check whether any additional information of biological relevance can be extracted.

Last: the authors were right that there are no cis peptides, apologies for that mistake of mine (Coot 0.9.x now signals "cis" peptides on each first residue after a gap...which is certainly a bug due to the way it calculates it)

Response: We thank the reviewer for taking the time to explore the space group issue with our coordinates. As stated in 1) – we found differences between the chains and also, with due respect, *argue that starting from a model built in P2₁ and switching to P2₁2₁2₁, will improve the density due to introduction of some model bias in the maps.*

We also thank the reviewer for highlighting Ramachandran outliers. We have gone through our model and corrected Ramachandran outliers as pointed out by the reviewer. The latter model has been updated in the PDB and a new Table 17 and validation report is submitted and attached with this letter. We did not find density for Mg²⁺ and decided not to add it to the model.

Since the contribution of the first experimentally determined structure was to demonstrate the kinase fold, we didn't – upon re-examination of the model - find changes of biological relevance beyond what was concluded initially – in line with the reviewer's conclusion with regards to the impact of the choice of space group for data processing.

3- Concerning my advice to avoid depositing hydrogen atoms at this resolution.

The authors did not agree. I must insist. What I absolutely agree with, is that, as the authors sensibly say, adding H atoms at riding positions is a very good practice (especially at lower resolutions). But that is done during refinement (as well as at validation time, particularly to calculate clash scores). You do not want to keep these atoms explicitly in the final pdb file. The authors actually avoided including side chains altogether if they are not seen in density: how would one keep H atoms? This leads to misuse afterwards by unaware PDB users. The authors probably used an option “by default” of some sorts, that keep the H atoms in their riding positions: it is just not a good idea at this resolution.

Response: Following the reviewer’s suggestion, we have removed the hydrogens from the refinement and the model deposited in the PDB as per attached documentation report.